# Dietary L-Glu sensing by enteroendocrine cells adjusts food intake via modulating gut PYY/NPF secretion

Junjun Gao [1], Song Zhang[1], Pan Deng[2,6], Zhigang Wu [2], Bruno Lemaitre [3], Zongzhao Zhai[4] ✉ & Zheng Guo [1,5] ✉

Amino acid availability is monitored by animals to adapt to their nutritional environment. Beyond gustatory receptors and systemic amino acid sensors, enteroendocrine cells (EECs) are believed to directly percept dietary amino acids and secrete regulatory peptides. However, the cellular machinery underlying amino acid-sensing by EECs and how EEC-derived hormones modulate feeding behavior remain elusive. Here, by developing tools to specifically manipulate EECs, we find that *Drosophila* neuropeptide F (NPF) from mated female EECs inhibits feeding, similar to human PYY. Mechanistically, dietary L-Glutamate acts through the metabotropic glutamate receptor mGluR to decelerate calcium oscillations in EECs, thereby causing reduced NPF secretion via dense-core vesicles. Furthermore, two dopaminergic enteric neurons expressing NPFR perceive EEC-derived NPF and relay an anorexigenic signal to the brain. Thus, our findings provide mechanistic insights into how EECs assess food quality and identify a conserved mode of action that explains how gut NPF/PYY modulates food intake.

Proper dietary protein intake has been increasingly recognized to promote growth and enable health and life span benefits[1]. Malnutrition due to insufficient protein consumption causes growth retardation and body wasting coupled with severe multiple tissue damages and anorexia, while dietary restriction for proteins or specific amino acids (AAs) extends lifespan in various organisms[1,2]. Apart from their roles as the building blocks of proteins and as neurotransmitters, AAs also regulate diverse animal physiology and behaviors[3,4]. Gaining mechanistic insights into AA detection and the physiological feedback signals emanating from AA limitation or excess should inform new strategies to maintain energy homeostasis and to improve health and lifespan.

In mammals, AA-sensing mechanisms have been deployed at multiple levels to detect the presence and survey the abundance of AAs available, further informing the nervous system on whether and how much to eat. First, taste receptors can detect AAs in the environment[5]. After ingestion, AAs become systemically available to organs and tissues and are further surveyed by cellular sensors including mTOR and GCN2[6,7]. During digestion, digested dietary proteins are presented to intestinal epithelial cells where both absorption and concurrent food content evaluation take place[8]. Scattered throughout the intestinal epithelium, enteroendocrine cells (EECs) are directly exposed to and sense luminal nutrients[9,10]. In response to AAs, EECs release neuropeptides[11,12] and modulate food intake[13–16]. In particular, enteroendocrine L-cells rapidly release the anorexigenic hormone, peptide YY (PYY), following ingestion of protein-rich food to avoid overeating[17,18]. EEC-derived PYY exerts its function through the

[1]Department of Medical Genetics, School of Basic Medicine, Institute for Brain Research, Tongji Medical College, Huazhong University of Science and Technology, Wuhan, China. [2]State Key Laboratory of Digital Manufacturing Equipment and Technology, Huazhong University of Science and Technology, Wuhan, PR China. [3]Global Health Institute, School of Life Sciences, Ecole Polytechnique Fédérale de Lausanne (EPFL), Lausanne, Switzerland. [4]State Key Laboratory of Developmental Biology of Freshwater Fish, College of Life Sciences, Hunan Normal University, Changsha, Hunan, PR China. [5]Cell Architecture Research Center, Huazhong University of Science and Technology, Wuhan, Hubei, China. [6]Present address: Department of Mechanical Engineering, University of British Columbia, Vancouver, British Columbia, Canada. ✉e-mail: zongzhao.zhai@foxmail.com; guozheng@hust.edu.cn

NPY family receptors expressed in the vagal afferent neurons[19], part of the enteric nervous system (ENS) that surveys the gastrointestinal milieu and relays information to the brain[20]. In vitro studies using EEC cell lines have shown that CaSR, GPRC6A and LPR5 are all general protein sensors that induce the secretion of regulatory peptides[21]. A specific L-glutamate sensor, metabotropic glutamate receptor 4 (mGluR4), is also expressed in a murine EEC cell line with considerable overlap with PYY expression[22]. However, the in vivo cellular mechanisms by which EECs detect AAs remain unknown[10].

*Drosophila* has long been a leading model organism to uncover the fundamental principles of AA-sensing and the consequential modulation of animal behaviors and physiology in an integrated and whole-organismal manner[23]. Apart from its powerful genetics, efficient dietary manipulation has enabled accurate analysis of the effects of any single dietary AA[24,25]. The molecular basis and neuronal organization of AA perception have been extensively studied[26–28]. A broadly expressed ionotropic receptor, Ir76b, is necessary for AA preference in larval and adult *Drosophila*, and Ir76b-expressing neurons physiologically respond to AAs and yeast[29–31]. As in mammals, systemic AA levels are mainly sensed by the GCN2-ATF4 axis and the mTOR pathway, with the former principally detecting deficits of any AA and the latter being activated by only a few AAs including Leucine and Arginine[7,32]. GCN2 also plays a key role in sensing AA imbalance, a condition that is detrimental to many juvenile and adult traits[33,34]. Diverse internal cell types, ranging from the fat body cells[35–40], enterocytes[33], intestinal stem cells (ISCs)[41] to neurons and glial cells[34,35,42], have been reported to sense systemic AA availability. AA-sensing in turn enables coordination of organismal growth[35–37,40] and energy metabolism[39] with nutrient availability to adapt to nutritional environment. Remarkably, AA-sensing also instructs feeding behaviors, driving animals not only to adjust the quantity of food intake[38,40,42], but also to choose between different food qualities to meet physiological demands so as to increase their fitness[33–35]. Despite the reports that several gustatory receptors can be detected in EECs[43] and that a subset of EECs expressing Diuretic Hormone 31 (DH31)[44] and tachykinin (Tk) can be activated by AAs[45], the mechanism by which fly EECs detect AAs and the mode of action that gut peptides modulate organismal physiology and behaviors, however, remain unknown.

Vertebrates and insects share many features in the origin, specification, and function of EECs[46]. Notch signaling and bHLH proneural factors act in concert to control stem cell lineage decision and to specify EEC fate[47–50]. Notably, in mammals, EECs are specified by a bHLH factor of the Neurogenin family, Ngn 3[51], for which a single homolog, called Target of Pox neuro (Tap) is encoded in *Drosophila* genome with expression in a subset of EEC[46,52]. In both mammals and flies, the gene networks active in EECs overlap largely with those controlling the neurons, in addition to the fact that both cell types are excitable and secrete via dense core vesicles (DCVs) and synaptic vesicle (SVs)[53]. Such similarities between EECs and neurons, together with the fact that most EEC derived neuropeptides are also produced in the brain by neurosecretory cells[54–56], make it technically challenging to clearly demonstrate the function of gut-derived neuropeptides in physiological studies using genetic approaches. Moreover, whether and how the same neuropeptide of gut or brain origin differs in its physiological function requires careful demonstration. An intriguing example of such discrepancy is the mammalian NPY family peptides, with NPY from the brain promoting feeding sharply contrasting with gut-derived PYY that conveys a satiety signal[57].

Here, we analyze the role of EECs in AA-sensing by developing methods to specifically manipulate EECs without affecting the central nervous system (CNS). We first found that flies ablated for EECs (EEC-less flies) dramatically increased food intake. Both loss of NPF+ EECs and gut-specific depletion of the NPF neuropeptide recapitulated the upregulated food appetite seen in EEC-less flies. We further uncovered that NPF+ EECs directly sense dietary L-Glu via the metabolic glutamate receptor (mGluR). L-Glu sensing reduced NPF release into circulation, by slowing down calcium ($Ca^{2+}$) oscillations that underlies the secretary activity of EECs. This in turn caused a drop in systemic NPF levels and promoted feeding by reducing the activity of a pair of enteric afferent neurons that express the NPF receptor (NPFR). Finally, we found that NPFR+ enteric neurons using dopamine synapsed with neurons in the subesophageal zone (SEZ), a brain center known to control feeding, to inhibit food intake. Hence, our work uncovers a key molecular basis of AA-sensing by EECs and reports a highly conserved mode of action by which gut-derived PYY/NPF restricts appetite by acting on ENS neurons.

## Results

### Loss of EECs increases food intake

To demonstrate the role of EECs in AA-sensing, EEC-specific manipulations without affecting the development or function of other cells, in particular neurons, are highly demanding. Because EEC specification shares a common root with that of sensory neurons[46] and most EEC-derived neuropeptide hormones are also produced in the brain[54–56], none of the available Gal4 drivers allows for EEC-specific manipulations[58,59]. *Tachykinin (gut)-Gal4* (*Tkg-Gal4*) was reported to be specific to TK+ EECs[60], and has been used in a number of intestinal studies[61–66]. However, *Tkg-Gal4* was later found to drive substantial expression in brain[62,67]. In addition, a Gal80 transgene driven by an enhancer fragment (R57C10) of *neuronal Synaptobrevin* (*nSyb*) is often used in combination with Gal4 drivers to suppress Gal4 transcriptional activity in CNS, with the expectation that only EECs are manipulated[68,69]. However, the R57C10 fragment is also active in EECs[58,70]. An attempt has been made to ablate EECs by knocking down a proneural factor *Scute* (*Sc*) using the intestinal progenitor driver, *esg-Gal4*[61]. Although EEC-less adult flies are generated with this method, it is not suitable for studying adult traits, since the ISCs are also eliminated by *sc* knockdown (Extended Data Fig. 1a–c, 1a′–b′). In addition, *esg-Gal4>sc^{RNAi}* may affect the development of the nervous system (Extended Data Fig. 1d). Therefore, new tools need to be developed to study EEC function.

An alternative way to remove EECs is to combine *esg-Gal4>sc^{RNAi}* with a temporal control using the TARGET system[71] and restrict *sc* knockdown to a critical time window of EEC specification. Sc is required in ISCs for EEC specification at mid-pupal stage[47], a stage when *esg-Gal4* is not expressed in the nervous system (Extended Data Fig. 1d). Using the temperature-inducible ISC driver *esg-Gal4 tub-Gal80^{ts} UAS-GFP* (*esg^{ts}*) to deplete *sc* for 10 h at 30 °C in this pupal stage (via shifting *esg^{ts}>sc^{RNAi}* pupae between different temperatures) (Fig. 1a, see Methods), rendered midguts with less than 10 EECs in young flies (3 days after eclosion (AE)) (Fig. 1b, c). In adults, EECs were slowly regenerated from ISCs, resulting in about 100 EECs on day 7 AE and about 500 on day 10 AE (Fig. 1b, c). We refer to this pupal-phase knockdown of *sc* to prevent EEC generation as *esg^P>sc^{RNAi}*. Notably, removing EECs using this method changed neither the number of adult ISCs nor the rate of ISC division (Extended Data Fig. 1e–h).

As impaired AA-sensing is often associated with abnormal feeding, we measured food intake of *esg^P>sc^{RNAi}* mated female flies using the Capillary Feeder (CAFE) assay[72]. *esg^P>sc^{RNAi}* flies ingested significantly more at 3 day AE compared with control flies of six different backgrounds (Fig. 1d). We also used a dye-based food intake measurement to examine the feeding levels of EEC-less flies on a standard cornmeal diet (SCD)[38,73]. Our results show a significant increase in the amount of *esg^P>sc^{RNAi}* flies feeding on SCD at 3 day AE compared to controls (Fig. 1e). In the following experiments, unless otherwise stated, we measured food intake using the CAFE assay.

Along with the gradual recovery of EECs, food intake of *esg^P>sc^{RNAi}* flies dropped to the level of control groups by 10 day AE (Fig. 1d). To further demonstrate that EEC loss was responsible for the rise in food intake, we continued to prevent EEC regeneration by placing

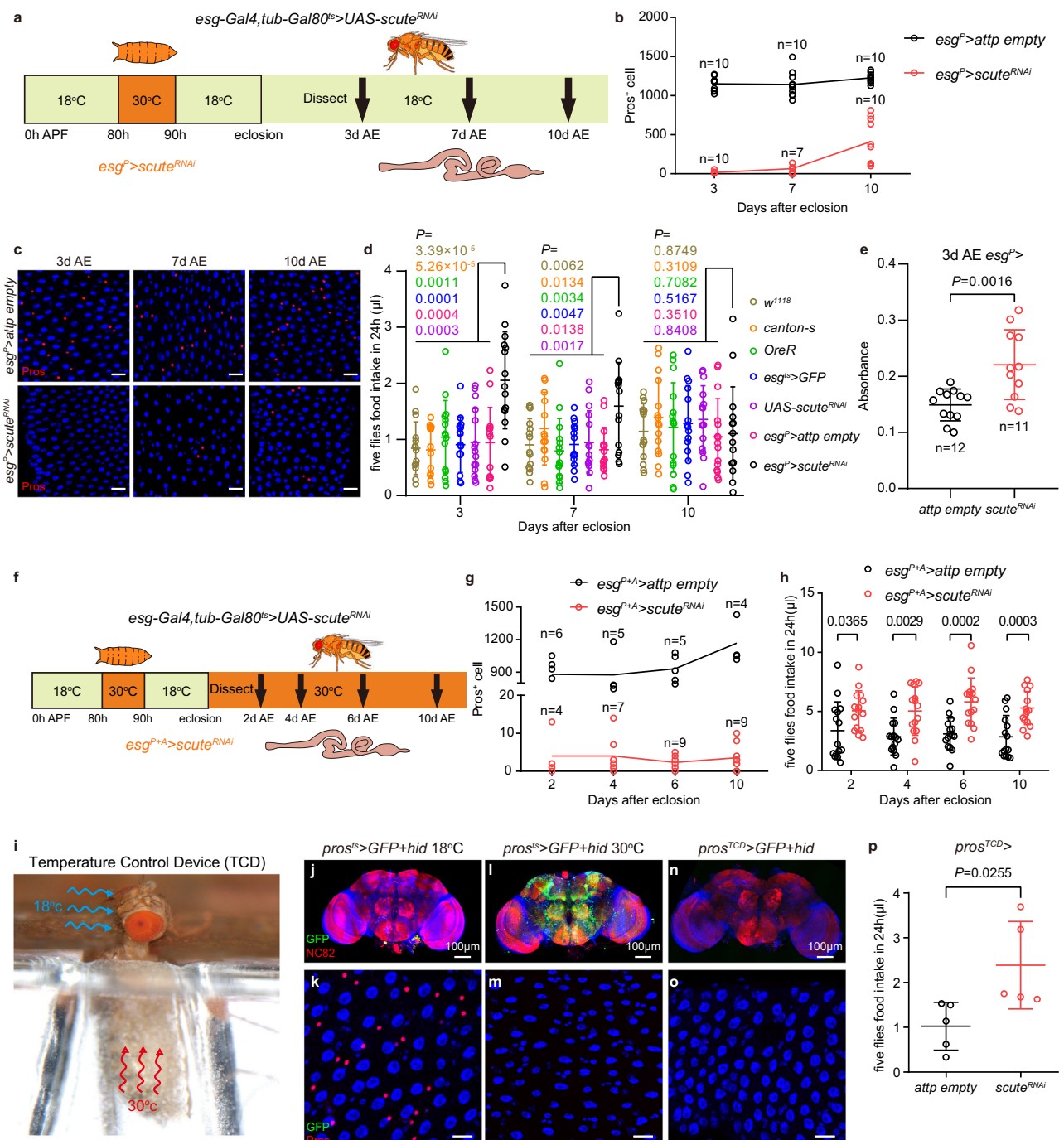

**Fig. 1 | Loss of EECs increases the food intake. a** Schematic representation of genetic manipulation to specifically eliminate EECs before eclosion (*esg^P^>scute^RNAi^*). APF, after pupal formation. AE, after eclosion. **b**, **c** Quantification **b** and representative images **c** of Pros+ (red) EECs in *control* and *esg^P^>scute^RNAi^* midguts at 3, 7 and 10 d AE. Here and in all images, cell nuclei are stained for 4′, 6-diamidino-2-phenylindole (DAPI; blue). **d** Food intake of six different *control* lines and *esg^P^>scute^RNAi^* flies at 3, 7 and 10 d AE. *n* = 15 in each genotype. **e** Standard cornmeal diet (SCD) consumption of *control* lines and *esg^P^>scute^RNAi^* flies at 3 d AE measured using the dye-based food intake measurement. **f** Schematic representation of genetic manipulation to eliminate EECs during pupal and adult (*esg^P+A^>scute^RNAi^*). **g** Quantification of Pros+ EECs in *control* and *esg^P+A^>scute^RNAi^* midguts at 2, 4, 6 and 10 d AE. **h** Food intake of *control* and *esg^P+A^>scute^RNAi^* flies at 2, 4, 6 and 10 d AE. n = 15 in each genotype. *P* values were shown in the figure. **i** The working status of the temperature control (TCD) device. **j-o**, Immunostaining of brains (red NC82 staining **j**, **l**, **n**) and midguts **k**, **l**, **o** of *pros^ts^ > GFP+hid* flies at 18 °C **j**, **k**, 30 °C **l**, **m** and in the TCD **n**, **o**. Note that GFP and Hid were not expressed in the head at 18 °C **j** and EECs were present **k**. EECs in *pros^ts^ > GFP+hid* flies **m** or *pros^TCD^ > GFP+hid* flies **o** were eliminated at 30 °C, while *pros^ts^ >* **l** but not *pros^TCD^ >* **n** drove GFP expression in the brain at 30 °C. 15 flies each were examined. **p** Food intake of *control* and *pros^TCD^>hid* flies. *n* = 5 in each genotype. Data are represented as mean ± SD. Significance was determined using two-sided unpaired *t*-test **d**, **e**, **h**, **p**. *n*, number of guts **b**, **g**, number of groups (5 flies in each group) performed for quantification of food intake **d**, **h**, **p**, or number of groups (20 flies in each group) performed for quantification of food consumption **e**. Source data are provided as a Source Data file. Scale bars, 20 μm except where otherwise specified.

*esg^P>scute^RNAi* adults at 30 °C upon eclosion (designated as *esg^{P+A}*) (Fig. 1f), thereby limiting the number of EECs to no more than 10 (Fig. 1g). In this setting, we found a significant increase in food intake of *esg^{P+A}>scute^RNAi* flies at 2, 4, 6 and 10 days AE compared to the *control* (Fig. 1h), again suggesting that the absence of EECs led to an increase in feeding. Despite overeating, these EEC-less flies defecated more (Extended Data Fig. 1i, j) and cleared the gut luminal contents faster than *control* (Extended Data Fig. 1k, l), with overall metabolic indexes (body mass (Extended Data Fig. 1m), glucose level (Extended Data Fig. 1n), protein (Extended Data Fig. 1o), triacylglyceride (TAG) (Extended Data Fig. 1p) and Oil red O staining (neutral lipids) of midgut epithelium (Extended Data Fig. 1q) indistinguishable from that of control flies.

As a complementary approach, we sought to eliminate EECs through targeted expression of a pro-apoptotic factor Hid[74] using the pan-EEC driver *prospero^V1* (*pros*)-*Gal4*[75,76]. However, *pros-Gal4* drives expression also in the brain[77]. To solve this problem, based on our previous study[78], we developed a temperature control device (TCD) that enables well-controlled heating of the fly abdomen at a sub-millimeter scale (Fig. 1i and Extended Data Fig. 2a–h, see Methods). Combined with the temperature-sensitive EEC driver (*pros-Gal4, tub-Gal80^ts*), TCD allows turning on *hid* expression only in EECs but not in the brain. We termed this method as *pros^TCD*. Indeed, *pros^TCD>hid* killed all EECs without triggering Hid expression in the brain (Fig. 1j–o). CAFE assays further confirmed a significant increase in food intake in these EEC-less *pros^TCD>hid* flies (Fig. 1p). Thus, our data obtained with two methods to specifically monitor all the EECs indicated that EECs function to inhibit food intake.

## EEC-derived NPF inhibits food intake

Next, we asked how the loss of EECs would lead to an increase in food intake. Since the gut microbiota-derived metabolites regulate food intake[79–81], we first examined the composition of gut microbiota in intestines without EECs. Our results show that there was no significant difference in the composition of gut microbiota in the intestine of *esg^P>scute^RNAi* 3d AE flies compared to the *control* (Extended Data Fig. 2i), suggesting that the rise in food intake due to EEC loss was not caused by changes in gut microbiota. In addition, we examined food intake between *control* flies and EEC-less flies (*esg^P>sc^RNAi*) reared under conventional and germ-free conditions 3 d AE. Our results show that regardless of microbiome status, EEC-less flies always consumed more food than *control* flies (Extended Data Fig. 2j, k), demonstrating that the gut microbiota is not responsible for the increased food intake due to the loss of EECs.

We then speculated that neuropeptides secreted by EECs inhibit feeding. EECs display a high degree of cellular diversity in the neuropeptides they secrete[54]. However, *esg^P>sc^RNAi* and *pros^TCD* are not compatible with sub-dissection of EECs. Since *Tkg-Gal4* drives expression in both brain and EECs (Extended Data Fig. 3a), an EEC-specific driver was still required. Encouraged by the homology between Tap and mammalian Ngn3 and the report that Tap is not a proneural protein in *Drosophila*[46,52], we checked if *tap* enhancers drove expression in EECs. A 1.3 kb enhancer fragment of *tap* conferred gene expression in both EECs and the brain (Extended Data Fig. 3b). To our delight, the gut and brain expression could be separated when this 1.3 kb element was sub-dissected (Extended Data Fig. 3c). While a 399 bp fragment, referred as *tap^{1.3}-A*, drove Gal4 expression in the brain (Extended Data Fig. 3d), a 432 bp fragment, termed *tap^{1.3}-B*, directed Gal4 expression only in EECs (Fig. 2a). Specifically, *tap^{1.3}-B-Gal4* is expressed in each one of the paired EECs in midgut regions R2c, R3 (copper cell region) and R4a (Fig. 2a)[82].

Neuropeptides are sorted into DCVs and released from peptidergic neurons by Ca^{2+}-triggered exocytosis[83]. To test if neuropeptides secreted from *tap^{1.3}-B* EECs regulate food intake, we blocked the secretion of *tap^{1.3}-B* EECs by expression of the *tetanus toxin light chain*

(*TNT*)[84], a protease that cleaves nSyb, a SNARE that is required for DCV fusion with the plasma membrane[85]. *tap^{1.3}-B > TNT* flies ingested significantly higher amounts of food than control flies (Fig. 2b). Moreover, exciting *tap^{1.3}-B* EECs by expressing the *transient receptor potential cation channel A1* (*TrpA1*), a temperature-sensitive cation channel[86], significantly decreased food intake at 30 °C (Fig. 2c). By contrast, *tap^{1.3}-B>TrpA1* flies did not reduce food intake at 18 °C. These results suggest that neuropeptide(s) secreted by *tap^{1.3}-B* EECs regulates food intake.

TK, NPF and Allatostatin C (Ast-C) are expressed in gut regions defined by *tap^{1.3}-B-Gal4*[54,56]. TK and NPF are expressed in the same EECs in this region, but TK-NPF and AstC display a mutually exclusive pattern in one pair of EECs[50,54,87]. TK (Extended Data Fig. 3e) and NPF (Fig. 2d) expression in *tap^{1.3}-B* EECs was confirmed by immunostaining, suggesting Ast-C is not expressed in those cells. Moreover, both TK (Extended Data Fig. 3f) and NPF (Fig. 2e) positive EECs were recovered in EEC-less *esg^P>sc^RNAi* flies raised to 10d AE, a time point that the overeating phenotype was suppressed. These expression analyses place TK and NPF as candidate neuropeptides that inhibit food intake.

Knock-down experiments of each neuropeptide genes were then performed. First, driving *Tk^RNAi* with either *tap^{1.3}-B-Gal4* or *Tkg-Gal4* eliminated TK expression in *tap^{1.3}-B* (Extended Data Fig. 3g–i) or all EECs (Extended Data Fig. 3k). However, food intake was not changed in either case (Extended Data Fig. 3j, l). By contrast, eliminating NPF in EECs but not in the brain using *tap^{1.3}-B-Gal4* (Fig. 2f, g and Extended Data Fig. 4a–h) or *Tkg-Gal4* (Extended Data Fig. 4i) to drive *NPF^RNAi*, significantly increased food intake (Fig. 2h and Extended Data Fig. 4j). This indicates that NPF, but not TK, secreted by *tap^{1.3}-B* EECs inhibits feeding. In addition to the CAFE assay, we also utilized the Manual Feeding (MAFE) assay to depict details of the feeding behavior of individual flies[88]. We found that depletion of *NPF* in *tap^{1.3}-B* EECs resulted in an increase in feeding time and total amount of food intake (Fig. 2i, j and Supplementary movie 1). Finally, not only sucrose food but also SCD was consumed significantly more by *tap^{1.3}-B > NPF^RNAi* flies (Fig. 2k), suggesting that knockdown of NPF in EECs increases the appetite of the flies.

In *tap^{1.3}-B > NPF^RNAi* flies, metabolic indexes of body mass (Extended Data Fig. 4k) and protein content (Extended Data Fig. 4l) were not changed compared with control flies. However, the glucose content (Extended Data Fig. 4m), body TAG level (Extended Data Fig. 4n) and Oil red O staining of guts (Extended Data Fig. 4o, p) were all significantly decreased, consistent with a previously described energy wasting status in flies depleted of gut *NPF*, which regulates lipid metabolism through glucagon-like and insulin-like hormones[66,69]. In summary, our genetic analysis demonstrates that EEC-derived NPF inhibits food intake.

## EECs sustain systemic NPF to restrict feeding

In insects, EECs secrete regulatory peptides into the hemolymph, an open circulatory system which most internal organs directly bathing in[44]. As in a previous work[66], our attempts to quantify the levels of NPF in the circulation with western blot or ELISA failed, likely due to the small size of mature NPF peptides. To support the idea that reduced NPF levels in the hemolymph (systemic NPF) underlies the increased food appetite seen in EEC-specific *NPF* knockdown (*tap^{1.3}-B > NPF^RNAi*) or EEC loss (*esg^P>sc^RNAi*), synthesized NPF peptides were directly injected into the body cavity of flies (Fig. 2l). Re-supplying systemic NPF in this way suppressed the increase in food intake seen in flies devoid of gut NPF and EECs (Fig. 2m, n). To rule out any contribution from the NPF neurons in the brain to systemic NPF, thorax NPF injection was again performed using two null mutants of *NPF*[62,89] and was still sufficient to reduce the food intake of *NPF* mutant flies (Fig. 2o). These results, together with our genetic evidence that EEC-derived NPF is required to avoid overeating, point to a specific role of EECs in maintaining NPF levels in the circulation. Of note, we noticed a significant reduction in food intake in these two null mutants of *NPF* that

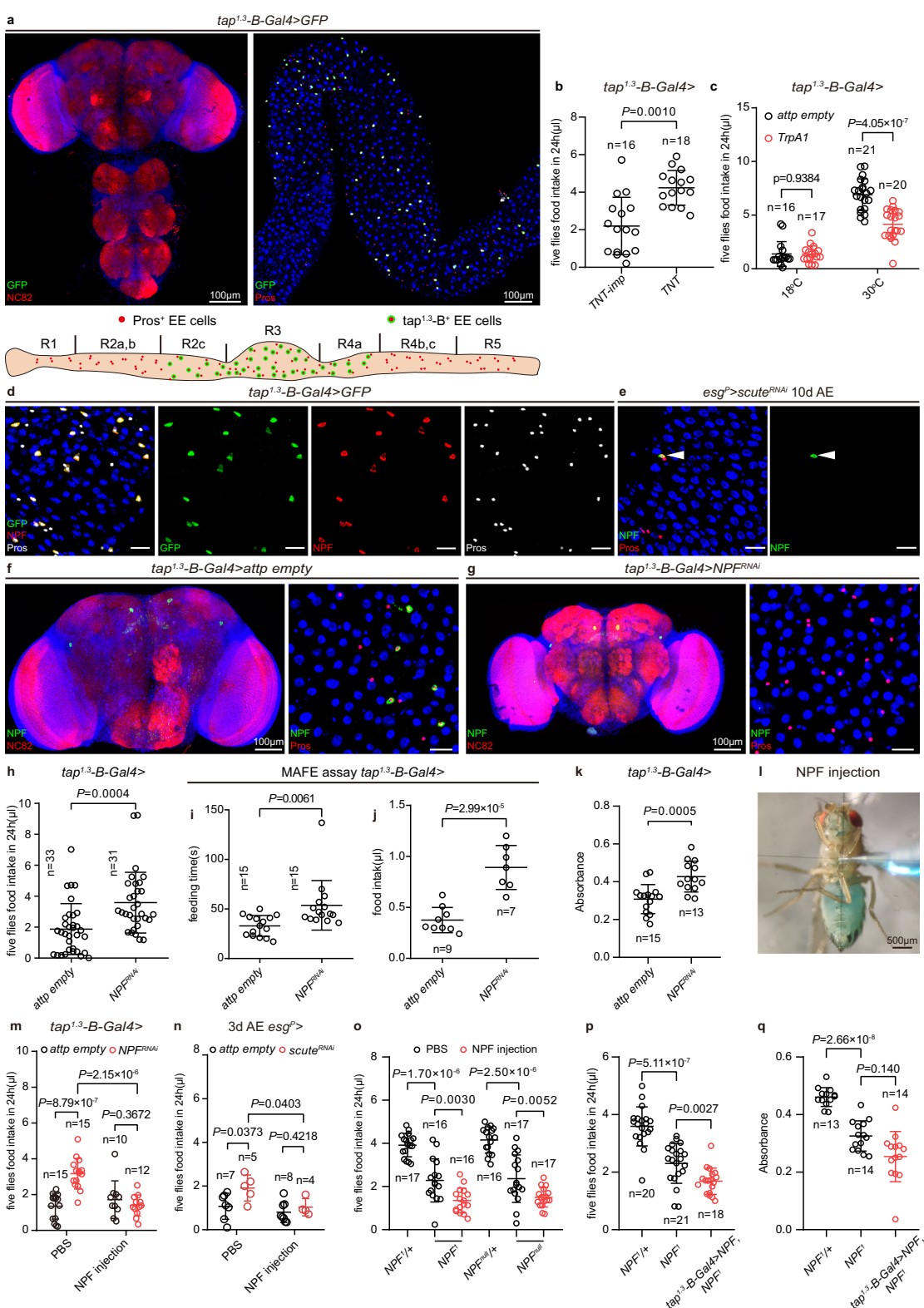

lack both brain *NPF* and gut *NPF* (Fig. 2o), consistent with a previous claim that brain-derived NPF promotes feeding[90,91]. It thus appears that the orexigenic effect of brain NPF overrides the role of gut NPF in restricting appetite. We then compared the food intake levels between NPF heterozygous mutant (*NPF[1/+]*), NPF homozygous mutant (*NPF[1]*) and gut-specific re-supply of NPF under the NPF mutant background (*tap[1.3]-B-Gal4 > NPF, NPF[1]*). Our results show that gut-derived NPF not only failed to rescue the reduced food intake caused by the NPF mutation, but also further suppressed food intake (Fig. 2p, q). These results support that NPF secreted by the brain and gut play opposing roles in appetite regulation, and that NPF secreted by the gut cannot replace the function of NPF secreted by the brain.

### L-Glu sensing reduces NPF secretion from EECs
Next, we wondered whether different nutrients would affect the secretion of NPF in EECs. Flies were allowed to ingest food containing

**Fig. 2 | EEC-derived NPF reduces food intake. a** Upper, expression pattern of *tap^1.3-B-Gal4 > GFP* in CNS and midgut. No GFP was observed in the CNS. Lower, schematic representation of the distribution of *tap^1.3-B-Gal4 > GFP* expressing EECs. 18 flies were examined. **b** Food intake of *control* and *tap^1.3-B-Gal4 > TNT* flies. An impotent TNT (*TNT-imp*) was used as a control. **c** Food intake of *control (attp empty)* and *tap^1.3-B-Gal4>TrpA1* flies at 18 °C or 30 °C. **d** *tap^1.3-B-Gal4 > GFP^+* cells were co-stained with NPF (red). 17 midguts were examined. **e** Emerging EECs were co-stained with NPF (green) in *esg^P>scute^RNAi* midguts after 10 days recovery. 17 midguts were examined. **f, g** NPF staining (green) in brains and midguts of *control* **f** and *tap^1.3-B-Gal4 > NPF^RNAi* **g** flies. 24 flies each were examined. **h** Food intake of *control* and *tap^1.3-B-Gal4 > NPF^RNAi* flies. **i, j** Feeding time **i** and food intake **j** of *control* and *tap^1.3-B-Gal4 > NPF^RNAi* flies measured using the MAFE assay. **k** Food consumption of *control* and *tap^1.3-B-Gal4 > NPF^RNAi* flies measured using the dye-based food intake measurement. **l** Ectopic NPF supplements were achieved by injecting 100 nM NPF into the thorax. **m** Food intake of *control* and *tap^1.3-B-Gal4 > NPF^RNAi* flies after PBS and NPF injection. **n** Food intake of 3 d AE *control* and *esg^P>scute^RNAi* flies after PBS and NPF injection. **o** Food intake of *heterozygous* (*NPF^1/+* and *NPF^null/+*) and homozygous *NPF* mutant (*NPF^1* and *NPF^null*) flies after PBS and NPF injection. **p, q** CAFE assay **p** and dye-based food intake measurement **q** of heterozygous (*NPF^1/+*), homozygous *NPF* mutant (*NPF^1*) and EECs-specific NPF recovery under *NPF* mutant condition (*tap^1.3-B-Gal4 > NPF, NPF^1*) flies. Data are represented as mean ± SD. Significance was determined using two-sided unpaired *t*-test **b, c, h, k, m–q**. *n*, number of groups performed for quantification of food intake (5 flies in each group) **b, c, h, m–p**, number of flies **i, j**, or number of groups (20 flies in each group) performed for quantification of food consumption **k, q**. Source data are provided as a Source Data file. Scale bars, 20 μm except where otherwise specified.

different major macronutrients and monitored for their gut NPF levels. We discovered that the intensity of NPF immunostaining in EECs was significantly increased when flies ingested high-protein food (5% yeast extract or yeast paste), but not high-sugar (10% sucrose) or high-fat (25% coconut oil)[92] diets (Fig. 3a, b). This implied a role for gut NPF in AA-sensing. To determine which AAs were responsible for the increased NPF immunostaining, flies were allowed to ingest food supplemented with each of the 20 AAs. From this screen, we determined that NPF staining was dramatically enhanced upon 1% L-glutamate (Glu) or 1% L-asparagine (Asn) supplementation (Fig. 3c, d and Extended Data Fig. 5a–c). Because of the important role of L-Glu in umami perception and metabolism[93,94], we focused here on the role of dietary L-Glu in regulating NPF secretion in EECs. Notably, L-Glu promoted NPF retention in EECs in a dose-dependent manner and its effect was prominent only at concentrations above 0.5% (Extended Data Fig. 5d, e). This is in line with the fact that commonly used cornmeal fly food with L-Glu content below 0.5% did not result in enhanced NPF retention (Fig. 3a). In addition, we examined the effect of 1% L-Glu on NPF expression in EECs of different regions of the midgut and in the brain. At the anterior end of midgut R2, if there was no NPF expression in the EEC before 1% L-Glu feeding, then high L-Glu failed to induce NPF staining in these regions (Extended Data Fig. 5f). In the brain, neither the high protein diets nor 1% L-Glu had an effect on NPF transcription or the intensity of NPF antibody staining (Extended Data Fig. 6a–c), suggesting that the high protein diet and 1% L-Glu only modulate NPF-expressed EEC in the midgut.

It is possible that the rise in NPF immunostaining in EECs was due to enhanced transcription of *NPF* and/or reduced peptide secretion. Since RT-qPCR revealed no transcriptional change in *NPF* mRNA from midgut of flies raised under multiple nutritional conditions (Fig. 3e), protein-rich food and L-Glu supplementation led to NPF retention in EECs was likely due to reduced secretion. To support this idea, we monitored neuropeptide secretion in DCVs by expressing a GFP-tagged rat atrial natriuretic factor (preproANF-EMD)[83] in *tap^1.3-B^+* EECs. After ruling out the possibility that *tap^1.3-B-Gal4* expression is regulated by high-protein diets or 1% L-Glu (Extended Data Fig. 6d), we revealed that pANF-EMD signals were significantly enhanced in EECs of flies ingesting high-protein and L-Glu diets, but not high-sugar or high-fat diet (Fig. 3f, g), indicating that protein/L-Glu-sensing by EECs reduced their secretory activity. To further confirm these observations, *tap^1.3-B>TrpA1* flies were reared on high-protein/L-Glu conditions and then underwent excitation. A concurrent and dramatic decrease in NPF immunostaining was observed when *tap^1.3-B>TrpA1* flies were shifted from 18 °C to 30 °C to open the TRP channels (Fig. 3h, i and Extended Data Fig. 6e–g), indicating that L-Glu triggered a rise in NPF staining by blocking the release of NPF from EECs. Although L-Glu greatly enhanced appetite, exciting *tap^1.3-B* EECs to release NPF into circulation was still able to decrease animal food intake on an L-Glu diet (Fig. 3j). Consistent with these findings, flies with knockdown of NPF (Fig. 3k) or EEC loss (Fig. 3l) consumed similar amounts of food to

controls only upon a high protein diet, suggesting that only a high protein diet inhibits the release of NPF into the circulation, whereas high sucrose and high fat diets do not. Thus, sensing of dietary L-Glu promotes feeding by inhibiting NPF secretion from EECs.

L-Glu can promote food intake of flies via Ir76b^+ neurons in the labellum and legs and DH44^+ neurons in the brain[28,29,42]. To integrate our findings of EEC perception of AAs with previously reported neuronal perception of L-Glu, we measured the effect of L-Glu in regulating food intake of animals with either normal or depleted gut NPF (Fig. 3m, n). While depletion of gut *NPF* led to increased food intake in a basic diet with only sucrose, supplying L-Glu in the diet to block NPF release blunted the effect of gut-specific loss of NPF although *tap^1.3-B > NPF^RNAi* flies trended to eat more but not to a level required for statistical significance (Fig. 3n). Moreover, on an L-Glu diet, NPF injection was still sufficient to significantly reduce food intake regardless of the presence or absence of gut NPF (Fig. 3n), further confirming a role of systemic NPF in restricting appetite. The observation that the anorexigenic effect of NPF injection only partially antagonized L-Glu-induced increase in food intake, supports the idea that NPF^+ EECs in the gut act as a secondary system that feeds back (to the brain) and adjusts feeding upon umami perception by neurons.

## L-Glu sensing slows down Ca²⁺ oscillation in EECs

Since the secretion of DCVs in neuroendocrine cells is regulated by $Ca^{2+}$ signaling[95,96], we hypothesized that high-protein/L-Glu diets inhibit NPF release by affecting $Ca^{2+}$ signaling in EECs. We expressed a genetically encoded $Ca^{2+}$ sensor GCaMP6f[97] under the control of *tap^1.3-B-Gal4* and performed $Ca^{2+}$ imaging in midguts dissected from flies reared on different diets (see Methods). Although the peak $Ca^{2+}$ activities did not differ between various feeding conditions (Extended Data Fig. 7a and Supplementary movie 2, 3), quantification of the frequency of $Ca^{2+}$ oscillations in individual *tap^1.3-B* EECs revealed that high-protein/L-Glu diets, but not high-sugar or high-fat diets, significantly decelerated $Ca^{2+}$ oscillations (Fig. 4a, b, Extended Data Fig. 7b–d and Supplementary movie 2, 3).

We then wondered if the speed of $Ca^{2+}$ oscillation in EECs underlies NPF secretion via DCVs. To this aim, we first sought to modify the frequency of $Ca^{2+}$ oscillations in EECs by knocking down known regulators of cytosolic $[Ca^{2+}]$ (Fig. 4c)[98–101]. Cytosolic $[Ca^{2+}]$ are dynamically controlled by influx and efflux processes[102]. Specifically, GPCR signaling activity produces 1, 4, 5-inositol trisphosphate (IP3) that binds to the IP3 receptor (IP3R), an ER $Ca^{2+}$ channel, allowing diffusion of $Ca^{2+}$ from the ER into the cytosol[101,103]. Decreased ER $[Ca^{2+}]$ is sensed by the stromal interaction molecule (Stim), an ER membrane protein that opens the plasma membrane $Ca^{2+}$ channel Orai, allowing influx of extracellular $Ca^{2+}$ into the cytosol[104–106]. Conversely, the sarco/endoplasmic reticulum $Ca^{2+}$-ATPase (SERCA) pumps cytosolic $Ca^{2+}$ into the ER while the plasma membrane $Ca^{2+}$ ATPase (PMCA) and Sodium calcium exchanger (NCX) channel extrudes $Ca^{2+}$ out of the cell[107–110].

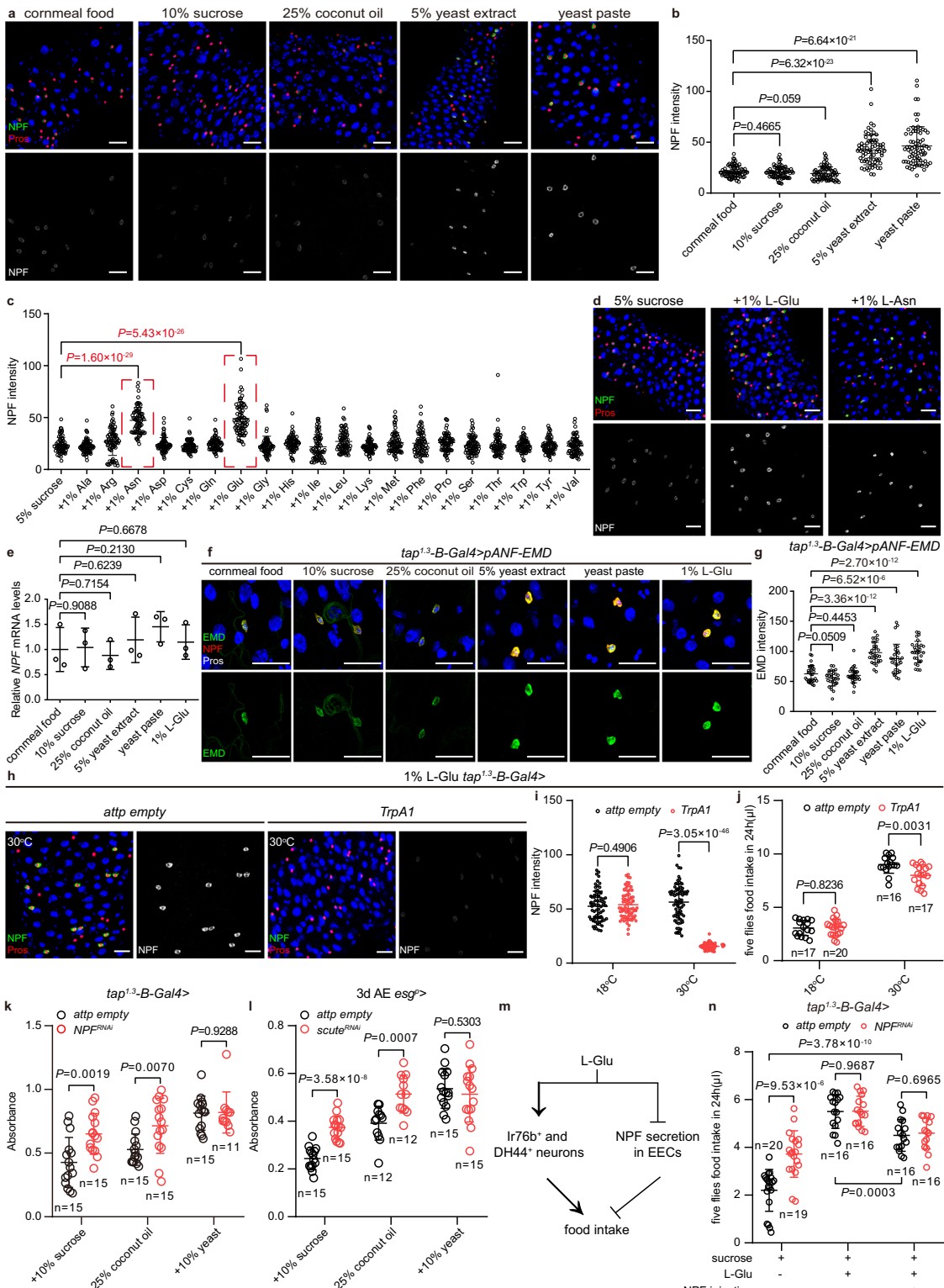

We found that knockdown of *stim* greatly accelerated $Ca^{2+}$ oscillations, while depletion of *SERCA*, *PMCA* and *IP3R* significantly decreased the oscillation frequency (Fig. 4d, e, Extended Data Fig. 7e–j and Supplementary movie 4-7). Using these tools to manipulate $Ca^{2+}$ oscillations specifically in $tap^{1.3}$-B EECs, we uncovered a strong correlation among the speed of $Ca^{2+}$ oscillations, the levels of DCV (Fig. 4f, g), NPF immunostaining (Fig. 4h, i) and animal food intake (Fig. 4j). During faster $Ca^{2+}$ oscillations (*stim*$^{RNAi}$), a reduction in the retention of both pANF-EMD and NPF in EECs (thereby increased NPF secretion) was

observed (Fig. 4f–i), and these flies ate significantly less (Fig. 4j). By contrast, slower $Ca^{2+}$ oscillations (*SERCA*$^{RNAi}$, *PMCA*$^{RNAi}$ and *IP3R*$^{RNAi}$), reminiscent of L-Glu feeding, elevated pANF-EMD and NPF retention in EECs (Fig. 4f–i), indicative of decreased NPF secretion. As a result, these flies consistently ingested more (Fig. 4j). Taken together, these data strongly support that the secretory capacity of EECs is instructed by cytosolic $Ca^{2+}$ oscillations rather than absolute $[Ca^{2+}]$. Thus, L-Glu sensing in EECs slows down $Ca^{2+}$ oscillations to reduce the secretion of NPF into the circulation, where NPF is anorexigenic.

**Fig. 3 | Dietary L-Glu inhibits NPF secretion from EECs. a, b** Representative images **a** and quantification **b** of NPF staining after ingestion of different foods. $n = 75$ in each group. **c** Quantification of NPF staining after feeding of single L-amino acids. Red dash line boxes indicate the two AAs, L-Asn and L-Glu, that significantly elevated NPF intensity. $n = 75$ in each group. **d** Representative images of NPF staining after feeding of 5% sucrose, 5% sucrose +1% L-Glu or 5% sucrose +1% L-Asn. **e** Normalized *NPF* mRNA levels after feeding different food by RT-qPCR. Each genotype corresponded to 3 biological replicates of 50 midguts each. **f, g** Representative images **f** and quantification **g** of pANF-EMD staining (green) after ingestion of different food. $n = 30$ in each group. **h, i** Under 1% L-Glu feeding condition, representative images **h** and quantification **i** of NPF staining in EECs of *control* and *tap[1-3]-B-Gal4>TrpA1* flies at 18 °C and 30 °C. $n = 75$ in each group. **j** Food intake of *control* and *tap[1-3]-B-Gal4>TrpA1* flies under 1% L-Glu feeding condition at

18 °C and 30 °C. **k, l** High-sugar (SCD + 10% sucrose), high-fat (SCD + 25% coconut oil) and high-protein (SCD + 10%yeast) food consumption of *control* and *tap[1-3]-B-Gal4 > NPF[RNAi]* flies **k** or *esg[P]>scute[RNAi]* flies at 3 d AE **l** measured using the dye-based food intake. **m** Schematic representation of the regulation of feeding by L-Glu that acts not only via neural perception, but also promotes appetite by inhibiting NPF release. **n** Food intake of *control* and *tap[1-3]-B-Gal4 > NPF[RNAi]* flies under different combinations of treatment (400 mM sucrose, 1% L-Glu, and NPF injection). Data are represented as mean ± SD. Significance was determined using two-sided unpaired *t*-test **b, c, e, g, i–l, n**. **n**, number of EECs **b, c, g, i**, number of groups performed for quantification of food intake (5 flies in each group) **j, n**, or number of groups (20 flies in each group) performed for quantification of food consumption **k, l**. Source data are provided as a Source Data file. Scale bars, 20 μm.

## EECs sense dietary L-Glu through mGluR

Sixteen glutamate receptors are encoded in the fly genome[89], including 2 metabotropic L-Glu receptors (mGluRs), 2 NMDA ionotropic receptors and 12 non-NMDA ionotropic receptors. We speculated that knocking down the L-Glu receptor(s) that mediates dietary L-Glu's inhibitory effects in the secretory capacity of *tap[1-3]-B* EECs would enhance NPF secretion into the hemolymph, and in turn suppress food intake. With this idea, we performed an RNAi screen for glutamate receptors that sustain flies' appetite to high L-Glu diet. We found that knockdown of *mGluR* (*CG11144*) but not other glutamate receptors in *tap[1-3]-B* EECs reduced the intake of L-Glu food (Fig. 5a), suggesting that mGluR in EECs senses dietary L-Glu to promote feeding. We subsequently found that knockdown of *mGluR* significantly accelerated $Ca^{2+}$ oscillations in *tap[1-3]-B* EECs of flies raised under high L-Glu diet and yeast diets (Fig. 5b, e and Extended Data Fig. 8a, d). Consistently, the faster $Ca^{2+}$ oscillations were accompanied with a decrease in the retention of both pANF-EMD (Fig. 5c, f and Extended Data Fig. 8b, e) and NPF (Fig. 5d, g and Extended Data Fig. 8c, f) in EECs. In sum, these results identified mGluR as the receptor that senses L-Glu by a subset of EECs.

## Two enteric neurons expressing NPFR (*NPFR[ENS]* neurons) inhibit food intake

We next sought to understand the mode of action that EEC-derived systemic NPF exerts its function in restricting appetite. A single *NPF receptor* (*NPFR*) is encoded in the fly genome[111]. In line with the strong orexigenic effect of brain NPF (Fig. 2o), *NPFR* mutant flies also ingested less food than that of heterozygous controls (Fig. 6a), suggesting that the food intake of *NPFR* mutant flies recapitulates NPFR function in the brain[111]. Since brain- and EEC-derived NPF have opposite effects on feeding, it is less likely that the systemic NPF maintained by EECs acts through NPFR in the brain. Therefore, we speculated that NPFR-expressing cells outside the CNS perceive the systemic NPF secreted from EECs.

Using an anti-NPFR antibody, we were able to detect NPFR expression in *tap[1-3]-B-Gal4[+]* EECs (Fig. 6b) and NPFR staining colocalizes with NPF antibody staining (Extended Data Fig. 9a). With the help of a transgenic reporter controlled by an NPF enhancer (*NPF-0.7-GFP*, Extended Data Fig. 9b, c), we confirmed that the same EECs express both NPF and NPFR (Extended Data Fig. 9d). However, knockdown of *NPFR* using *tap[1-3]-B-Gal4* did not change food intake (Fig. 6b, c). To better follow endogenous NPFR expression, we generated a *NPFR[3xHA]* knock-in line, in which a 3xHA tag was inserted immediately before the stop codon of *NPFR* using homologous recombination assisted by CRISPR/Cas9 (Extended Data Fig. 9e). However, HA staining was too weak to be detected in tissues except EECs (Extended Data Fig. 9f). To find additional tissues expressing NPFR, we further examined two *T2A-Gal4* knock-in lines that report *NPFR* isoform-specific expression patterns, *NPFR[RA/C]-Gal4* and *NPFR[RB/D]-Gal4*[89]. While both lines drove similar expression pattern in the brain, ventral nerve cord (VNC), visceral muscles and neuronal projections to the hindgut and rectal ampulla

regions, *NPFR-RA/C-Gal4* was additionally expressed in EECs, corpora cardiaca (CC)[66,69] and enteric neurons in the hypocerebral ganglion (HCG) (Fig. 6d and Extended Data Fig. 10a)[63].

Guided by the expression pattern, we investigated if NPFR is required in the visceral muscles or enteric neurons for feeding. Knocking down *NPFR* by muscle drivers *vm-Gal4*[112] or *How-Gal4*[113] did not alter food intake (Extended Data Fig. 10b), excluding a role for *NPFR* from gut muscles. To obtain a driver in NPFR[+] enteric neurons, we screened a collection of putative *NPFR* enhancer-Gal4 lines[114]. Among them, *GMR60E02-Gal4* containing 667 bp of the fourth intron of *NPFR* drove expression in HCG neurons (Fig. 6e and Extended Data Fig. 10c, d). Detailed inspection revealed a pair of enteric neurons with cell bodies located immediately anteriorly to the proventriculus of the adult gut (inset in Fig. 6e and Extended Data Fig. 10e). Their neurites ascend to the subesophageal zone (SEZ), a well-known brain center for feeding control[115], and descend along the midgut wall to the end of the R1 region (Fig. 6e and Extended Data Fig. 10f)[82]. Stochastic labeling by MultiColor-FlpOut technique[116] reveals that these two neurons have similar but diverse projections to the SEZ (Extended Data Fig. 10g). With an intersectional strategy (*NPFR[A/C]-LexA ∩ GMR60E02-Gal4*)[117], we determined that *GMR60E02-Gal4* neurons are truly NPFR expressing cells (Extended Data Fig. 10h). These two neurons are not the previously described NPFR-expressing cells in the CC[66], as they stained negative for AKH, a CC marker (Fig. 6e and Extended Data Fig. 10i). This driver was termed as *NPFR[ENS]-Gal4* to refer its highly specific expression in the enteric nervous system. Strikingly, depleting *NPFR* using *NPFR[ENS]-Gal4* greatly increased animal food intake (Fig. 6f), implicating the two *NPFR[ENS]* neurons in relaying the appetite control signal emanating from gut-derived NPF.

We then carried out functional characterization of the *NPFR[ENS]* neurons in more detail. First, targeted ablation of *NPFR[ENS]* neurons by expressing the proapoptotic factor Hid[74,118] relieved restriction on fly appetite (Fig. 6f). Second, inhibiting *NPFR[ENS]* neuronal activity by expressing a temperature-sensitive, dominant-negative form of Dynamin, *shibire[ts]* (*shi[ts]*), elevated food intake when the releasable pool of synaptic vesicles was disrupted by raising flies at 30 °C (Fig. 6g). Third, activating *NPFR[ENS]* neurons by expressing *TrpA1* led to feeding inhibition at 30 °C (Fig. 6g), a condition that the *TrpA1* cation channel is opened to depolarize neurons. Thus, *NPFR[ENS]* neurons function to suppress feeding.

We further tested whether *NPFR[ENS]* neurons mediate the physiological changes imposed by dietary L-Glu. A calcium-sensitive reporter CaLexA[119] that drives GFP expression proportionally to cumulative neuronal activity, was applied to check if *NPFR[ENS]* neurons respond to L-Glu supplementation by changing their activity. We determined that L-Glu or high-protein diets that were found to reduce gut secretion of NPF into the circulation, inhibited the activity of *NPFR[ENS]* neurons compared to cornmeal food, 10% sucrose and 25% coconut oil food (Fig. 6h, i and Extended Data Fig. 10j, k). Conversely, directly supplying systemic NPF by injecting NPF peptides into the hemolymph significantly excited the *NPFR[ENS]* neurons and completely blunted the

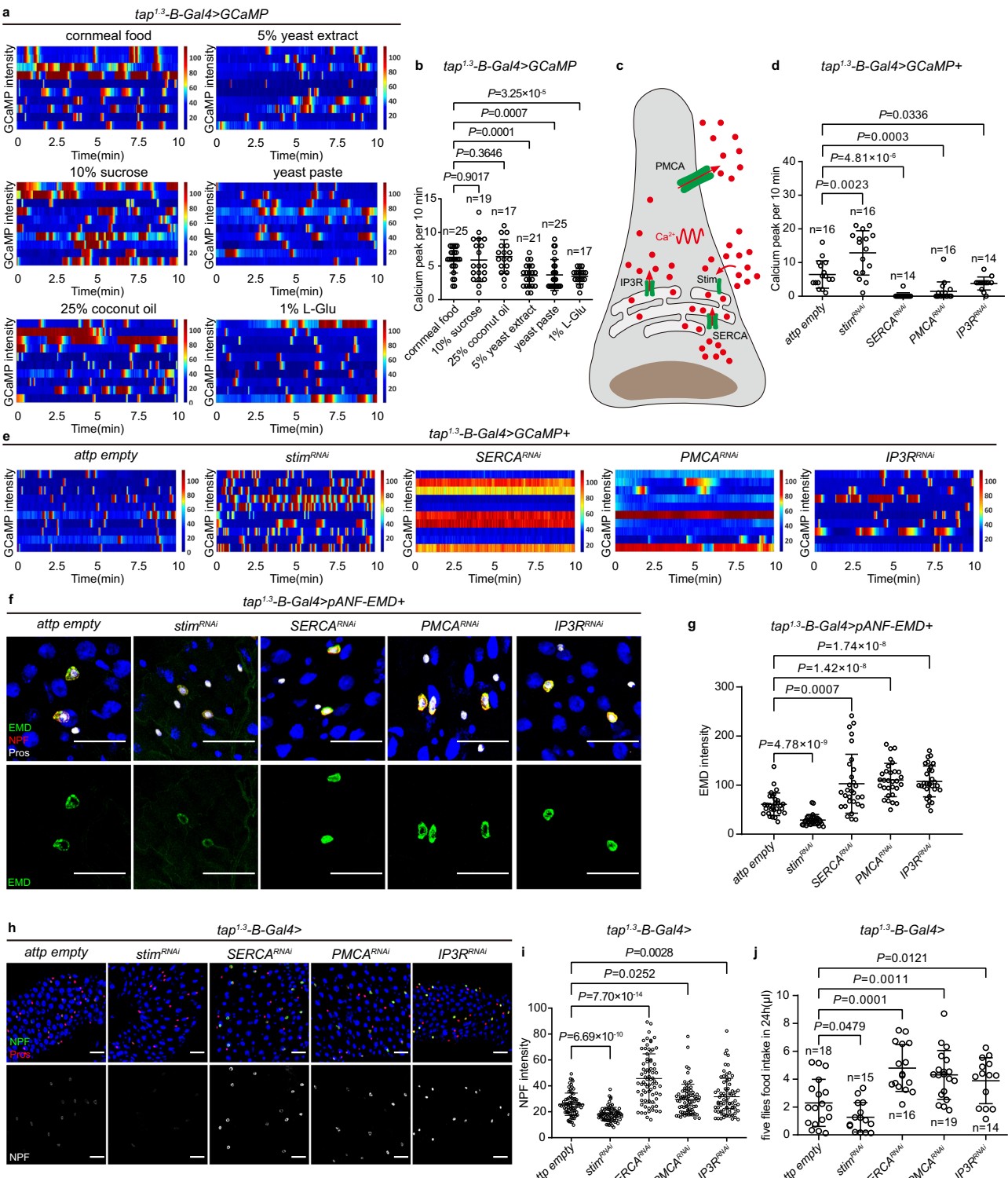

**Fig. 4 | Calcium oscillations of EECs regulate NPF secretion. a**, **b** Representative heatmap records of GCaMP intensity of 10 individual EECs **a** and quantification of calcium peaks in EECs **b** within 10 min (660 frames) under different feeding conditions. **c** Schematic representation of the regulation of Ca²⁺ flux. IP3R causes release of Ca²⁺ (red dots) from the ER to the cytoplasm. Stim senses the decline of Ca²⁺ in the ER, and induces extracellular Ca²⁺ influx into the cytoplasm, forming a high [Ca²⁺]. Excessive cytoplasmic Ca²⁺ is pumped into the ER by SERCA or out of the cell by PMCA, resulting in a decrease in cytoplasmic [Ca²⁺]. **d**, **e** Quantification of calcium peaks in EECs **d** and representative heatmap records of GCaMP intensity of

10 individual EECs **e** of flies with the indicated genotypes within 10 min. **f**, **g** Representative images **f** and quantification **g** of pANF-EMD staining in EECs of flies with the indicated genotypes. *n* = 30 in each genotype. **h**, **i** Representative images **h** and quantification **i** of NPF staining in EECs of flies with the indicated genotypes. *n* = 75 in each genotype. **j** Food intake of flies of *control* and the indicated genotypes. Data are represented as mean ± SD. Significance was determined using two-sided unpaired *t*-test **b**, **d**, **g**, **I**, **j**. *n*, number of EECs **a**, **b**, **d**, **g**, **i**, or number of groups performed for quantification of food intake (5 flies in each group) **j**. Source data are provided as a Source Data file. Scale bars, 20 μm.

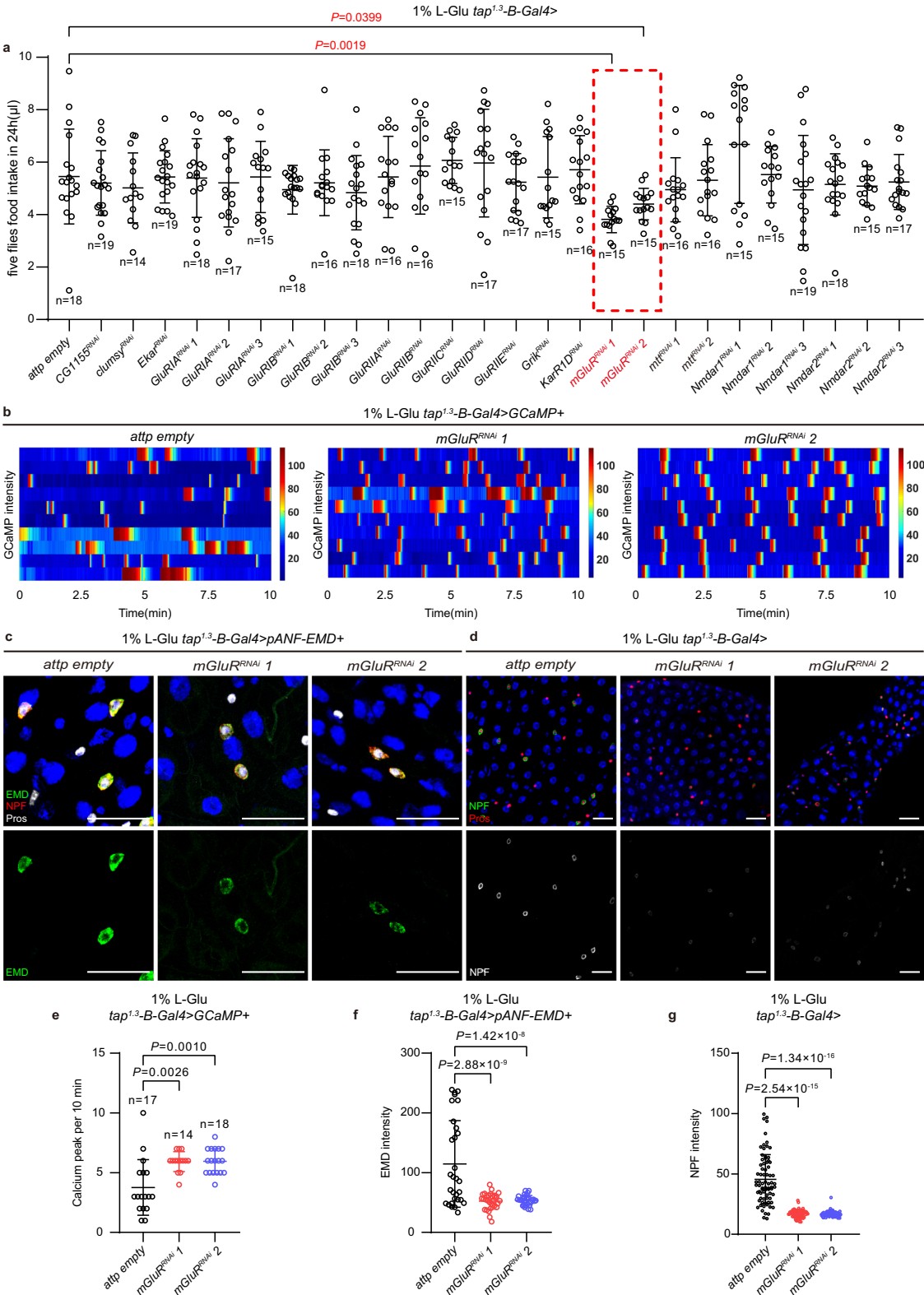

**Fig. 5 | mGluR regulates the secretion of NPF from EECs. a** Food intake of flies depleted for different glutamate receptors in EECs under 1% L-Glu feeding condition. Red-dash line box indicates the two mGluR$^{RNAi}$ lines that significantly decreased the food intake. **b** Under 1% L-Glu feeding condition, representative heatmap records of GCaMP intensity of 10 individual EECs in *control* and *tap$^{1.3}$-B-Gal4>mGluR$^{RNAi}$* flies within 10 min. **c** Representative images of pANF-EMD staining in EECs of *control* and *tap$^{1.3}$-B-Gal4>mGluR$^{RNAi}$* flies under 1% L-Glu feeding condition. **d** Representative images of NPF staining in EECs of *control* and *tap$^{1.3}$-B-*

*Gal4>mGluR$^{RNAi}$* flies under 1% L-Glu feeding condition. **e**–**g** Quantification of calcium peaks **e**, pANF-EMD **f** and NPF **g** staining in EECs of *control* and *tap$^{1.3}$-B-Gal4>mGluR$^{RNAi}$* flies under 1% L-Glu feeding condition. $n = 30$ **f** and =75 **g**. Data are represented as mean ± SD. Significance was determined using two-sided unpaired $t$-test (**a**, **e**–**g**). $n$, number of groups performed for quantification of food intake (5 flies in each group) **a**, or number of EECs **e**–**g**. Source data are provided as a Source Data file. Scale bars, 20 μm.

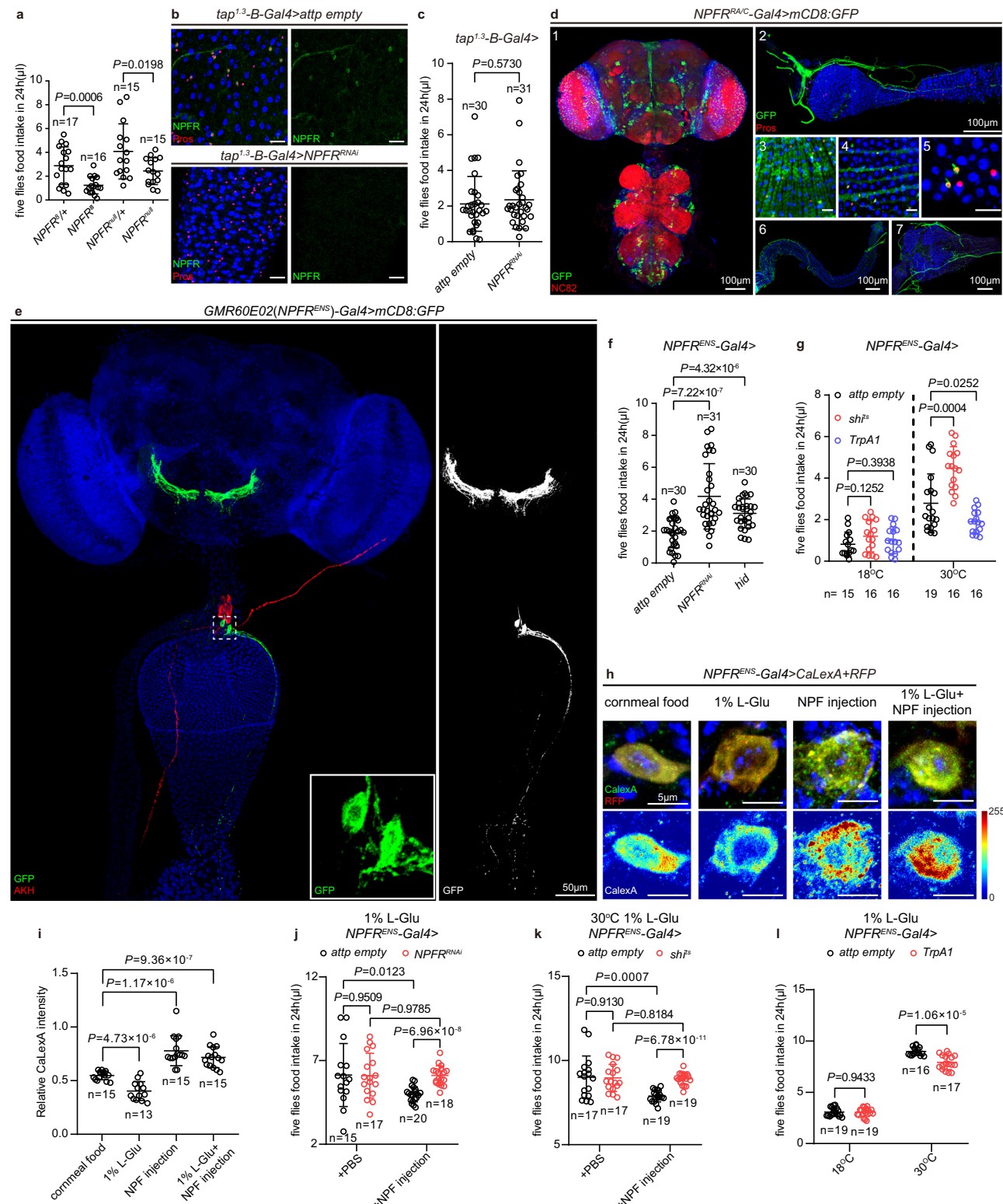

suppressive effects imposed by L-Glu or high-protein diets (Fig. 6h, i and Extended Data Fig. 10j, k). Food intake was further measured to confirm that *NPFR^ENS* neurons mediate the anorexigenic effects of systemic NPF released from EECs. As previously described, 1% L-Glu feeding resulted in reduced NPF secretion from the EECs, and in this condition, knocking down *NPFR* in the *NPFR^ENS* neurons, or inhibiting *NPFR^ENS* neuronal activity by *shi^ts*, did not alter the levels of food intake (Fig. 6j, k). This indicates that when systemic NPF levels turn low, *NPFR^ENS* neurons become no longer essential for the feeding control. By

contrast, while NPF injection was sufficient to reduce the food intake of wild type control (*NPFR^ENS>attp*) flies raised on L-Glu diet, it no longer caused a drop in food intake of flies with depleted *NPFR* in *NPFR^ENS* neurons (Fig. 6j) or in flies whose *NPFR^ENS* neurons were silenced by *shi^ts* (Fig. 6k). These data are consistent with a model that the two *NPFR^ENS* neurons are required to perceive systemic NPF levels and control feeding.

Further supporting our model, activation of *NPFR^ENS* neurons by expressing *TrpA1*, reduced feeding of flies raised both on normal diets

**Fig. 6 | A pair of NPFR-expressing enteric neurons senses NPF secreted from EECs and inhibits feeding. a** Food intake of *NPFR* heterozygous control (*NPFR*[8]/+ and *NPFR*[null]/+) and *NPFR mutant* (*NPFR*[8] and *NPFR*[null]) flies. **b** NPFR staining (green) in EECs of *control* and *tap*[1-3]-*B-Gal4 > NPFR*[RNAi] flies. 30 midguts each were examined. **c** Food intake of *control* and *tap*[1-3]-*B-Gal4 > NPFR*[RNAi] flies. **d** The GFP expression pattern driven by *NPFR*[RA/C]-*Gal4* in CNS (1), enteric neurons in the hypocerbral ganglion (HCG) (2), midgut circular muscle (3) and longitudinal muscle (4), EECs (5) and neuronal projection to the hindgut (6) and rectal ampulla (7). 25 flies were examined. **e** GFP expression pattern driven by *GMR60E02* (*NPFR*[ENS])-*Gal4*. White dashed box frames the cell body of a pair of enteric neurons, with magnified view shown in the lower right corner. AKH staining (red) indicates the location of the corpora cardiaca. The enhanced GFP channel (white) is shown on the right. 31 flies were examined. **f** Food intake of flies with *NPFR* knockdown in *NPFR*[ENS] neurons or elimination of this pair of neurons (*> hid*). **g** Food intake of flies with the indicated genotypes. Note that inhibition of *NPFR*[ENS] neurons (*> shi*[ts], 30 °C) promoted feeding, whereas exciting *NPFR*[ENS] neurons (*> TrpA1*, 30 °C) inhibited food intake. **h, i** Upon indicated manipulations, representative images **h** and quantification **i** of relative CaLexA intensity in *NPFR*[ENS] neurons. **j, k** Food intake of flies with the indicated genotypes under 1% L-Glu feeding condition. *NPFR* knockdown in *NPFR*[ENS] neurons **j** or inhibition of *NPFR*[ENS] neuron function (**k,** *>shi*[ts], 30 °C) had a similar food intake as control in PBS injection group, whereas *NPFR* knockdown **j** or inhibition of *NPFR*[ENS] neurons **k** had a higher food consumption than *control* in NPF injection group. **l** Under 1% L-Glu feeding condition, activation of *NPFR*[ENS] neurons (*> TrpA1*, 30 °C) inhibited food intake. Data are represented as mean ± SD. Significance was determined using two-sided unpaired *t*-test **a, c, f, g, i–l**. *n*, number of groups performed for quantification of food intake (5 flies in each group) **a, c, f, g, j–l**, or the number of *NPFR*[ENS]-*Gal4*[+] cells **i**. Source data are provided as a Source Data file. Scale bars, 20 μm unless otherwise specified.

(Fig. 6g) and on L-Glu diet, a condition with low systemic NPF (Fig. 6l). This indicates that permanently exciting *NPFR*[ENS] neurons decouples feeding from the control by systemic NPF and is sufficient to convey a dieting signal.

### Dopamine is required for *NPFR*[ENS] neuron function

Encouraged by the crucial role of the two *NPFR*[ENS] neurons in relaying the gut "feeling" of food quality into the brain, we went on to characterize the cellular and molecular nature of *NPFR*[ENS] neurons. Combining the GFP-tagged presynaptic marker (*nSyt::GFP*)[120] and the RFP-tagged dendritic marker (*DenMark*)[121], we revealed that the neurites of *NPFR*[ENS] neurons in the SEZ are axonal while the neurites innervating the midgut are dendrites (Fig. 7a).

To further investigate the molecular mechanism whereby *NPFR*[ENS] neurons inhibit feeding, we carried out an RNAi screen for genes coding for synthetases or transporters of neurotransmitters, by specifically knocking them down in *NPFR*[ENS] neurons followed by food intake analyses (Fig. 7b). Inhibiting dopaminergic signaling by RNAi against Dopa decarboxylase (Ddc) or Vesicular monoamine transporter (Vmat) dramatically increased food intake (Fig. 7b). Consistent with the functional assay, the dopaminergic nature of the *NPFR*[ENS] neurons was supported by their co-labeling with the dopaminergic marker *Ddc-LexA > GFP* both in the cell body and the neurites (Fig. 7c). Furthermore, immunostaining against Tyrosine hydroxylase (TH), an enzyme required for dopamine synthesis, confirmed the dopaminergic identity of *NPFR*[ENS] neurons (Fig. 7d). Taken together, our data indicate that *NPFR*[ENS] neurons use dopamine to signal feeding inhibition.

Finally, anterograde trans-synaptic labeling was performed to map the postsynaptic partners of *NPFR*[ENS] neurons using a genetically encoded reporter *trans-Tango*[122]. This method identified neurons in the SEZ and antennal lobe (AL) that synapse with *NPFR*[ENS] neurons (Fig. 7e). The dendritic pattern and cell body locations of those SEZ neurons revealed by *trans-Tango* reminded us of motor neurons[115] and interneurons[123,124] that control feeding. Such synaptic organization of *NPFR*[ENS] neurons, reminiscent of the mammalian vagal afferent neurons[125], is consistent with their role in facilitating communication between the periphery and the brain, by dynamically surveying the intestine and talking to the SEZ, the central pattern generator for feeding behaviors[115].

### Discussion

Our study has identified EECs as critical intestinal sensors of AAs. EECs along with the established gustatory and systemic AA sensors constitute a complete AA-sensing network dynamically evaluating food quality at each step of food ingestion and further informing the brain to adjust appetite. Through developing three approaches, we managed to perform clean manipulations of EECs. Remarkably, we uncovered that the modulation of specific features of intracellular Ca[2+] signaling in EECs following L-Glu sensing adjusts animal feeding behavior via a gut-brain axis sustained by the NPF/NPFR system (Fig. 7f). Of note, our study highlights the secretory capacity of EECs is regulated by the frequency rather than peak intensity of Ca[2+] oscillations and that gut-derived neuropeptides do not necessarily enter the brain to impact animal behaviors.

Upon AA sensing, EECs also regulate food intake in rodent models[13–16]. Further adding to the parallel, the two AAs (L-Glu and L-Asn) identified in our study that limit the secretion of NPF are also the two main AAs that trigger secretion of EECs via Ca[2+] signaling in mammals. Thus, EECs in flies and in mammals share a high degree of functional similarities, suggesting the mechanisms that we have provided here with the unique power of *Drosophila* as a research paradigm should greatly advance understanding of the fundamental principles of EEC nutrient sensing process in human.

EECs are primary nutrient sensors, detecting luminal content and trans-epithelial flux of nutrients ranging from sugar, fat to protein and AAs[126]. The nutrient sensing process is usually initiated via recognition of specific nutrient molecule by receptors or transporters located in the plasma membrane[8,10,127–131]. However, the molecular engine driving the EEC secretory machinery following nutrient sensing had not been previously studied. As is the case with the excitation of neurons, fly work reveals that EECs respond to dietary proteins by changing cytosolic Ca[2+] activity. CaLexA and GCaMP Ca[2+] indicators revealed that a subset of EECs co-expressing DH31, CCHa1 and TK in the posterior midgut were activated by proteins and AAs[45]. These EECs responded to both essential and nonessential amino acids, but not to either single AAs, sugar or fat[44,132]. Thus, it appears that EECs of the II-p population[54] dynamically evaluate the overall dietary protein levels but not specific AAs and in turn enhance secretory activity through elevated intensity of Ca[2+] signaling.

This is in sharp contrast to NPF[+] EECs that sense specific AAs as demonstrated here. NPF[+] EECs were recently reported to sense dietary sugar and modulate fly feeding and metabolism[66,69], although different SLC2-family sugar transporters (sut1 vs sut2) were deemed important in mediating sugar sensing in these studies. The discrepancy with our conclusion may have arisen from different feeding protocols. In our experiment, flies were only fasted for 3 h, or treated without fasting period (dye-based food intake measurement), after which we measured the food intake of flies over a 24-h period, whereas the two studies mentioned above looked at NPF function under acute starvation and sugar-refeeding conditions. Furthermore, Rewitz and colleagues found that NPF release upon sugar sensing or NPF injection limited sugar intake but promoted protein consumption indirectly through the glucagon-like factor AKH that mobilizes stored energy in adipose tissues[69]. In light of our findings that the two identified *NPFR*[ENS] enteric neurons perceive NPF in circulation and directly synapse with SEZ neurons in the brain to terminate feeding, it is less likely that the NPF[+] EEC-NPFR[ENS] enteric neuron-SEZ circuit we identified in this work is responsible for nutrient-specific feeding decisions. Nevertheless, it is

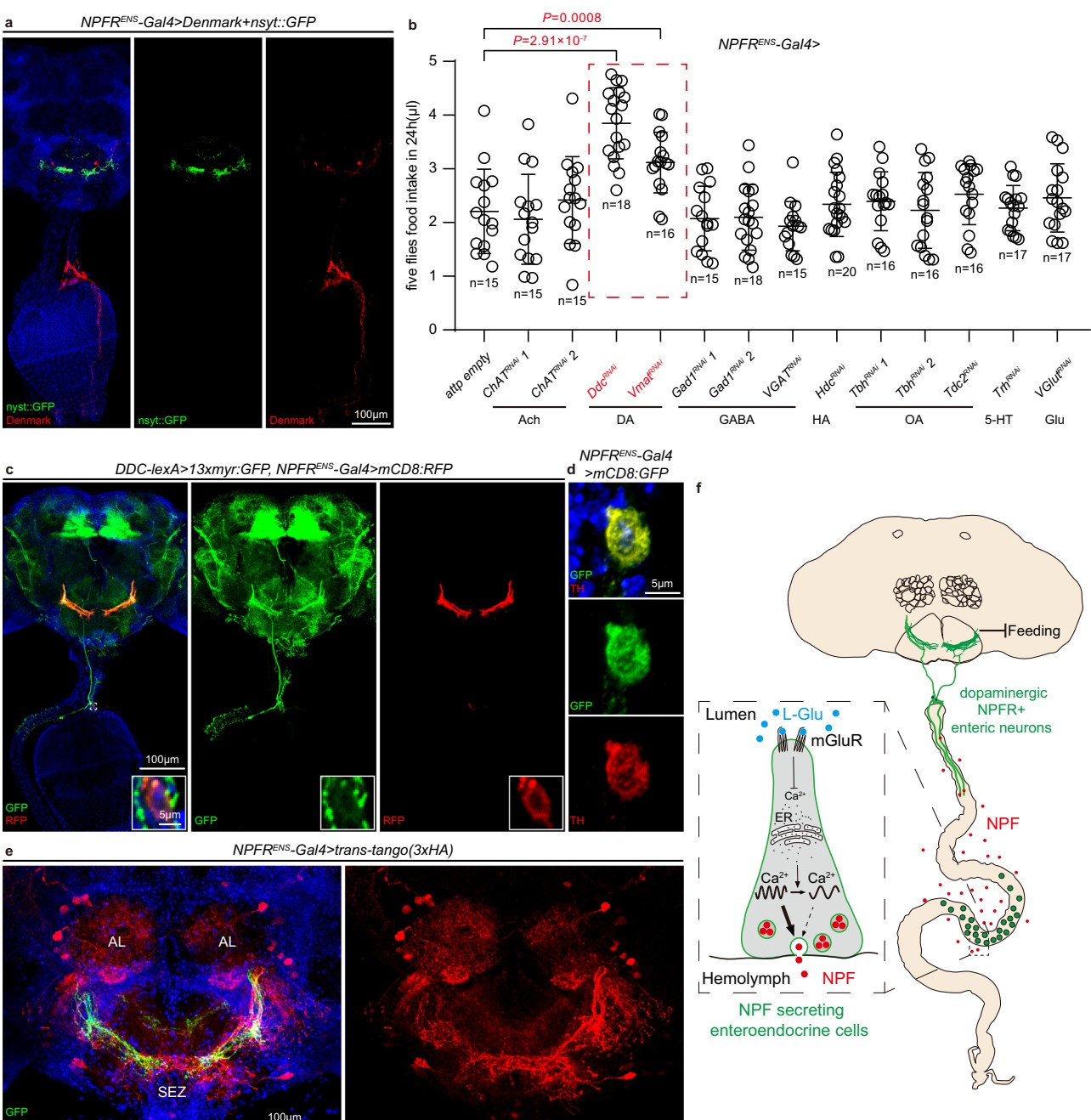

**Fig. 7 | Axons of dopaminergic *NPFR^ENS* neurons connect to the SEZ and AL regions in the brain. a** *NPFR^ENS* neurons are labeled by nSyt::GFP (green, axons) and Denmark (red, dendrites). 24 flies were examined. **b** Food intake of flies expressing *RNAi* against key factors for the synthesis and function of different neurotransmitters in *NPFR^ENS* neurons. Red-dash line box indicates expressing *RNAi* lines against two key enzymes for the synthesis of dopamine (DA) significantly increases the food intake. **c** *NPFR^ENS* neurons (*NPFR^ENS-Gal4 > mCD8:RFP*, red) are co-labeled with the dopaminergic neuron marker *Ddc-LexA>myr:GFP*. Note they have the same dendritic pattern in the SEZ region. The white dashed box frames the cell body of *NPFR^ENS* neurons, with magnified views in the lower right corner. 23 flies were examined. **d** *NPFR^ENS* neurons (*NPFR^ENS-Gal4 > mCD8:GFP*, green) stained positive for

Tyrosine hydroxylase (TH, red). 15 flies were examined. **e** Trans-tango experiment shows that *NPFR^ENS* neurons are synaptically connected with neurons (red, HA staining) in the subesophageal zone (SEZ) and antennal lobe (AL). 25 flies were examined. **f** Proposed model of EEC sensing of L-Glu and its downstream circuit. L-Glu sensing by EECs inhibits NPF secretion from EECs by slowing down Ca²⁺ oscillations, thereby blocking the activation of dopaminergic NPFR⁺ enteric neurons that inhibit feeding. Data are represented as mean ± SD. Significance was determined using two-sided unpaired *t*-test. *n*, number of groups performed for quantification of food intake, 5 flies in each group **b**. Source data are provided as a Source Data file. Scale bars are indicated in panels.

highly possible that NPF⁺ EECs can sense both AAs and sugar and adjust feeding behavior tightly depending on the exact feeding context and the downstream circuits.

By combining live Ca²⁺ imaging and genetic perturbations that alter Ca²⁺ oscillations, we noticed that L-Glu supplementation induced an mGluR-dependent deceleration of Ca²⁺ oscillations in EECs, causing

retention of DCVs and their neuropeptide cargos. Our study reveals a crucial role of the frequency of Ca²⁺ oscillations in driving EEC secretion. By contrast, peak intensity of Ca²⁺ oscillations did not correlate with the secretory capacity of EECs. This finding is remarkable, as previous studies often simply highlight the intensity of Ca²⁺ oscillations as critical for cellular activities of neurons and EECs, without detailing

the oscillation frequency. We reason that compared to neurons that use fast-acting small-molecule transmitters at synapses, EECs act via slow-acting neuromodulator peptides mostly through circulation and therefore need to keep releasing peptides to generate systemic concentrations above a critical threshold required to signal to the receptor in remote tissues.

Dietary L-Glu also activates *Drosophila* intestinal stem cells (ISCs) in an mGluR-dependent manner. Similarly, L-Glu slows down Ca$^{2+}$ oscillations in ISCs as well and induces ISC proliferation by creating high cytosolic Ca$^{2+}$ concentrations that drive stem cell dividing[101]. Thus, EECs and ISCs favor Ca$^{2+}$ oscillation frequency and intensity respectively for their activity (secretion vs proliferation). In this way, different epithelial cell types generate a concerted response to L-Glu ingestion by simultaneously reducing release of NPF from EECs to increase food intake and activating stem cell activity to support intestinal growth and regeneration. It is plausible that distinct features of Ca$^{2+}$ signaling have been opted for various cellular activities, necessitating examining oscillatory features of Ca$^{2+}$ activity in future work.

NPY family of peptides including NPY itself, peptide YY (PYY) and pancreatic polypeptide (PP), are well known central regulators of feeding behavior in mammals. *Drosophila* encodes a single homolog of the NPY family peptide, NPF[133]. As a gut-brain peptide, our study reveals opposite roles for brain NPF and gut NPF in regulating feeding. We first confirmed previous claims that brain NPF promotes feeding[90,91] and further mechanistically dissected the role and mode of action of gut-derived NPF. Similar to brain NPF, NPY is mainly expressed in the brain and promotes feeding[134,135]. Moreover, reminiscent of gut NPF in flies, PYY secretion is postprandially activated in enteroendocrine L-cells to restrict feeding[13,57]. Together, NPY/NPF are deeply conserved in feeding control depending on the location where the peptide is released.

The compartmentalized function of brain- and gut-derived NPF on feeding raises an interesting notion that some peptide hormones do not cross the blood-brain barrier (BBB), a specialized endothelial structure governing entry and exit of all small molecules to and from the brain interstitial space[136], and therefore can act on target tissues in different ways. Our data do not support the notion that EEC-derived NPF interferes with the action of brain NPF, and vice versa. Our study provides an example of the functional compartmentalization of hormones between the brain and the periphery in *Drosophila*. The ability of BBB penetration may differ between neuropeptides as a few studies have reported that gut peptides are able to excite brain neurons despite no direct evidence supporting their BBB crossing[44,66,132,137]. While visualizing neuropeptide release and diffusion through circulation remains technically challenging[138,139], future work should define the permeability and transportation features of the BBB.

While EECs release PYY upon ingestion of protein-rich food to limit appetite in mammals[17], our genetic analysis together with NPF injection experiments shows that gut-derived NPF sustains a systemic function of NPF in restricting feeding in flies. Thus, intestinal epithelium-derived NPF/PYY exhibit an evolutionarily conserved role in restricting food appetite from flies to mammals. Intriguingly, PYY/NPF secretion from EECs appears to have been differentially regulated to fulfill respective nutritional demands of flies and mammals. Ingestion of protein-rich food leads to a reduction in NPF secretion from *Drosophila* gut, but instead promotes PYY secretion in mice. This is consistent with a notion that while mammals need to tightly adjust the overall energy balance to avoid metabolic disorders associated with uncontrolled food intake[140], insects tend to maximize the acquisition of nutritious protein food for their reproduction and adaptation into the fast-changing nutritional environment. As a striking example, mosquitoes can typically consume an amount more than their own body weight in a single blood meal that is rich in proteins, and are then locked in a satiety state for 3-4 days, a process that requires the activity

of an NPY-like receptor although its in vivo ligand and tissue source remain unclear[141]. The disparate control of NPF/PYY secretion upon AA-sensing in EECs of flies and mammals remains an interesting question and warrants further work to mechanistically dissect such diversified EEC response to the same nutrients.

Our study has provided an integrated view of how a gut peptide modulates animal behavior by acting on very specific enteric neurons. Enteric neurons form the "enteric" brain that not only execute all basic functions in the absence of input from the brain[142], but also physically connect the gut to the brain with vagal afferent nerves[143]. While the mammalian ENS shows great complexity[144,145], the gut innervations by neurons have recently been detailed in flies[63]. Enteric neurons regulate many aspects of physiology in flies and mammals[20,146]. Given their sensory capabilities, vagal afferents are best positioned to regulate food intake, either through gut hormones[147,148] or by distension of the GI tract[63,149–152].

Surprisingly, the two NPFR-expressing enteric neurons identified in this work exhibit striking capacity in controlling feeding. This pair of enteric neurons translate signals on food nutrition sent by NPF$^+$ EECs. Importantly, their depolarization and silencing are both sufficient to decrease and increase food intake respectively, regardless of feeding conditions and systemic NPF levels, thus establishing themselves as previously unrecognized enteric neurons that play central role in appetite regulation. Like dedicated vagal afferent neurons, they have their cell bodies in the HCG outside the brain, innervate the anterior midgut to collect information and further send axons to the SEZ in the brain. The organization and function of the fly NPFR$^{ENS}$ neurons should stimulate the search for specific vagal afferent neurons that upon activation reduce appetite in human.

## Methods

### Fly strains and culture

Flies were reared on a standard cornmeal diet (210 g dry inactivated yeast, 900 g yellow cornmeal, 120 g soy flour, 100 g agar (Biosharp), 800 ml light corn syrup, 150 ml propionic acid and 12 L water) at 25 °C and 65% humidity with a 12-h light:12-h dark daily cycle, unless otherwise indicated. The animals were transferred to fresh food every third day. Only mated female flies were used in all our experiments. The following lines were obtained from the TsingHua Fly Center: *UAS-scute$^{RNAi}$* (THU2205); *UAS-NPF$^{RNAi}$* (THU2569); *UAS-Tk$^{RNAi}$* (THU2022); *UAS-stim$^{RNAi}$* (THU2581); *UAS-SERCA$^{RNAi}$* (THU2107); *UAS-PMCA$^{RNAi}$* (THU1887); *UAS-IP3R$^{RNAi}$* (TH02220.N); *UAS-CG1115S$^{RNAi}$* (THU3285); *UAS-Ekar$^{RNAi}$* (THU3080); *UAS-GluRIA$^{RNAi}$ 1* (TH201500449.S); *UAS-GluRIA$^{RNAi}$ 2* (THU2683); *UAS-GluRIA$^{RNAi}$ 3* (THU5238); *UAS-GluRIB$^{RNAi}$ 1* (THU2758); *UAS-GluRIB$^{RNAi}$ 2* (THU5273); *UAS-GluRIB$^{RNAi}$ 3* (THU5358); *UAS-GluRIIA$^{RNAi}$* (THU2659); *UAS-GluRIIB$^{RNAi}$* (THU3089); *UAS-GluRIIC$^{RNAi}$* (THU2049); *UAS-GluRIID$^{RNAi}$* (THU2151); *UAS-GluRIIE$^{RNAi}$* (THU3986); *UAS-Grik$^{RNAi}$* (THU3979); *UAS-KaiR1D$^{RNAi}$* (THU3982); *UAS-mGluR$^{RNAi}$ 1* (THU5288); *UAS-mGluR$^{RNAi}$ 2* (THU2115); *UAS-mtt$^{RNAi}$ 1* (THU0827); *UAS-mtt$^{RNAi}$ 2* (THU5594); *UAS-Nmdar1$^{RNAi}$ 1* (THU2118); *UAS-Nmdar1$^{RNAi}$ 2* (THU5286); *UAS-Nmdar1$^{RNAi}$ 3* (THU5287); *UAS-Nmdar2$^{RNAi}$ 1* (THU5240); *UAS-Nmdar2$^{RNAi}$ 2* (THU5249); *UAS-Nmdar2$^{RNAi}$ 3* (THU5862); *UAS-NPFR$^{RNAi}$* (THU2116); *UAS-ChAT$^{RNAi}$ 1* (TH02505.N); *UAS-ChAT$^{RNAi}$ 2* (TH201500313.S); *UAS-Ddc$^{RNAi}$* (THU2416); *UAS-Vmat$^{RNAi}$* (TH01473.N); *UAS-Gad$^{RNAi}$ 1* (TH02214.N); *UAS-Gad$^{RNAi}$ 2* (TH201500431.S); *UAS-VGAT$^{RNAi}$* (THU4304); *UAS-Hdc$^{RNAi}$* (THU2140); *UAS-Tbh$^{RNAi}$ 1* (TH02221.N); *UAS-Tbh$^{RNAi}$ 2* (TH201500898.S); *UAS-Tdc2$^{RNAi}$* (THU2075); *UAS-Trh$^{RNAi}$* (THU2052); *UAS-VGlut$^{RNAi}$* (THU2700). The following lines were obtained from the University of Indiana Bloomington Drosophila Stock Center (BDSC): *esg-Gal4* (BL#93857); *UAS-attp2 empty* (BL#36303); *UAS-attp40 empty* (BL#36304); *canton-s* (BL#64349); *tap$^{1.3}$-Gal4* (BL#46377); *UAS-nls-GFP* (BL#4776); *UAS-clumsy$^{RNAi}$* (BL#28351); *vm-Gal4* (BL#48547)[112]; *How-Gal4* (BL#1767)[153]; *GMR60E02-Gal4* (BL#39250); *GMR60G05-Gal4* (BL#39259); *GMR61H06-Gal4* (BL#39281); *GMR65C12-Gal4*

(BL#39348); *20XUAS-6xGFP* (BL#52262); *nsyb-FlpL;; UAS>stop > FLAG, UAS>stop > HA* (BL#64087); *UAS-shi^ts* (BL#66600); *Ddc-LexA* (BL#54218). *w^1118* was obtained from Vienna *Drosophila* Resource Center. *NRE-LacZ, esg-Gal4, tub-Gal80^ts, UAS-GFP*[154] and *UAS-hid* was kindly donated from Benjamin Ohlestin (University of Texas Southwestern Medical Center). *OreR* (BL#5) was kindly donated from the laboratory of Jianhua Huang (Zhejiang University). *Tkg-Gal4*[60] was kindly donated from the laboratory of Wei Song (Wuhan University). *UAS-TNT-imp* (BL#28841) and *UAS-TNT-G2* (BL#28838) were kindly donated from the laboratory of Zhihua Liu (Hubei University). *UAS-TrpA1* (BL#26263)[155]; *UAS-Denmark,UAS-nsyt:GFP* (BL#33065)[156]; *13XLexAop-myr:GFP, UAS-mCD8:RFP;;10XUAS-CaLexA*[119]; *8xLexAop-FlpL,UAS>stop>myr:GFP*[157]; *trans-tango*[122] and *13xLexAop-myr:GFP,UAS-mCD8:RFP*[157] strains were kindly donated from the laboratory of Yufeng Pan (Southeast University). *NPFR^RA/C-Gal4; NPFR^RB/D-Gal4; NPFR^RA/C-LexA; NPF^attP (NPF^null)* and *NPFR^attP (NPFR^null)* were kindly donated from the laboratory of Yi Rao (Peking University)[89]. *Pros^v1-Gal4,tub-Gal80^ts,UAS-GFP*[158] was kindly donated from Jean-François Ferveur (Université Paris-Sud). *UAS-pANF-EMD*[83] was kindly donated from David Deitcher (Cornell University). *NPF^sk1 (NPF^1)* and *NPFR^8* were kindly donated from Shu Kondo (Tokyo University of Science)[62]. *UAS-GCaMP6f* was kindly donated from Shan Jin (Hubei University). *UAS-tdTomato* was kindly donated from Kenneth Irvine (Rutgers University). *UAS-NPF*[111] was kindly donated from Todd Schlenke (University of Arizona). *UAS-mCD8:GFP, UAS-Redstinger*[159,160] was kindly donated from Woo Jae Kim (HIT Center for Life Sciences, HIT). The fly lines used are listed in Supplementary Table 1. No ethical approval is needed for the use of the fruit fly *Drosophila*.

## Generation of transgenic flies

**tap^1.3-A-Gal4, tap^1.3-B-Gal4 and NPF-0.7-GFP.** To generate gut specific driver and reporter constructs, primers shown below were used to amplify the regulatory regions of *tap* and *NPF*. The PCR products were first cloned into pENTR-D-TOPO (Thermo Fisher Scientific, Cat# K240020SP) vector, and then swapped into pBPGUw (to make Gal4 reporter) or pBPGUw-eGFP (to make GFP reporter) destination vector[161]. Germline transformation was performed in BestGene Inc to insert the tap^1.3-A-Gal4 at attP2 site, tap^1.3-B-Gal4 at attP40 and attP2 site and NPF-0.7-GFP at attP40 site. All the constructs were verified by sequencing.

Primer sequences:
*tap^1.3-A*_F: CACCTAAATTAGCCCCCTCGACAC
*tap^1.3-A*_R: AGATTCAATTACCATCAACTC
*tap^1.3-B*_F: CACCACGAGCTTTGATGATGCCG
*tap^1.3-B*_R: CGTCTCGCGTGCCCGCAC
*NPF-0.7*_F: CACCAGCGTTAATTAGTCAGAACGC
*NPF-0.7*_R: TGGGTGGGCGGTATGGAAATG

**NPFR^3XHA.** *NPFR^3XHA* was constructed using a CRISPR/Cas9 mediated homologous recombination method. Cas9 targeting site (GACTACCCTGTGCTTTAccg) was selected near the stop codon of *NPFR* to induce double strand breaks (DSBs).

To obtain guide RNA vector *(NPFR-gRNA)*, one pair of primers with targeting site was synthesized: *NPFR-gRNA-F*: gtcgGACTACCCTGTGCTTTACCG

*NPFR-gRNA-R*: aaacCGGTAAAGCACAGGGTAGTC

After annealing, guide RNA was subcloned into single guide RNA (sgRNA) vector (modified *PMD18T*, a kind gift from Haiyang Chen's lab), which was digested using BbsI (NEB, Cat# R3535S), by T4 DNA Ligase (NEB, Cat# M0202S). To assemble the sgRNA into the *PCR8* vector, one pair of primers with adaptor sequences:

*BsaI-U6-F*: ATGCGGTCTCCTGACGCTCACCTGTGATTGCTC

*BsaI-SgRNA-R*: ATGCGGTCTCGGAGTAAAAAAAGCACCGACTCGGTGC
was used to amplify the guide RNA. The PCR product and *PCR8* vector was digested using BsaI (NEB, Cat# R0535V). The digestion products were

assembled through the T4 DNA Ligase. The sgRNA (*PCR8-NPFR-gRNA*) was then exchanged to the *pUAST-attB* vector through attP/attB recombination (Invitrogen Gateway® LR Clonase® Enzyme Mix, Cat# 11791019) to obtain the *pUAST-attB-NPFR-gRNA*.

To induce homolog based integration and the plasmid cutting by the Cas9 vector, a *NPFR-Hom-3XHA* plasmid carrying a *3XHA* at the C-terminal of *NPFR* with two flanked homolog arms ( ~ 0.9 k and ~1.7 k respectively) was constructed as follows: the homolog arms were amplified (TOYOBO, Cat# KOD-211) from the fly genome

(primer pairs sequences:
*NPFR-5′*_F:GTGATCGTGTACCCCACGC *NPFR-5′*_R:CCGCGGCATCAGCTTGGT

*NPFR-3′*_F:AGCACAGGGTAGTCCTAAGG *NPFR-3′*_R:AAGTTAAGTGTTCGGCGGGT)and sub-cloned into *pEASY-Blunt* (TransGen Biotech, Cat# CB111-01). Then, three pairs of primers with a linker sequence were used to amplify

the N terminal homolog arm:
*NPFR-5′-1_F*:gccagtgccaagcttgcatgcGTGATCGTGTACCCCACGCG
*NPFR-5′-1_R*:aggaacatcgtatgggtaCCGCGGCATCAGCTTGGT
*3XHA tag:*
*HA-5′_F*:ggTACCCATACGATGTTCCTGACTATG
*HA-5′_R*:taggactaccctgtgctTCACGTGGACCGGTGTCCG and the C terminal homolog arm:
*NPFR-3′-1_F*:tgaAGCACAGGGTAGTCCTAAGGTCC
*NPFR-3′-1_R*:tacgaattcgagctcggtaccAAGTTAAGTGTTCGGCGGGTC.

The three segments were assembled into the *NPFR-Homo* plasmid by replacing the sequences between SphI (NEB, Cat# R3182V) and KpnI (NEB, Cat# R3142S) sites on modified *PMD18T* plasmid using the multi-site clone Kit (Vazyme, Cat# C113-02). All the constructs were verified by sequencing.

The *pUAST-attB-NPFR-gRNA* was integrated into the 51D site by microinjection (performed by Unihuaii. Ltd) to obtain the *NPFR-gRNA* transgenic fly. The *NPFR-gRNA* transgenic fly was crossed with *yw; nos-Cas9 (II-attP40)* to induce DSBs. The F1 embryos with DSBs were injected with *NPFR-Hom-3XHA* plasmid. After eclosion, they were single crossed with *yw122; If/CyO; MKRS/TM6B* flies of the opposite sex. The F2 male flies were single crossed with *yw122; If/CyO; MKRS/TM6B*, and the recombination events were verified with PCR (*NPFR-seq-F:* GCCGCGGTACCCATACGATG, *NPFR-seq-R:* CGAGCTCTTAGTCGCGTGTG, 997 bp) and immunostaining of HA. The efficiency of the recombination was about 6.5% (3/46).

## Generation of NPF antibody

Rabbit anti-NPF serum was generated by Eurogentec. Antigen was a synthetic peptide GEFARGFNEEEIF, which corresponds to the C-terminus of the NPF precursor. We thank Jan Veenstra for sharing the antigen.

## Immunostaining and fluorescent microscopy

Flies were anesthetized by $CO_2$. Then the intestines and brains of mated female flies were dissected in 1 × PBS (Solarbio, Cat# P1010) solution. The samples were fixed in 4% formaldehyde (Sigma, Cat# F8775) for 3 h for intestines or 30 min for brains at room temperature, washed three times for 20 min by 0.3% PBT (1 × PBS solution containing 0.3% Triton X-100 (Sangon Biotech, Cat# A110694-0500). Intestines and brains were incubated with primary antibodies for 3 h at room temperature, washed three times for 20 min by 0.3% PBT. Then samples were incubated in secondary antibodies for 3 h at room temperature, washed three times for 20 min by 0.3% PBT. Lastly, samples were incubated with 100 µL DAPI (1 µg/mL, Sigma, Cat# D9542) for 5 min, washed three times for three times by 0.3% PBT and mounted in 70% glycerol (Sinopharm Chemical Reagent, Cat# 10010618). Images were taken with Carl Zeiss LSM 800 confocal microscopy and then processed by Adobe Photoshop and Adobe Illustrator. The following primary antibodies were used in this paper:

chicken anti-GFP (1:10,000, Abcam, Cat# AB13970), rabbit anti-RFP (1:10000, Abcam, Cat#62341), mouse anti-Pros (1:200, Developmental Studies Hybridoma Bank, Cat# 528440), mouse anti-NC82 (1:100, Developmental Studies Hybridoma Bank, Cat# 2314866), rabbit anti-LacZ (1:4000, CUSABIO, Cat# CSB-PA009476LA01ENV), rabbit anti-PH3 (1:10000, Millipore, Cat# MMI-06-570), mouse anti-NPF (1:200, a kind gift from Veenstra, J. A.)[56], rabbit anti-NPF (1:4000, this paper), rabbit anti-Tk (1:4000, a kind gift from Benjamin Ohlstein)[162], rabbit anti-HA (1:4000, Cell Signaling Technology, Cat# 3724S), rabbit anti-NPFR (1:2000, RayBiotech, Cat# RB-19-0003-200), rabbit anti-AKH (1:10000, a kind gift from Wei Song)[163], rabbit anti-TH (1:4000, Abcam, Cat# AB112). The following secondary antibodies were used: Alexa Flour goat anti-chicken 488 (1:4000, Invitrogen, Cat# A11039), Alexa Flour goat anti-rabbit 488 (1:4000, Invitrogen, Cat# A11008), Alexa Flour goat anti-rabbit 555 (1:4000, Invitrogen, Cat# A21428), Alexa Flour goat anti-mouse 555 (1:4000, Invitrogen, Cat# A21422), Alexa Flour goat anti-mouse 647 (1:4000, Invitrogen, Cat# A21235).

For NPF intensity and pANF-EMD intensity measurement, guts from mated female flies were dissected, fixed, stained in the same setting. Fresh primary antibodies were used each time. Images were taken with Carl Zeiss LSM 800 confocal microscope using the same setting. The average protein intensity of single cell was calculated by ImageJ.

For relative CaLexA intensity measurement, *13XLexAop-myr:GFP, UAS-mCD8:RFP;; 10XUAS-CaLexA/NPFR^{ENS}-Gal4* mated female flies were used in this experiment. Brains and gut were dissected together and put on ice. Samples were fixed, stained in the same setting. Fresh primary antibodies were used each time. Images were taken with Carl Zeiss LSM 800 confocal microscopy in the same setting. The total GFP and RFP intensity of single cell body was calculated by ImageJ. Measuring the total GFP and RFP intensity in the same area next to the cell body as blank intensity. Relative CaLexA intensity = (total GFP intensity - blank GFP intensity) / (total RFP intensity - blank RFP intensity).

### Food intake measurement

The Capillary Feeder (CAFE) assay[72], Manual Feeding (MAFE) assay[88] and dye-based food intake measurement were used to measure the food intake of 3-5 d mated female flies in this paper.

For the CAFE assay, flies of the indicated ages were fasted for 3 h by placing them in vials containing only water. Five flies were collected as a group and transferred to a vial containing $ddH_2O$ at the bottom and a capillary tube (World Precision Instruments, Cat# 1B100F-4) inserted through a 10 µl pipette tip. The capillary contained 10 µl of 5% sucrose (Sinopharm Chemical Reagent, Cat# 10021418) with 0.25% (v/v) blue dye solution (AmeriColor, Cat# 102) (unless otherwise stated) and Halocarbon oil 700 (Sigma, Cat# H8898) at the top. To account for evaporation, we placed 2 vials with capillary tubes containing 10 µL of 5% sucrose with 0.25% (v/v) blue dye without flies as a negative control. The liquid level in each capillary tube was marked at the start of the assay. Flies were allowed to feed for 24 h, after which we marked the level of fluid in each capillary. Total food consumption was calculated as the difference in fluid levels in the capillaries, corrected for the average evaporation that occurred in the negative control vials.

For the MAFE assay, flies of the indicated ages were fasted for 36 h by placing them in vials containing only water. Flies were then individually fixed in a 200 µl pipette tip and blocked with cotton. The proboscis was exposed. Flies were then presented with 5 µl of 5% sucrose containing 0.25% (v/v) blue dye liquid food in a glass capillary until they stopped responding to food stimuli for ten serial food stimuli. Food consumption was calculated on the basis of the volume change before vs. after feeding and the time of feeding.

For the dye-based food intake measurement[38,164], 20 flies of the indicated genotypes were collected as a group. 10% sucrose, 25% coconut oil (v/v) or 10% yeast were added to standard cornmeal diet to produce a high-sugar, high-fat or high-protein diet, respectively. In

order to measure the food intake of the flies under physiological conditions and to reduce the effect of defecation on the measurements, fasting was omitted in these experiments. Flies were transferred to new vials with food containing 0.5% erioglaucine disodium salt (Sigma, Cat# 861146) for 24 h to allow flies to consume blue food. To avoid food and fly tissue interference, 20 flies of the same genotype and age were placed on food without erioglaucine disodium salt as a negative control. Flies were collected in 1.5 ml tubes and processed at −20 °C for 2 h. Flies were snap frozen in liquid nitrogen for 1 min and then were shaken vigorously to remove the heads, legs and wings of flies. The remaining parts of the flies were collected in new tubes. 600 µL of PBS solution was added to the tubes, homogenized and centrifuged (15900 × g, 30 min). 100 µL supernatants were added to a 96-well plate and the absorbance was measured at 620 nm. Three measurements were made for each sample. Absorbance was calculated as (mean absorbance of flies feeding on blue food) - (mean absorbance of negative control flies).

### Defecation and gut-clearance assay

We performed the defecation and gut-clearance assay according to the previously described method with slight modifications[61]. For the defecation assay, we first fed the mated female flies by placing them in vials containing 5% sucrose/blue dye for 24 h. We then divided 5 flies in each group into new vials. Two capillaries containing 10 µl of 5% sucrose with 0.25% (v/v) blue dye solution with Halocarbon oil 700 at the top were inserted into the vials using 10 µl pipette tips. The filter papers were placed on the top and the wall of each vial. The blue deposits on the filter paper of each vial were counted after 24 h.

For gut clearance assays, mated female flies were first fed 5% sucrose containing 0.25% (v/v) blue dye for 48 h, and ten flies with blue abdomen were transferred to a new vial containing 5% sucrose only. After 24 h, flies were counted according to whether they still had a blue abdomen or not.

### Measurement of the mass and metabolite content

To measure the mass of flies, 10 mated female flies at indicated ages were anesthetized by $CO_2$ and collected in a tube. Measuring the mass of flies and the tube by precision balance (Sartorius, Cat# BSA223S). The mass of single fly was calculated as (the mass of flies and the tube - the mass of the tube) /10.

To measure the glucose content of flies, 5 mated female flies were weighed and then homogenized in 1 ml 70 °C $ddH_2O$. Glucose (Go) assay kit (Sigma, Cat# GAGO20) was used to measure the glucose of supernatant. The absorbances at 540 nm were recorded after reaction.

BCA protein quantification kit (Thermo Fisher, Cat# 23225) was used to measure the protein content of flies. Before measuring, 20 mated female flies were collected in a tube, weighed and then homogenized in 1 ml PBS solution. Heat-inactivate at 95 °C for 5 min. The absorbances at 562 nm were recorded after reaction.

To measure the content of TAG, 10 mated female flies were collected on ice in screw-cap tubes and weighed. Add 250 ul 1xPBS containing 0.1% Tween-20 (Sigma, Cat# P1379) into the tube and homogenize for 30 s. Heat-inactivate (HI) at 70 °C for 5–10 min. Centrifuge for 3-5 min and transfer 150 ul supernatant to new tubes. Distribute 20 ul HI homogenate and add 20 ul PBS (control) or Triglyceride Reagent (Sigma, Cat# T2449) to 96-well plate. Gently tap plate to mix and centrifuge at maximum speed for 3 min. Incubate for 30 min at 37 °C. Add 40 ul/well standards (free glycerol, Sigma, Cat# G7793) to plate plus blank background, 140 ul $H_2O$ with no reagents. Add 100 ul Free Glycerol Reagent (Sigma, Cat# F6428) to samples and standards. Incubate for 5–10 min at 37 °C. The absorbances at 540 nm were recorded after reaction. TAG = free glycerol (Triglyceride reagent-treated) - free glycerol (PBS-treated).

For Oil Red O staining, midguts from mated female flies were dissected in cold 1xPBS, then fixed in 4% formaldehyde for 20 min.

After fixation, specimens were rinsed three times with distilled water and incubated for 25 min in Oil Red O (Sigma, Cat# OO625) solution (mix of 6 ml isopropanol with 0.1% Oil Red O and 4 ml distilled water, prepared fresh and filtrated to remove the precipitation).

### Rearing in germ-free conditions

Germ-free flies were generated as previously described[165] with slight modifications. *esg-Gal4, tub-Gal80^ts, UAS-GFP* virgins were allowed to mate with control (*UAS-attp empty*) or *UAS-scute^RNAi* males and lay eggs on 1% agar plate covered with diluted yeast paste at 18 °C for no more than 8 h. Embryos of the indicated genotype were collected from the agar plate and washed three times with 1 ml 3.3% Walch (1 ml Walch + 30 ml sterile water). The embryos were then washed once with 1 ml 70% absolute ethanol (Sinopharm Chemical Reagent, Cat# 10009218) and 1 ml 2.7% sodium hypochlorite solution (Macklin, Cat# S817439). Finally, embryos were washed three times with sterile 0.3% PBST and transferred to sterile standard cornmeal feed at 18 °C. The development of flies in germ-free condition is slower than that of flies in conventionally fed condition, so at 90 h APF, *esg^ts>scute^RNAi* pupae were transferred to 30 °C for 10 h to block the formation of EECs and then returned to 18 °C until eclosion. Food intake of 5 d AE conventionally fed and germ-free flies was measured by both the CAFE assay and the dye-based food intake measurement. For the CAFE assay, germ-free flies were collected in a sterile environment and then fasted for 3 h by placing them in sterile vials containing only sterile water. The vials, ddH$_2$O, capillary tubes, 10 µl pipette tips and 5% sucrose with 0.25% (v/v) blue dye used in the CAFE assay were sterilized using a vertical autoclave (Zealway, Cat# GI80TW). The CAFE assay was performed in a sterile environment. For the dye-based food intake assay, germ-free flies were collected in a sterile environment and then transferred to new sterile vials with sterile food containing 0.5% erioglaucine disodium salt (Sigma, Cat# 861146) for 24 h to allow the flies to consume blue food. Experiments were conducted in a sterile environment.

### Eliminating EECs during pupal development

EECs from adult flies were generated in the pupal stage by pupal ISCs after 44 h APF (after pupal formation) at 25 °C[47]. Therefore, we performed a genetic approach to inhibit the production of EECs during the pupal stage. *esg^ts>scute^RNAi* flies were reared at 18 °C. 80 h after pupal formation (APF), *esg^ts>scute^RNAi* pupae were transferred to 30 °C for 10 h, and then were returned to 18 °C. Flies blocked in EEC formation during the pupal stage were designated as *esg^P>scute^RNAi* flies. The midguts of *esg^P>scute^RNAi* flies were dissected at 3d, 7d and 10d AE.

### Fabrication of the Temperature Control Device

The Temperature Control Device (TCD) was composed of five parts: (1) the removable fly-placing pad, (2) the heating element, (3) the temperature sensor and control circuit, (4) computer and the temperature control software, and (5) a fanner to decrease the temperature of the head.

(1) The removable fly-placing pad consisted of three parts: (a) a glass slide (7.5 cm × 2.5 cm × 0.1 cm, Citoglas, China), (b) a copper metallized polyester film (8.0 cm × 2.0 cm) that was sticked on the longer side of the glass slide and (c) an adiabatic Polydimethylsiloxane (PDMS) layer (7.3 cm × 2.6 cm × 0.5 cm) to reduce heat loss. To fix the neck of flies, we made eleven rectangular-shaped gaps (0.02 cm × 0.1 cm) by a UV laser marker (HGL-LSU3/5EI, Huagong Laser, Wuhan, China) on the copper metallized polyester film. To immobilize flies, we made eleven trapezoidal holes (0.25 cm × 0.35 cm × 0.4 cm × 0.35 cm) on the PDMS layer relative to each gap on the copper metallized polyester film.

(2) The heating element was an aluminum alloy resistance wire (12 Ω) sealed inside a polyimide film (10 cm × 4.5 cm × 0.02 cm, QINGBANG) connected with the control circuit.

(3) The temperature sensor was an analog temperature sensor (LM35D, ZHONGBEST). After fixing in PDMS (5.5 cm × 2.0 cm × 0.85 cm) cuboid, this cuboid was loaded on a glass slide (7.5 cm × 2.0 × 0.1 cm) and sticked on the heating element.

(4) An application was developed in LabVIEW to provide a readable user interface for temperature monitoring. A PC with Windows 10 operating system was used in this experiment. The information for control circuit and temperature control application had been uploaded to figshare (https://figshare.com/articles/software/Drosophila_local_temperature_control_device/13451204).

### Eliminating EECs in adult midgut by TCD

*UAS-hid, pros^v1-Gal4, tub-Gal80^ts, UAS-GFP* (*pros^ts > GFP+hid*) mated female flies were reared at 18 °C. 5 days AE, *pros^ts > GFP+hid* females were used to eliminate EECs. The TCD was placed in a cold room at 18 °C. After fixing the flies in the fly-placing pad, an adiabatic PDMS layer was placed over the flies. The flies and fly-placing pad were placed on the heating element for 12 h, after which the flies were transferred to new vials at 18 °C with standard fly food for further experiments. The temperature setting in the application was 30 °C. The heads of the flies were outside the heating region, so the TCD only kept the abdomens of the flies at 30 °C. We named the flies in which adult EEC elimination was processed in the TCD *pros^TCD > GFP+hid* flies.

### Gut microbiota sequencing

20 midguts of mated female flies of the indicated genotypes and ages were dissected in 1x PBS solution, and DNA was extracted using TIANamp Genomic DNA Kit (TIANGEN Biotech, Cat# DP304-02). 16 s rRNA sequencing and analysis was performed by Majorbio. The number of sequences obtained from all 6 samples was 359,363. The number of bases was 137,171,954 bp. The average length of the sequences was 381.708617748 bp. The species taxonomy was determined using operational taxonomic units (OTU). The species differences between the gut microbiota in *esg^P>attp empty* 3d AE and *esg^P>scute^RNAi* 3d AE flies were performed using the Wilcoxon rank sum test method at the phylum level based on OTU. The results were plotted as a histogram. The gut-microbiota sequencing data generated in this study have been deposited in the Figshare [https://doi.org/10.6084/m9.figshare.25458226.v1].

### NPF feeding and injection

C-terminal amidated NPF peptide (SNSRPPRKNDVNTMA-DAYKFLQDLDTYYGDRARVRF-NH2) was synthesized by DgPeptides co., ltd. The synthetic NPF peptide was diluted to 10 µM in 1xPBS and stored at −80 °C. 100 nM NPF peptide solution was loaded from the capillaries into micropipettes produced by Micropipette Pullers (Sutter instrument, Cat# P-1000) and then injected into the thoraces between the first and second legs of female flies cooled on ice using a pneumatic PicoPump (World Precision Instruments, Cat# SYS-PV820). A Zeiss Stemi 508 stereomicroscope with M stand was used to visualize the micropipettes and the thoraces of the flies. Approximately 40nL NPF solution was injected into a fly. Injected flies were transferred to vials containing standard fly food for further experiments.

### Yeast extract, yeast paste feeding and single amino acid screen

3 days AE mated female flies reared at 25 °C were used for those feeding experiments. Oxoid™ Yeast Extract Powder (Cat# LP0021T) was purchased from Thermo Scientific. For yeast extract feeding experiments, flies were fed a 5% sucrose solution containing 5% yeast extract for 48 h. Yeast paste was a mixture of distilled water and yeast in a 1:1 weight ratio. For yeast paste feeding experiments, flies were fed yeast paste for 48 h. L-alanine (Cat# A7627), L-argine (Cat# A5131), L-asparagine (Cat# A0884), L-aspartate (Cat# A8949), L-cysteine (Cat# C1276), L-glutamate (Cat# G1251), L-glutamine (Cat# G3126), L-glycine (Cat# G7126), L-histidine (Cat# H8000), L-isoleucine (Cat# I2752),

L-leucine (Cat# L8912), L-methionine (Cat# M9625), L-phenylalanine (Cat# P2126), L-proline (Cat# P0380), L-serine (Cat# S4500), L-threonine (Cat# T8625), L-tryptophan (Cat# T0254), L-tyrosine (Cat# T3754), L-valine (Cat# V0500), L-lysine (Cat# L5626) were purchased from Sigma-Aldrich. For single amino acid screening experiments, 5% sucrose solution containing 1% single amino acid was used to feed flies for 48 h.

### RT-qPCR
Total RNA was extracted from dissected midguts (50 guts per sample) or brains (150 brains per sample) using RNAprep Pure Tissue Kit (TIANGEN Biotech, Cat# DP431). cDNA was synthesized using GoScript™ Reverse Transcription kit (Promega, Cat# A2790). 0.5 mg total RNA was used for reverse transcription, and the cDNA was diluted 10 times with water and further used in real time PCR. Real time quantitative PCR was performed in at least triplicate for each sample using GoTaq® qPCR System (Promega, Cat# A6001). Expression values were calculated using the ΔΔCt method and relative expression was normalized to *RpL23*. The expression in control sample was further normalized to 1.

Primer sequences are indicated in Supplementary Table 2.

### Calcium imaging
Calcium live imaging was performed as previously described[76,166]. For calcium imaging, *UAS-GCaMP6f, UAS-tdTomato* was expressed under the control of *tap^{1-3}-B-Gal4*. Mated female flies were used in all the experiments.

### Prepare live imaging buffer (LIB)
8 ml Schneider medium (Thermo Fisher, Cat# 21720001) was supplemented with 2 ml fetal bovine serum (Thermo Fisher, Cat# 10091148) and 50 μl insulin solution (Biosharp, Cat# BS901-25mg, dissolved in Hcl, 40 μg/μl). pH was adjusted to 7.0.

### Prepare live imaging gel (LIG)
0.5 g of gelatin (Sigma-Aldrich, Cat# G2500) was added to 5 mL of LIB and then heated at 50 °C to melt the gel. Both LIB and LIG were divided into 500 mL aliquots and stored at 4 °C for up to 1 week. Aliquots of LIG were heated to 37 °C prior to experiments.

### Prepare midguts for live imaging
Two pieces of cover glass with a size of 10 × 22 mm were attached to a lumox® dish 50 (Sarstedt, Cat# 15090935) using LIG, with a gap of ~1 cm between them. Intact guts were dissected in LIB and transferred to a 22 × 22 mm cover glass. Excess LIB was carefully removed with filter paper. A volume of 80 μl LIG at 37 °C was dropped into the 1 cm gap, then the 22 × 22 mm cover glass was quickly placed on the top of the 10 × 22 mm cover glasses to cover the guts with LIG without air bubbles. After the LIG was cooled down and stabilized, the cover glasses were finally sealed with Halocarbon oil 27 (Sigma-Aldrich, Cat# H8773) to prevent evaporation.

### Setting up time-lapse experiments on confocal microscopy
*GCaMP6f* calcium signals and *tdTomato* signals were captured using a Zeiss LSM 800 confocal microscope. Zeiss Definite Focus 2 was used to avoid focus drift. Time lapse images were acquired using ZEN 2.1 with Time Lapse Module. A single-layer image of 512 × 512 pixels (319.45 μm × 319.45 μm) was acquired every second for 10 min at room temperature (25 °C) with a pixel time of 1.03 μs and fixed laser power, pinhole and other settings for all time-lapse experiments. *GCaMP6f* emission was recorded at 400-533 nm and *tdTomato* emission was recorded at 579-700 nm. *GCaMP6f* and *tdTomato* fluorescence quantification of each cell was performed manually using ImageJ for each frame. Oscillation frequency was determined by counting individual peaks of the *GcaMP6f/tdTomato* fluorescence emission ratio observed during 10 min recordings. Heat maps were generated using Matlab. Videos were exported uncompressed from ZEN 2. Genotypes, feeding conditions, scale bars and relative time were added in ZEN 2.

### Statistics
Statistical significance was determined using the two-sided unpaired *t*-test in GraphPad Prism 8 (GraphPad software) and expressed as *P* values. All statistics results are presented as mean ± SD. Results of mRNA expression obtained by qPCR are presented as mean ± SD of at least 3 independent biological samples. All statistics graphs were generated using GraphPad Prism 8. No sample size estimation or inclusion/exclusion of data or subjects was performed in this study.

### Reporting summary
Further information on research design is available in the Nature Portfolio Reporting Summary linked to this article.

## Data availability
All data generated or analyzed during this study are available as a Source data file. Source data are provided with this paper. The gut-microbiota sequencing data generated in this study have been deposited in the Figshare database without accession code [https://doi.org/10.6084/m9.figshare.25458226.v1]. Additional data are available upon request to Dr. Zheng Guo (guozheng@hust.edu.cn). Source data are provided with this paper.

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

## Acknowledgements

We thank B. Ohlstein, G. Struhl, W. Song, X. Huang, F. Guo, YF. Pan, ZH. Liu, LL. Zhang, BDSC, VDRC, DGRC, and Tsinghua Fly Center for fly strains; YM. Lu, W. Song and DSHB for antibodies. HY. Chen, ZH. Wang and JQ. Ni for plasmids. B. Ohlstein, ZH. Liu, LY. Zhang, YF. Pan, QP. Wang, F. Guo, ZF. Gong, Y. Chen and YM. Lu for critical comments and insightful suggestions. This work was supported by grants from the National Natural Science Foundation of China to Z.G. (31970817, 31771625, 32271074) and to ZZ.Z. (31871469, 32170509).

## Author contributions

JJ.G., ZZ.Z. and Z.G. conceptualized, designed experiments. JJ.G. performed genetics, screens, immunohistochemistry, measurement of metabolic indexes, feeding assay, calcium imaging, statistical analyses and figure design. S.Z. identified the conditions to eliminate EECs during the pupal stage. JJ.G., P.D., ZG.W. and Z.G. designed the temperature control device. B.L. and ZZ. Z. designed and identified the EEC specific driver. ZZ.Z and Z.G. supplied resource and funding. ZZ.Z. and Z.G. supervised the project. JJ.G., ZZ.Z. and Z.G. wrote the manuscript.

## Competing interests

The authors declare no competing interests.
