## [Peer Review File · Nature Communications]

Dietary L-Glu Sensing by Enteroendocrine Cells Adjusts Food Intake via Modulating Gut PYY/NPF SecretionREVIEWER COMMENTS

Reviewer #1 (Remarks to the Author):

Manuscript: Dietary L-Glu Sensing by Enteroendocrine Cells Adjusts Food Intake via Modulating Gut PYY/NPF Secretion

This manuscript by Gao et al examines a gut-brain pathway that regulates feeding via NPF neuropeptide. They identify a population of gut enteroendocrine cells (EECs) that can sense dietary L-Glutamate via a metabotropic glutamate receptor. This in turn influences NPF release from the EECs which then activates its receptor on dopaminergic enteric neurons to regulate food intake. Overall, this is a fantastic study that carefully dissects the entire pathway from sensory stimuli to behavioral output. It complements 2 recent studies on gut NPF signaling (<https://doi.org/10.1038/s41467-021-25146-w> and <https://doi.org/10.1038/s42255-022-00672-z>). One interesting finding of note is the link between calcium oscillations and peptide release.

The study is performed to a high-standard and the manuscript is well-written and easy to follow. I only have a few minor comments regarding the methods.

Figure 2A - what about expression in the VNC? Please include these images.

Figure 3E: Could you please comment on the calcium levels with SERCA and PMCA RNAi? What is the cause of the large variability between different samples? Some of them have high calcium levels throughout and some have none.

Line 457: cardiac should be cardiaca

Line 654: add "in" after "expressed"

Line 869: "Use fresh primary antibodies every time". Please correct the grammar

Lines 1037 to 1039: please correct the sentences.

RT-PCR : Was genomic DNA removal step included? Or are the primers designed over exon-exon boundaries?

Calcium imaging: what was the reason for making the buffer with insulin and FBS instead of using hemolymph like saline? At what temperature was the experiment performed? RT or 40deg?

Sex of the flies used: I assume that females were used for all experiments. Were they mated or virgin? Please include this information.

Reviewer #2 (Remarks to the Author):

Title: Dietary L-Glu sensing by EEC adjusts food intake via modulating gut PYY/NPF secretion.

Proper sensing of dietary protein intake is believed to act as an essential internal system for the maintenance of physiological homeostasis of an animal. In this study, Gao et al. analyzed the role of enteroendocrine cells (EECs) in protein sensing by taking advantages of genetic tools available in *Drosophila* model system. They showed that (1) Specific elimination of EECs by generating *esg-p>scute-RNAi* flies and *pros-TCD>hid* flies led to a significant increase of food intake, indicating that EECs function to inhibit food intake; (2) By analyzing microbiome, they showed no significant differences between control flies and EEC-less flies; (3) By developing EEC-specific *gal4* (*tap1.3B-gal4*), they showed that neuropeptide secreted from *tap1.3B*-positive EECs inhibits food intake; (4) When synthetic NPF peptide was injected into the body cavity, increased food appetite seen in *tap1.3B>NPF-RNAi* flies or EEC-less flies (*esg-p>scute-RNAi* flies) was abolished; (5) protein-rich diets or two specific amino acids (Glu and Asn) enhance the intensity of NPF staining in EECs (i.e. reducing NPF secretion by promoting NPF retention in EECs); (6) Glu sensing attenuates calcium oscillation in EECs to reduce NPF secretion; (7) EECs sense Glu through mGluR; (8) Glu inhibits gut NPF release, and thus inactivates two neurons (using *NPFR-ENS-gal4*) located in HCG; (9) *NPFR-ENS*-positive neurons use dopamine to control food intake. Taken all together, they concluded that dietary Glu sensing by EECs adjusts food intake via modulating gut NPF secretion and subsequent the activity of *NPFR*-positive enteric neurons.

General comments.

Although I appreciate technical quality of the experiments, there are problems in the logic of the experimental setting and data interpretation. More experiments are needed to support their claims. At present, at least in a current form of the manuscript, I do not agree the conclusion of the study.

1. The key finding that the gut-derived NPF suppressed sugar intake (since the authors measured sugar intake in their assay) has been already shown by Kim Rewitz group (Nature Metabolism 2022). The novelty has been undercut.
2. The finding that a non-essential amino acid (L-glu) promotes food intake is not consistent with other previous literatures including mammalian ones, which showed that protein consumption suppresses food intake. The authors measured food intake with a food containing only sugar (5% sucrose), not any other macronutrients. For the experiments of food intake, they should examine other types of foods including a standard cornmeal diet (SCD), protein-rich diet (also food containing essential amino acids or non-essential amino acids or both), carbohydrate-rich diet and fat-rich diets.
3. The relationship between L-glu and *DH44* neurons is very tenuous at best. In the experiments found in previous publications, they used sucrose solution containing amino acid mixture for food

intake assay. For example, Kim and Kanai et al demonstrated that DH44 neurons are dispensable for amino acids or essential amino acids sensing. The authors should take the part out from the manuscript.

4. The amount of single non-essential amino acids such as L-glu may not be abundant in the proximal R2 region of the gut. Dietary proteins get digested and degraded in the R3 compartment where acidic digestive enzymes are profusely available. Before R3, most of the dietary proteins may be present as peptides, not as single amino acids. So it begs a question as to why NPF would be regulated in the proximal R2 by L-glu. In this context, it would be interesting to see whether NPF secretion in R2-located EECs is differently regulated by Glu ingestion, when compared with NPF secretion in R3/4-located EECs.

5. The authors used the CAFE assay to measure the feeding amount of five flies, using 5% sucrose as the food source. However, the average feeding amounts of the control flies in each figure are diverse, even at the same temperature. For example, in Figure 1d, the control flies' feeding amount is around 1 microL daily, but in Figure 1g, the control flies eat around 2.5~3 microL daily at the same temperature. Similarly, in Figure 5a, the control flies consumed around 5 microL in a day. It is possible that the authors mixed the male and female feeding data, and they need to clarify this point. Also, the different feeding amounts may indicate that the genetic background of the flies in each figure is not identical. Additionally, the authors did not clearly indicate whether they used male or female flies in some figures. As the authors measured the feeding assay in a group of five flies, they cannot rule out the possible social interactions in the group of flies.

6. In vivo function of EECs is unclear. They showed that EEC-less flies (*esg-p>scute-RNAi* flies and *pros-TCD>hid* flies) showed similar metabolic phenotypes when compared to control flies. How do they maintain those flies? For the experiment, do they use flies reared on a standard cornmeal diet (SCD)? If this is the case, do EEC-less flies increase intake of a SCD? If elimination of EECs led to an increase of SCD intake, while showing metabolic phenotypes similar to those of the control flies, what is the in vivo function of EECs?

7. The involvement of microbiome is unclear. In their analyses, it is very strange to see the Bacteroidetes in the fly microbiome (Normally, *Drosophila* microbiome does not contain Bacteroidetes). Do they have some explanation about this? They showed no significant differences between control flies and EEC-less flies. However, they analyzed the microbiome only at a single time point (at 3d AE, i.e. 3 days after eclosion). Microbiome of the young flies (such as those at 3d AE) is very simple and immature. Therefore, to see the cross relationship between microbiome and EEC-less condition (or phenotype), they should also analyze the microbiome of older flies (e.g. 10 d AE). Furthermore, they should examine the difference in terms of food intake between germ-free EEC-less flies and conventional EEC-less flies.

8. Similar to the above point-5, do *tap1.3B>NPF-RNAi* flies show increased intake of a SCD?

9. There is a difference (when compared to control flies) in terms of glucose content, TAG, and oil red O staining between *tap1.3B>NPF-RNAi* flies and EEC-less flies (*esg-p>scute-RNAi* flies). Why?

10. It is important to see whether brain NPF expression level and patterns are altered in *tap1.3B>NPF-RNAi* flies or EEC-less flies (*esg-p>scute-RNAi* flies). Similarly, are brain NPF expression levels and patterns altered in response to protein-rich (or Glu) diet?

11. It is important to see whether brain NPFR signaling is activated by synthetic NPF injection into the body cavity. Similarly, is brain NPFR signaling activated in response to protein-rich (or Glu) diet?

12. They claimed that the orexigenic effect of brain NPF override the role of gut NPF in restricting appetite. To prove this issue, they should introduce NPF expression in NPF mutant animals in a tissue-specific manner. For this, they should examine food intake level (sucrose as well as SCD) by using NPF mutant flies expressing gut NPF (using tap1.3B-gal4>UAS-NPF) or brain NPF (brain-specific gal4>UAS-NPF).

13. Based on the results in Fig3a-b, they concluded that “this implied a role for EECs in AA sensing”. However, these data in fact indicate that expression of gut NPF is induced by protein-rich diet, but not carbohydrate-rich or fat-rich diet. And this just implied a possible role for gut NPF in protein or AA-sensing.

14. It would be interesting to see the NPF expression level between germ-free and conventional flies.

15. For the figure 3C, they used only 1% of each amino acid. Therefore, with this experimental setting, it would be impossible to conclude that NPF is regulated by only Glu and Asn. As each type of food or protein has unique amino acid content, the assay should be performed in a dose-dependent manner using increasing amounts of each amino acid.

16. They should check whether tap1.3B-gal4 expression (by using tap1.3B-gal4>uas-gfp) is altered in response to protein-rich (or Glu) diet.

17. Are there any cell bodies of NPFR-ENS-positive neurons in the brain? In the fig 7a, it would be possible that cell bodies of NPFR-ENS-positive neurons are also seen in the brain. This should be checked.

18. Do Axons from cell bodies of NPFR-ENS-positive neurons located in HCG actually reach SEZ of brain?

19. The authors ruled out L-Asn as a candidate without conducting any experiments. However, Figure 3c and 3d shows that L-Asn successfully prevents the secretion of NPF in EEC. Therefore, they cannot rule out the possibility that L-Glu and L-Asn work together to suppress the secretion of NPF. The authors should investigate the potential role of L-Asn in this story to provide a complete understanding. For example, it is important to see whether gut NPF secretion and the activity of enteric NPFR-positive neuron are also modulated by Asn ingestion.

20. The authors developed a well-controlled temperature control device. However, this technique may not be necessary to use in this paper because previous studies and authors have already established better ways of manipulating gut-EEC specifically. While I agree that this technique is useful, it may be slightly out of focus for this story and not crucial for the paper's main message.

21. The authors identified two EEC-secreted hormones, NPF and Tk, as potential regulators of feeding behavior. They ruled out Tk as a possible EEC-derived feeding regulator; however, based on their data, they should consider increasing the trial number and re-examining this point. For instance, in Extended Data Figure 3d, knocking down Tk using tap1.3B-Gal4 seems to increase food intake. While they showed that the increase in food intake amount is not significant, if they increase the trial number, it may result in significant food overconsumption. Every figure they showed about Tk exhibits a similar pattern with NPF manipulation. Therefore, they should increase the trial number and recheck this point to confirm their findings.

22. The authors argued that EEC only secretes NPF after activation, but not Tk. If both NPF and TK-containing vesicles exist in EEC, they should explain how only NPF-containing vesicles are secreted,

but not those of TK. Although there are several possibilities that only one vesicle can be secreted while the whole cell is activated, it is difficult to understand how one specific vesicle type is secreted after the whole cell activation.

23. To clearly confirm their hypothesis, it would be essential to measure the circulating amount of NPF before and after L-Glu treatment. Western blotting would not be an ideal method to measure circulating hormone levels. The field commonly uses ELISA and mass spectrometry to measure hormone levels. Therefore, to confirm their hypothesis, the authors should measure the changes in the circulating NPF levels using appropriate methods.

24. The previous study (Malita et al., 2006) already clearly showed the notion that "gut-NPF suppresses food intake." Although the previous study showed the critical role of the mating status of a female in regulating protein feeding in flies, the authors did not mention the sexual dimorphism and post-mating behavioral switch in females that significantly affect food intake. Therefore, the authors should include this information to provide a comprehensive view of the regulation of feeding behavior.

25. In the discussion section, the authors paralleled PYY in mammals and gut-NPF in flies. However, because of the multiple discrepancies between these hormones, it is difficult to accept their suggestion. Therefore, the authors should provide a more detailed comparison of the two hormones to strengthen their argument.

Reviewer #3 (Remarks to the Author):

This study provides a quite complete picture of a gut-brain signaling axis in regulating food intake in *Drosophila*. The authors found that neuropeptide F (NPF) secreted from a subset of enteroendocrine cells in the middle midgut acts on a pair of NPFR-expressing enteric neurons, which then projects to the presumptive feeding center to inhibit food intake. Counter intuitively, the authors found that protein-rich diet can inhibit NPF secretion from these EECs, thereby eliciting a feed forward mechanism to promote food intake.

Pros:

The analysis in general is very thorough and with rigor, and most conclusions are quite compelling. The authors have developed several approaches for gut-specific manipulation of EECs, which has not been achieved by the majority, if not all, of the previously published studies in this field.

Cons:

The regulation of gut NPF secretion specifically by amino acid reported here is not consistent with the previously reported observations. In the mechanistic part, the signaling route independent of AKH+ neurons is also not consistent with the previous observations. Amino acid sensing by taste sensory organs is known to have a role in appetite regulation, other variables such as gender and mating status may also have an effect on food preference. These factors might complicate the interpretations of some results reported in this study. The specificity of some GAL4 drivers used in the study warrants further validation.

Specific points:

1. The results from this study suggest the exact opposite functions of NPFs in the brain and the periphery system in controlling feeding behavior, as the reduced food intake were observed in NPF null mutant flies. To strengthen the conclusion about the opposite functions, the authors should perform brain-specific manipulation of NPF side-by-side with the gut-specific manipulations to see if the brain-derived NPF is indeed orexigenic. Perhaps the TCD device described in this study can be adopted to specifically manipulate NPF neurons in the brain.
2. The authors show co-localization of NPF or tap 1.3B-gal4 with NPFR. However, the previous single cell RNA-seq data showed mutually exclusive expression of NPF and NPFR in EE cells (Guo et al., 2019). Malita, et al, 2022 also demonstrated the expression of NPFR in AstC+ EECs, but not TK+ EEs that express NPF. Therefore, it is worrisome whether the described pattern of NPF/NPFR is correct, and whether tap 1.3B-gal4>NPFR-RNAi can effectively deplete NPFR in EECs. The expression pattern of NPF/NPFR need to be further validated such as by co-immunostaining or by intersectional strategies (such as NPFR-LexA \cap Tap1.3B-Gal4 or NPF-Gal4).
3. NPFR is expressed in many tissues and organs, and the authors only examined its role in EECs and in the AKH negative neurons in HCG. Their observation about a pair of AKH negative neurons that mediate NPF signaling from the gut appears to contradict several previous observations (such as Yoshinari et al., 2021 and Malita et al., 2022). The authors may need to examine whether AKH+ neurons in HCG are NPFR+, and whether they also mediate NPF signaling from the gut.
4. The expression pattern of mGluRs should be determined. Are they expressed specifically in NPF+ EECs?
5. The authors showed that the frequency of the Ca⁺⁺ oscillation, rather than absolute Ca⁺⁺ concentration, is associated with the secretory capacity of EECs. In order to reliably evaluate the intensity of the calcium oscillation on the secretory capacity, a reference signal such as UAS-RFP should be introduced to the calcium imaging assay, and the GCaMP fluorescence intensity should be measured as normalized to the RFP fluorescence intensity.
6. Please show separate channels for Figures 2D-F and Figures 6b-c.
7. Extended Data Fig.1 vs. Extended Data Fig.3, knocking down NPF by tap1.3B-gal4 suppresses overeating and promotes energy wasting, but depleting the entire EECs only causes overeating. Could authors speculate on what causes the difference?

We would like to thank all the Reviewers for their kind words, complimenting the significance and technical quality of our work, and for their constructive suggestions for improvement. As a result of addressing these comments, we believe that our manuscript has improved significantly. Below we provide a point-by-point response (in blue) to the Reviewers' feedback.

REVIEWER COMMENTS

Reviewer #1 (Remarks to the Author):

Manuscript: Dietary L-Glu Sensing by Enteroendocrine Cells Adjusts Food Intake via Modulating Gut PYY/NPF Secretion

This manuscript by Gao et al examines a gut-brain pathway that regulates feeding via NPF neuropeptide. They identify a population of gut enteroendocrine cells (EECs) that can sense dietary L-Glutamate via a metabotropic glutamate receptor. This in turn influences NPF release from the EECs which then activates its receptor on dopaminergic enteric neurons to regulate food intake. Overall, this is a fantastic study that carefully dissects the entire pathway from sensory stimuli to behavioral output. It complements 2 recent studies on gut NPF signaling(<https://doi.org/10.1038/s41467-021-25146-w> and <https://doi.org/10.1038/s42255-022-00672-z>). One interesting finding of note is the link between calcium oscillations and peptide release.

The study is performed to a high-standard and the manuscript is well-written and easy to follow. I only have a few minor comments regarding the methods.

We thank the Reviewer for the positive evaluation of our work and for appreciating our finding that calcium oscillation is linked to peptide release in EECs.

Figure 2A - what about expression in the VNC? Please include these images.

We thank the Reviewer for reminding us. *tap^{1,3}-B-Gal4* is not expressed in the VNC. We have replaced the Figure 2a with a picture including the VNC.

Fig. 2a, Upper, expression pattern of *tap¹⁻³-B-Gal4>GFP* in central nervous system (CNS) and midgut. No GFP was observed in the CNS. Lower, schematic representation of the distribution of *tap¹⁻³-B-Gal4>GFP* expressing EECs.

Figure 3E: Could you please comment on the calcium levels with SERCA and PMCA RNAi? What is the cause of the large variability between different samples? Some of them have high calcium levels throughout and some have none.

This is a very interesting question. Because cytosolic Ca^{2+} is pumped into the ER by the sarco-endoplasmic reticulum calcium ATPase (SERCA) or out of the cell by the plasma membrane calcium ATPase (PMCA), knocking down SERCA or PMCA in the EEC will therefore make it difficult for Ca^{2+} in the cytoplasm to re-enter the ER or to leave the cell. Both manipulations were expected to keep cytosolic Ca^{2+} at a high level.

However, we did find that a significant proportion of EECs with SERCA or PMCA knockdown were in a state of no/low calcium activation during the 10 minutes of live imaging. We speculate that the following two possibilities may have contributed to this result. One is that knockdown of SERCA or PMCA in some EECs results in abnormal calcium signaling in these EECs, impairing their response to stimuli that elevate calcium in the cytoplasm. Another possibility is that high calcium levels in some EECs leads to the release of negative feedback factors that inhibit the activation of calcium signaling in neighboring EECs.

How different intestinal cell types respond to various signals to collectively adjust their behaviors to maintain intestinal homeostasis and host physiological fitness remains an intriguing topic. Pioneering work from the Jasper lab has established Ca^{2+} signaling as a key regulator of midgut stem cell activity, which was followed by studies to characterize Ca^{2+} signalling in other committed cell types (the Perrimon lab and the O'Brien lab). Still, we do not know exactly how Ca^{2+} signaling is modulated molecularly in different intestinal cell types. As in the case of EEC here, how Ca^{2+} signaling is initiated and how it is suppressed to produce constant oscillations was completely unknown. Our data now fill this gap by showing that L-Glu sensing slows down Ca^{2+} oscillations in EECs through mGluR to modulate the secretion of EECs. We expect that future work from our laboratory and others will provide further insight into Ca^{2+} oscillations in the EECs.

Line 457: cardiac should be cardiaca

We thank the Reviewer and have made this correction.

Line 654: add "in" after "expressed"

We thank the Reviewer and have made this correction.

Line 869: "Use fresh primary antibodies every time". Please correct the grammar

We thank the Reviewer and have made the corrections as "Fresh primary antibodies were used each time".

Lines 1037 to 1039: please correct the sentences.

We thank the reviewer. We have rewritten this method section of "Calcium imaging".

RT-PCR: Was genomic DNA removal step included? Or are the primers designed over exon-exon boundaries?

Yes, the total RNA was extracted using RNeasy Pure Tissue Kit (QIAGEN Biotech, Cat# DP431), which contains the DNAase step to remove genomic DNA. Primers were designed over exon-exon boundaries. The NPF forward primer was designed at the start of the first exon and the reverse primer was designed at the second exon. Primer sequences are indicated in Supplementary Table 2.

Calcium imaging: what was the reason for making the buffer with insulin and FBS instead of using hemolymph like saline? At what temperature was the experiment performed? RT or 40deg?

We thank the reviewer for asking. In our earliest attempts to study the *Drosophila* midgut using live imaging, we used the formulation of medium developed by Denise J Montell's lab for culturing egg chambers for live imaging¹. Montell's lab developed Schneider's medium, which contains insulin and FBS, allows time-lapse imaging for 6 hours and supports egg chamber growth, oocyte growth and cytoplasmic streaming, in addition to border cell migration.

Using Schneider's medium containing insulin and FBS, we did not observe significant changes in the appearance of the midgut, peristaltic frequency and morphology of the *esg*⁺ cells during the 2 hours of in vitro culture and performing live imaging. We also observed the mitotic divisions of ISCs and the oscillation of calcium signals in ISCs, in agreement with previous reports^{2, 3}. In addition, we observed the asymmetric division of pupal ISCs during the pupal stage using this medium⁴. Therefore, in this study, we continued to use this formula for our study of calcium oscillations in EECs. It may work equally well, but we

haven't tried the hemolymph like saline yet.

The time-lapse imaging experiments were performed at room temperature (25°C). We have clarified these experimental details in the modified Method section.

Sex of the flies used: I assume that females were used for all experiments. Were they mated or virgin? Please include this information.

We thank the Reviewer for pointing this out. We used mated female flies. We have included this information in the main text and in the Methods section.

1. Prasad, M., Jang, A.C.C., Starz-Gaiano, M., Melani, M. & Montell, D.J. A protocol for culturing *Drosophila melanogaster* stage 9 egg chambers for live imaging. *Nature protocols* **2**, 2467-2473 (2007).
2. Deng, H., Gerencser, A.A. & Jasper, H. Signal integration by Ca(2+) regulates intestinal stem-cell activity. *Nature* **528**, 212-217 (2015).
3. Martin, J.L. *et al.* Long-term live imaging of the *Drosophila* adult midgut reveals real-time dynamics of division, differentiation and loss. *eLife* **7** (2018).
4. Wu, S. *et al.* Apical-basal polarity precisely determines intestinal stem cell number by regulating Prospero threshold. *Cell reports* **42**, 112093 (2023).

Reviewer #2 (Remarks to the Author):

Title: Dietary L-Glu sensing by EEC adjusts food intake via modulating gut PYY/NPF secretion.

Proper sensing of dietary protein intake is believed to act as an essential internal system for the maintenance of physiological homeostasis of an animal. In this study, Gao et al. analyzed the role of enteroendocrine cells (EECs) in protein sensing by taking advantages of genetic tools available in *Drosophila* model system. They showed that (1) Specific elimination of EECs by generating *esg-p>scute-RNAi* flies and *pros-TCD>hid* flies led to a significant increase of food intake, indicating that EECs function to inhibit food intake; (2) By analyzing microbiome, they showed no significant differences between control flies and EEC-less flies; (3) By developing EEC-specific *gal4* (*tap1.3B-gal4*), they showed that neuropeptide secreted from *tap1.3B*-positive EECs inhibits food intake; (4) When synthetic NPF peptide was injected into the body cavity, increased food appetite seen in *tap1.3B>NPF-RNAi* flies or EEC-less flies (*esg-p>scute-RNAi* flies) was abolished; (5) protein-rich diets or two specific amino acids (Glu and Asn) enhance the intensity of NPF staining in EECs (i.e. reducing NPF secretion by promoting NPF retention in EECs); (6) Glu sensing attenuates calcium oscillation in EECs to reduce NPF secretion; (7) EECs sense Glu through mGluR; (8) Glu inhibits gut NPF release, and thus inactivates two neurons (using *NPFR-ENS-gal4*) located in HCG; (9) *NPFR-ENS*-positive neurons use dopamine to control food intake. Taken all together, they concluded that dietary Glu sensing by EECs adjusts food intake via modulating gut NPF secretion and subsequent the activity of *NPFR*-positive enteric neurons.

General comments.

Although I appreciate technical quality of the experiments, there are problems in the logic of the experimental setting and data interpretation. More experiments are needed to support their claims. At present, at least in a current form of the manuscript, I do not agree the conclusion of the study.

1. The key finding that the gut-derived NPF suppressed sugar intake (since the authors measured sugar intake in their assay) has been already shown by Kim Rewitz group (Nature Metabolism 2022). The novelty has been undercut.

We thank the Reviewer for raising this point, but we do not consider that Rewitz's paper undermines the novelty of our findings due to the following reasons. First, the scientific questions of the two studies are rather different. Our focus is the modulation of EEC function at the mechanistic level, and the strength of our work lies at revealing a previously unrecognized fundamental principle of how the EEC regulates its secretion through the Calcium signaling in response to nutrient cues. Our finding that the EEC senses L-Glu in food and regulates neuropeptide release by altering the frequency of calcium oscillations is novel and revealing for experts in EEC biology and gut physiology. Furthermore, the circuit downstream of NPF release that we have characterized is different from the report from the Rewitz lab. In contrast to the orexigenic function of NPF in the brain, the EEC-

derived NPF suppresses appetite by activating a pair of dopaminergic enteric neurons that stain negative for AKH, while the post-mating shift in dietary preference reported by the Nature Metabolism paper appears to act exclusively via AKH cells. Lastly, as suggested by this Reviewer in the following comments, we examined the effect of gut-derived NPF on feeding using a variety of diets. Consistent with our claims, EEC-derived NPF suppressed the overall appetite rather than just the sugar intake. Rewitz's paper did report that NPF derived from the EECs suppresses sugar intake. However, instead of working on the mechanism by which gut NPF suppresses sugar intake, they turned to the mechanism by which mating induces sugar satiety and increase consumption of protein-rich food. Therefore, our paper and theirs both go deep but to quite different directions with a clear distinction of downstream circuits, despite the fact that our study and theirs are consistent with each other in term of the role of gut-derived NPF in feeding control.

2. The finding that a non-essential amino acid (L-glu) promotes food intake is not consistent with other previous literatures including mammalian ones, which showed that protein consumption suppresses food intake.

We would like to clarify that protein foods do not always suppress appetite. The effect of protein consumption on food intake is highly context-dependent and may differ depending on the exact design of the feeding experiment, including the duration of the participant's fasting period prior to the start of the feeding, the pre-treatment, the duration of the feeding period, the type of food, and other factors.

For example, in an experiment in which a high-protein diet inhibited food intake, flies were first fasted for 24 hours and then, after being pre-fed with tryptone, food intake was measured over the next 10 minutes¹. In another study of amino acid and protein suppression of food intake, flies were similarly measured for feeding levels within 10 minutes after 24 hours of fasting². In our experiment, however, flies were only fasted for 3 hours, or treated without fasting period (dye-based food intake measurement), after which we measured the food intake of flies over a 24-hour period. It is noteworthy that dietary amino acids, as perceived "umami" flavors, in fact increase acute feeding levels in both *Drosophila*^{3, 4} and mammals⁵. Therefore, we propose that whether protein consumption suppresses food intake depends on animal's demand for protein, and such demand can change over time along with feeding.

The authors measured food intake with a food containing only sugar (5% sucrose), not any other macronutrients. For the experiments of food intake, they should examine other types of foods including a standard cornmeal diet (SCD), protein-rich diet (also food containing essential amino acids or non-essential amino acids or both), carbohydrate-rich diet and fat-rich diets.

Thanks to the Reviewer for this suggestion. We have repeated food intake experiments in 3-5 d AE mated females using dye-based food intake measurement after blocking EEC generation in the pupal stage (*esg^P>sc RNAi*, Fig. 1e, 3l) and knockdown of NPF (*tap¹⁻³⁻*

B-Gal4>NPF RNAi, Fig. 2k, 3k). In these experiments we examined the food intake on standard cornmeal diet (SCD), carbohydrate-rich diet (SCD+10% sucrose), fat-rich diet (SCD+25% coconut oil) and protein-rich diet (SCD+10% yeast). Our results show that EECs and NPF they secreted not only suppress sucrose intake but also restrict the overall appetite (Fig. 1e, 2k, 3k and 3l). Consistent with the conclusion that protein diets inhibit NPF release, knockdown of NPF no longer increased the feeding of a high protein diet (Fig. 3k).

Fig. 1e, Food consumption of *control* lines and *esg^Δ>scute^{RNAi}* flies at 3 d AE measured using the dye-based food intake measurement. $n \geq 11$ in each genotype.

Fig. 3l, High-sugar (standard cornmeal food + 10% sucrose), high-fat (standard cornmeal food + 25% coconut oil) and high-protein (standard cornmeal food + 10% yeast) food consumption of *control* and *esg^Δ>scute^{RNAi}* flies at 3 d AE measured using the dye-based food intake measurement. $n \geq 12$ in each genotype.

Fig. 2k, Food consumption of *control* and *tap^{1.3}-B-Gal4>NPF^{RNAi}* flies measured using the dye-based food intake measurement. $n \geq 13$ in each genotype.

Fig. 3k, High-sugar (standard cornmeal food + 10% sucrose), high-fat (standard cornmeal food + 25% coconut oil) and high-protein (standard cornmeal food + 10% yeast) food consumption of *control* and *tap^{1.3}-B-Gal4>NPF^{RNAi}* flies measured using the dye-based food intake measurement. $n \geq 11$ in each genotype.

3. The relationship between L-glu and DH44 neurons is very tenuous at best. In the experiments found in previous publications, they used sucrose solution containing amino acid mixture for food intake assay. For example, Kim and Kanai et al demonstrated that DH44 neurons are dispensable for amino acids or essential amino acids sensing. The authors should take the part out from the manuscript.

We would like to clarify that Yang et al. published a paper showing that six brain neurons expressing DH44 could be rapidly and directly activated by L-Glu⁴. They show that a putative amino acid transporter, CG13248, is enriched in DH44+ neurons and is required for L-Glu sensing. Kim and Kanai's paper⁶ actually cited the Yang et al. paper as "Dh44-expressing neurons in the brain ... have been previously shown to respond to few amino acids". The results in Kim and Kanai's paper suggest that DH44+ neurons are not involved in the compensatory preference for L-EAAs in protein-deprived flies (a specific context), but it did not directly re-assess the role of DH44 neurons in AA sensing. We think it would be appropriate to keep the description of the perception of L-Glu by DH44 neurons in the manuscript.

4. The amount of single non-essential amino acids such as L-glu may not be abundant in the proximal R2 region of the gut. Dietary proteins get digested and degraded in the R3 compartment where acidic digestive enzymes are profusely available. Before R3, most of the dietary proteins may be present as peptides, not as single amino acids. So it begs a question as to why NPF would be regulated in the proximal R2 by L-glu.

Thanks to the Reviewer for this interesting question. First, the L-Glu in dietary protein may already be accessible at the boundary of R2/R3. Secondly, even if L-Glu is released onward from the R3, based on our unpublished observations and the literature on insect larval gut studies, peristalsis in the insect gut is not unidirectional but bidirectional: intestinal peristaltic waves periodically reverse anterograde and retrograde directions⁷. It is possible that R3 digested amino acid could reach to the proximal R2 region under the retrograde peristalsis, and could explain why NPF is regulated in the proximal R2 by dietary protein.

In this context, it would be interesting to see whether NPF secretion in R2-located EECs is differently regulated by Glu ingestion, when compared with NPF secretion in R3/4-located EECs.

Following the suggestion of the Reviewer, we examined NPF staining in different regions of the midgut. As shown below (as Extended Data Fig. 5f), in the proximal R2 and R3, NPF staining is significantly increased after 1% L-Glu feeding, whereas in regions of R2 that don't have NPF expression under cornmeal feeding, NPF staining was not changed upon 1% L-Glu feeding. L-Glu feeding excludes the digestion of proteins in different intestinal compartments, so we conclude that the reason why L-Glu does not affect the secretion of NPF at the anterior end of R2 is not because only R3 is able to digest proteins, but rather because different regions of the EEC express neuropeptides differently and respond

differently to different nutrients.

5. The authors used the CAFE assay to measure the feeding amount of five flies, using 5% sucrose as the food source. However, the average feeding amounts of the control flies in each figure are diverse, even at the same temperature.

We would like to point out that our experiments had to be carried out at different temperatures for various genetic tools. As temperature has a significant effect on the fly's food intake, the control group's food intake varied from experiment to experiment.

For example, in Figure 1d, the control flies' feeding amount is around 1 microL daily, but in Figure 1g, the control flies eat around 2.5~3 microL daily at the same temperature.

The experiments in Figure 1d were performed at 18°C and the experiments in Figure 1h were performed at 30°C. This is the reason why the control group ate different amounts in these two sets of experiments.

In Figure 1d, the *esg^P* method was used to deplete EECs during the pupal stage, where the temperature-inducible ISC driver *esg-Gal4 tub-Gal80^{ts} UAS-GFP (esg^{ts})* was used to deplete sc for 10 h at 30°C in the pupal stage, then flies were transferred to 18°C to avoid continuous sc knockdown. In this scenario, EECs were gradually generated by adult ISCs, allowing us to compare the physiology consequence of “less than 10 EECs (3 days After Eclosion)”, “~100 EECs (7 days AE)” and “~500 EECs (10 days AE)”. Therefore, the food intake experiments were performed at 18°C at 3/7/10 days AE.

In Figure 1h, to prevent the EEC regeneration during the adult stage, we placed *esg^P>scute^{RNAi}* adults at 30°C upon eclosion and we termed this method as *esg^{P+A}*. Therefore, the food intake experiments were performed at 30°C at 2/4/6/10 days AE.

Similarly, in Figure 5a, the control flies consumed around 5 microL in a day.

In Figure 5a, we used the 5% sucrose+1% L-Glu as the food for the CAFÉ assay. As noted above, in our experimental design, 1% L-Glu increases the level of food intake.

It is possible that the authors mixed the male and female feeding data, and they need to

clarify this point. Also, the different feeding amounts may indicate that the genetic background of the flies in each figure is not identical. Additionally, the authors did not clearly indicate whether they used male or female flies in some figures.

We used 3-5 d mated female flies in all experiments. For clarity, we emphasized the mating status and sex identity in the Results and Methods of the revised manuscript.

As the authors measured the feeding assay in a group of five flies, they cannot rule out the possible social interactions in the group of flies.

In addition to the CAFE assay, our claims are also supported by the Manual Feeding (MAFE) assay to depict details of the feeding behavior of individual flies (Fig. 2i, j and Supplementary movie 1).

Fig. 2i, j, Feeding time (i) and food intake (j) of control and *tap^{1.3}-B-Gal4*>*NPP^{RNAi}* flies measured using the MAFE assay.

6. In vivo function of EECs is unclear. They showed that EEC-less flies (*esg-p>scute-RNAi* flies and *pros-TCD>hid* flies) showed similar metabolic phenotypes when compared to control flies. How do they maintain those flies?

We maintained the EE-less flies on a standard cornmeal diet.

For the experiment, do they use flies reared on a standard cornmeal diet (SCD)?

Yes.

If this is the case, do EEC-less flies increase intake of a SCD?

As shown above, EEC-less flies increased the food intake on a standard cornmeal diet (Figure 1e).

Fig. 1e, Food consumption of *control* lines and *esg^p>scute^{RNAi}* flies at 3 d AE measured using the dye-based food intake measurement. $n \geq 11$ in each genotype.

If elimination of EECs led to an increase of SCD intake, while showing metabolic phenotypes similar to those of the control flies, what is the in vivo function of EECs?

We understand and appreciate the Reviewer's question. Our current understanding is that EECs, as a population of cells, have very complex and diverse functions in food digestion, energy sensing and metabolic regulation. And it is worth noting that EECs can regulate opposite metabolic processes by secreting different neuropeptides depending on the physiological environment. For example, in the data from ourselves (Extended Data Fig. 4 k-p), Rewitz lab⁸ and Niwa lab⁹, EECs promote lipid anabolism in adipose tissue by secreting NPF. However, in addition to NPF, EECs also secrete other neuropeptides such as AstA and AstC, both of which have been reported to promote the release of the adipokinetic hormone (Akh), thereby mobilizing systemic energy reserves^{10, 11}. We therefore speculate that although EEC-less flies lost the ability to respond to changes in feeding conditions, they may still exhibit a similar level of metabolic phenotypes to those of the control flies under laboratory SCD rearing conditions.

In addition, it has been reported that EEC-less flies have a substantial reduction in intestinal digestive enzyme activities, including trypsin, chymotrypsin, aminopeptidase and acetate esterase¹². Reduced digestive capacity is consistent with the fact that EEC-less flies eat more but do not show significant changes in metabolism.

Extended Data Fig. 4 **k-n**, Mass (**k**), protein (**l**), glucose (**m**) and TAG content (**n**) of *control* and *tap^{1.3-B-Gal4>}**NPF^{RNAi}* flies. Each genotype corresponded to 12 (**k**), 9 (**l-n**) samples of 20 (**k**), 5 (**l**), 20 (**m**) and 10 (**n**) flies each. **o,p**, Oil Red O staining of *control* (**o**) and *tap^{1.3-B-Gal4>}**NPF^{RNAi}* (**p**) midguts.

7. The involvement of microbiome is unclear. In their analyses, it is very strange to see the Bacteroidetes in the fly microbiome (Normally, drosophila microbiome does not contain Bacteroidetes). Do they have some explanation about this?

Thanks to the Reviewer for the question. This is a very interesting point. We searched the microbiome of 7 d AE and 10 d AE flies in our lab data and also found no Bacteroidetes. We agree with the reviewer's speculation in the next paragraph that the microbiome we sequenced was the immature microbiome in the early adult midgut.

They showed no significant differences between control flies and EEC-less flies. However, they analyzed the microbiome only at a single time point (at 3d AE, i.e. 3 days after eclosion). Microbiome of the young flies (such as those at 3d AE) is very simple and immature. Therefore, to see the cross relationship between microbiome and EEC-less condition (or phenotype), they should also analyze the microbiome of older flies (e.g. 10 d AE).

We sequenced the microbiome only at 3d AE on purpose. The EEC-less fly generated by our *esg^P* method consumed significantly more food than the control fly at 3 d AE. At 10 d AE, ~500 EECs are regenerated from the adult intestinal stem cells, and at this time point the fly's feeding level was no longer significantly different from the control (Figure 1d). To see whether the difference in feeding levels at 3 d AE was due to differences in microbiome, we sequenced the microbiome of flies of 3 d AE.

By 10 d AE, EEC-less flies and control flies had already consumed the same amount of food, so it is not possible that microbiome plays a role here in regulating food intake, even if the microbiome differs between the two groups.

Furthermore, they should examine the difference in terms of food intake between germ-free EEC-less flies and conventional EEC-less flies.

Thanks to the Reviewer for this suggestion. We examined sugar food intake by the CAFÉ assay and SCD food intake by the dye-based method between control flies (*esg^P>attp empty*) and EEC-less flies (*esg^P>sc^{RNAi}*) reared in conventional and germ-free conditions 3 d AE (Extended Data Fig. 2j, k). Our results show that regardless of microbiome status, EEC-less flies always consumed more food than control flies.

Extended Data Fig. 2j, Food intake of *control* and *esg^P>scute^{RNAi}* flies at 3 d AE under conventional fed and germ free conditions. $n \geq 17$ in each genotype.
k, Food consumption of *control* and *esg^P>scute^{RNAi}* flies at 3 d AE under conventional fed and germ free conditions measured using the dye-based food intake measurement. $n \geq 5$ in each genotype.

8. Similar to the above point-5, do *tap1.3B>NPF-RNAi* flies show increased intake of a SCD?

As we have answered this point above, yes.

9. There is a difference (when compared to control flies) in terms of glucose content, TAG, and oil red O staining between *tap1.3B>NPF-RNAi* flies and EEC-less flies (*esg-p>scute-RNAi* flies). Why?

This is basically the same question as question #6 (why EEC-specific knockdown of NPF has different metabolic consequences than EEC-less flies) that we have addressed.

In addition, Rewitz and Niwa labs have investigated the metabolic function of NPF^{8,9}. We agree with their interpretation that NPF regulates lipid metabolism through glucagon-like and insulin-like hormones. We have added this explanation in the main text and cited their publications.

10. It is important to see whether brain NPF expression level and patterns are altered in

tap1.3B>NPF-RNAi flies or EEC-less flies (*esg-p>scute-RNAi* flies). Similarly, are brain NPF expression levels and patterns altered in response to protein-rich (or Glu) diet?

As suggested by the Reviewer, we examined the immunostaining and transcript levels of NPF in the brain. We found that neither *tap^{1.3}-B-Gal4>NPF RNAi* (Extended Data Fig. 4a-d) nor *esg^P>sc RNAi* (Extended Data Fig. 4e-h) affected NPF expression in the brain.

Extended Data Fig. 4 a-c, Representative images (a,b) and quantification (c) of NPF staining in brains of control (a) and *tap^{1.3}-B-Gal4>NPF^{RNAi}* (b) flies. n=30 in each group. d, Normalized brain NPF mRNA levels of control and *tap^{1.3}-B-Gal4>NPF^{RNAi}* flies by RT-qPCR. Each genotype corresponded to 3 biological replicates of 150 brains each. e-g, Representative images (e,f) and quantification (g) of NPF staining in brains of control (e) and *esg^P>scute^{RNAi}* (f) flies at 3 d AE. n=30 in each group. h, Normalized brain NPF mRNA levels of control and *esg^P>scute^{RNAi}* flies at 3 d AE by RT-qPCR. Each genotype corresponded to 3 biological replicates of 150 brains each.

We also examined the effects of different diets, including cornmeal food, yeast paste, 5% yeast extract and 1% L-Glu, on NPF immunostaining and transcript levels in the brain. No changes were found (Extended Data Fig. 6a-c).

11. It is important to see whether brain NPFR signaling is activated by synthetic NPF injection into the body cavity. Similarly, is brain NPFR signaling activated in response to protein-rich (or Glu) diet?

To visualize NPFR signaling in the brain, CaLexA was driven by *NPFR^{RAC}-Gal4* and *NPFR^{RB/D}-Gal4* (two Gal4 KI lines) feeding on cornmeal food, the cornmeal food with the addition of 1% L-Glu and upon being injected with synthetic NPF into the body cavity. Our results showed that the NPFR neuronal activities were not altered by L-Glu feeding or synthetic NPF injection (Response Figure 1).

Response Fig. 1 | L-Glu feeding and NPF supplement has no effect on NPFR signaling in the brain.
a-d, Representative images (a-c) and quantification (d) of CaLexA intensity of region of interest (ROI) in

the brains of *NPFR^{RA/C}-Gal4>CaLexA* flies under cornmeal food (a), 1% L-Glu feeding (b) and NPF injection (c) conditions. n=30 in each group (d). The area framed by the dotted line in the Response Fig.1a is an example of an ROI. e-h, Representative images (e-g) and quantification (h) of CaLexA intensity of ROI in brains of *NPFR^{RB/D}-Gal4>CaLexA* flies under cornmeal food (e), 1% L-Glu feeding (f) and NPF injection (g) conditions. n=30 in each group (h). The area framed by the dotted line in the Response Fig.1e is an example of an ROI. Data are represented as mean ± SD. Significance was determined using Student's *t* test (d,h). n.s., non-significant. n, number of brains (d,h). Scale bars, 100µm.

12. They claimed that the orexigenic effect of brain NPF override the role of gut NPF in restricting appetite. To prove this issue, they should introduce NPF expression in NPF mutant animals in a tissue-specific manner. For this, they should examine food intake level (sucrose as well as SCD) by using NPF mutant flies expressing gut NPF (using *tap1.3B-gal4>UAS-NPF*) or brain NPF (brain-specific *gal4>UAS-NPF*).

We thank the Reviewer for this suggestion. We compared the food intake levels (sucrose by CAFÉ assay and SCD by dye-based absorbance) between NPF heterozygous mutant (*NPF^{1/+}*), NPF homozygous mutant (*NPF¹*) and gut-specific re-supply of NPF under the NPF mutant background (*tap^{1.3}-B-Gal4>NPF, NPF¹*). Our results show that gut-derived NPF not only failed to rescue the reduced food intake caused by the NPF mutation, but also further suppressed food intake (Figure 2p, q). These results support that NPF secreted by the brain and gut play opposing roles in appetite regulation, and that NPF secreted by the gut cannot replace the function of NPF secreted by the brain.

Fig. 2p and q, CAFE assay (p) and dye-based food intake measurement (q) of heterozygous (*NPF^{1/+}*), homozygous *NPF* mutant (*NPF¹*) and EECs-specific NPF recovery under *NPF* mutant condition (*tap^{1.3}-B-Gal4>NPF, NPF¹*) flies. n≥18 (p) and n≥13 (q) in each genotype. Data are represented as mean ± SD. Significance was determined using Student's *t* test. *p < 0.05, **p < 0.01, **** p < 0.0001. n, number of groups performed for quantification of food intake (5 flies in each group) (p), or number of groups (20 flies in each group) performed for quantification of food consumption (q).

13. Based on the results in Fig3a-b, they concluded that “this implied a role for EECs in AA sensing”. However, these data in fact indicate that expression of gut NPF is induced by

protein-rich diet, but not carbohydrate-rich or fat-rich diet. And this just implied a possible role for gut NPF in protein or AA-sensing.

We agree with the Reviewer and made the following modification.

Old: "This implied a role for EECs in AA-sensing".

New: "This implied a role for gut NPF in AA-sensing".

14. It would be interesting to see the NPF expression level between germ-free and conventional flies.

We thank the Reviewer for this suggestion. We compared the NPF immunostaining and transcript levels between germ-free and conventional flies. We found no significant difference between these two rearing conditions (Response Figure 2).

Response Fig. 2 | The transcriptional and staining level of NPF in EECs is not affected in germ-free flies.

a-c, Representative images (**a,b**) and quantification (**c**) of NPF staining under conventional fed (**a**) or germ-free (**b**) conditions. $n=75$ in each group. n , number of EECs. **d**, Normalized *NPF* mRNA levels under conventional fed and germ-free conditions by RT-qPCR. Each genotype corresponded to 3 biological replicates of 50 midguts each. Data are represented as mean \pm SD. Significance was determined using Student's *t* test (**c,d**). n.s., non-significant. Scale bars, 20 μ m.

15. For the figure 3C, they used only 1% of each amino acid. Therefore, with this experimental setting, it would be impossible to conclude that NPF is regulated by only Glu and Asn. As each type of food or protein has unique amino acid content, the assay should be performed in a dose-dependent manner using increasing amounts of each amino acid.

To address the Reviewers' concerns, we repeated the NPF immunostaining experiments in EECs using saturating concentrations of various amino acids, respectively. The results

of our experiments confirmed that amino acids other than L-Glu and L-Asn did not lead to elevated NPF immunostaining (Extended Data Fig. 5 b, c).

16. They should check whether *tap1.3B-gal4* expression (by using *tap1.3B-gal4>uas-gfp*) is altered in response to protein-rich (or Glu) diet.

We thank the Reviewer for this suggestion. We found that *tap^{1.3}-B-Gal4* driven GFP expression was not significantly altered in the midgut when fed a protein-rich or 1% L-Glu diet (Extended Data Fig. 6d).

17. Are there any cell bodies of NPFR-ENS-positive neurons in the brain? In the fig 7a, it would be possible that cell bodies of NPFR-ENS-positive neurons are also seen in the brain. This should be checked.

To address the Reviewers' concerns, we recombined a UAS nuclear localized *Redstinger* with *NPFR^{ENS}-Gal4* to visualize the neuron cell body (In fact, it's the nucleus). As shown below, only a pair of NPFR^{ENS}-positive neurons were seen close to the cardia of the midgut, but there were no cell bodies in the brain (Extended Data Fig. 10e).

Extended Data Fig. 10e The expression pattern of *NPFR^{ENS}-Gal4>mCD8:GFP+Redstinger*. Redstinger shows the location of cell bodies of *NPFR^{ENS}-Gal4⁺* neurons.

18. Do Axons from cell bodies of NPFR-ENS-positive neurons located in HCG actually reach SEZ of brain?

To address the Reviewers' concerns, we crossed the *NPFR^{ENS}-Gal4* with a strong UAS-GFP reporter (*20xUAS-6xGFP, 120xGFP*) to visualizing the ascending projection from the cell bodies of *NPFR^{ENS}*-positive neurons. As shown below, the axon indeed reached the SEZ of the brain (Extended Data Fig. 10f).

Extended Data Fig. 10f, The expression pattern of *NPFR^{ENS}-Gal4>120xGFP*. The yellow arrow head shows the axon of *NPFR^{ENS}-Gal4⁺* neurons reaches SEZ of brain.

19. The authors ruled out L-Asn as a candidate without conducting any experiments. However, Figure 3c and 3d shows that L-Asn successfully prevents the secretion of NPF in EEC. Therefore, they cannot rule out the possibility that L-Glu and L-Asn work together to suppress the secretion of NPF. The authors should investigate the potential role of L-Asn in this story to provide a complete understanding. For example, it is important to see whether gut NPF secretion and the activity of enteric NPFR-positive neuron are also modulated by Asn ingestion.

We thank the Reviewer for this suggestion. We measured the activity of *NPFR^{ENS}*-positive neurons of flies (*NPFR^{ENS}-Gal4>CaLexA*) fed with 1% Asn. Our data show that Asn feeding

significantly attenuated the activity of $NPFR^{ENS}$ neurons (Response Figure 3), suggesting that Asn feeding also reduced NPF secretion in EECs.

Response Fig. 3 | L-Asn feeding decreases the CaLexA intensity in $NPFR^{ENS}$ neurons.

Upon cornmeal food (a) or 1% L-Asn (b) feeding conditions, representative images (a,b) and quantification (c) of CaLexA intensity in $NPFR^{ENS}$ neurons. n, number of $NPFR^{ENS}$ neurons. $n \geq 18$ in each group. Data are represented as mean \pm SD. Significance was determined using Student's *t* test (c). **** $p < 0.0001$. Scale bars, 20 μ m.

20. The authors developed a well-controlled temperature control device. However, this technique may not be necessary to use in this paper because previous studies and authors have already established better ways of manipulating gut-EEC specifically. While I agree that this technique is useful, it may be slightly out of focus for this story and not crucial for the paper's main message.

As we mentioned in the manuscript (line 122-130), stocks/methods used in previous studies were problematic. *Tachykinin(gut)-Gal4 (Tkg-Gal4)* was reported to be specific to TK^+ EECs¹³, and has been used in a number of intestinal studies^{9, 12, 14-17}. However, *Tkg-Gal4* was later found to drive substantial expression in brain^{14, 18}. In addition, a Gal80 transgene driven by an enhancer fragment (*R57C10*) of neuronal *Synaptobrevin (nSyb)* is often used in combination with Gal4 drivers to suppress Gal4 transcriptional activity in CNS, with the expectation that only EECs are manipulated^{8, 11}. However, the *R57C10* fragment is also active in EECs^{19, 20}.

Therefore, our temperature control device (TCD) physically enables gut-specific manipulation, allowing us to rigorously study gut and brain function separately. As this Reviewer also appreciate that this technique is useful, we believe TCD should provide new ideas and means of implementation for similar questions in the field.

21. The authors identified two EEC-secreted hormones, NPF and Tk, as potential regulators of feeding behavior. They ruled out Tk as a possible EEC-derived feeding regulator; however, based on their data, they should consider increasing the trial number and re-examining this point. For instance, in Extended Data Figure 3d, knocking down Tk using *tap1.3B-Gal4* seems to increase food intake. While they showed that the increase in food intake amount is not significant, if they increase the trial number, it may result in significant food overconsumption. Every figure they showed about Tk exhibits a similar pattern with NPF manipulation. Therefore, they should increase the trial number and recheck this point to confirm their findings.

To address the Reviewers' concerns, we increased the sample size from 17 to 35-37. The additional data further confirmed that EEC-derived Tk is not involved in the regulation of feeding behavior (Extended Data Fig. 3j, l).

22. The authors argued that EEC only secretes NPF after activation, but not Tk. If both NPF and TK-containing vesicles exist in EEC, they should explain how only NPF-containing vesicles are secreted, but not those of TK. Although there are several possibilities that only one vesicle can be secreted while the whole cell is activated, it is difficult to understand how one specific vesicle type is secreted after the whole cell activation.

Thanks to the Reviewer for asking this fundamental question probably for the whole area of neuroscience. We further examined the staining of Tk in EECs on high protein/1% L-Glu diets and found that, unlike NPF, there was no significant change in the intensity of immunostaining for TK (Response Figure 4). With such data, we can now only speculate

that 1) NPF and TK are packed into different secretory vesicles, or 2) the vesicles carrying TK are not sensitive to calcium oscillations. Either way, we currently have no evidence to support our speculations. We feel that the regulation of neuropeptide release from the EECs (and from neurons) is extremely complicated.

Response Fig. 4 | High L-Glu feeding has no effect on Tk secretion from EECs.

Upon cornmeal food (a), 5% yeast extract (b), yeast paste (c) and 1% L-Asn (d) feeding conditions, representative images (a-d) and quantification (e) of Tk staining in EECs. n, number of EECs. n=75 in each group. Data are represented as mean \pm SD. Significance was determined using Student's *t* test (e). n.s., non-significant. Scale bars, 20 μ m.

23. To clearly confirm their hypothesis, it would be essential to measure the circulating amount of NPF before and after L-Glu treatment. Western blotting would not be an ideal method to measure circulating hormone levels. The field commonly uses ELISA and mass spectrometry to measure hormone levels. Therefore, to confirm their hypothesis, the authors should measure the changes in the circulating NPF levels using appropriate methods.

We are also very interested in detecting changes in circulating NPF. We have been trying to achieve this aim for over a year, but as reported by the Niwa and Rewitz labs, the antibody to NPF does not work in Western blot. We have also tried ELISA and dot blot assays, and neither have been able to reproducibly detect signal changes in the hemolymph. We have also tried mass spectrometry to measure circulating NPF levels, but mass spectrometry requires at least 200 μ l of hemolymph per sample (BioTree Inc.). We collected hemolymph from 1000 7-day-old adult mated female flies and still could not get 200 μ l. Therefore, the current detection of circulating NPF levels in adult flies is unfortunately beyond technical capabilities. However, we believe that our genetic dissection of NPF function is thorough and supports a change of NPF secretion upon L-Glu treatment.

24. The previous study (Malita et al., 2006) already clearly showed the notion that "gut-

NPF suppresses food intake." Although the previous study showed the critical role of the mating status of a female in regulating protein feeding in flies, the authors did not mention the sexual dimorphism and post-mating behavioral switch in females that significantly affect food intake. Therefore, the authors should include this information to provide a comprehensive view of the regulation of feeding behavior.

We thank the Reviewer for pointing this out. We used mated female flies in all experiments. We have included this information in the main text and in the Methods section.

25. In the discussion section, the authors paralleled PYY in mammals and gut-NPF in flies. However, because of the multiple discrepancies between these hormones, it is difficult to accept their suggestion. Therefore, the authors should provide a more detailed comparison of the two hormones to strengthen their argument.

We would like to clarify that the comparisons between the mammalian neuropeptide Y family and the *Drosophila* NPF are not our ideas and innovations. Ping Shen's lab published this comparison almost 25 years ago²¹, and his lab has been dedicated to this comparison for over two decades²²⁻²⁴. As we have already cited their literature in the discussion section, we will not repeat the detailed comparison between NPY and NPF.

1. Sun, J. *et al.* *Drosophila* FIT is a protein-specific satiety hormone essential for feeding control. *Nature communications* **8**, 14161 (2017).
2. Song, T.T. *et al.* Dietary cysteine drives body fat loss via FMRamide signaling in *Drosophila* and mouse. *Cell research* **33**, 434-447 (2023).
3. Ganguly, A. *et al.* A Molecular and Cellular Context-Dependent Role for Ir76b in Detection of Amino Acid Taste. *Cell reports* **18**, 737-750 (2017).
4. Yang, Z. *et al.* A post-ingestive amino acid sensor promotes food consumption in *Drosophila*. *Cell research* **28**, 1013-1025 (2018).
5. Nelson, G. *et al.* An amino-acid taste receptor. *Nature* **416**, 199-202 (2002).
6. Kim, B. *et al.* Response of the microbiome–gut–brain axis in *Drosophila* to amino acid deficit. *Nature* (2021).
7. Slama, K. & Lukas, J. Myogenic nature of insect heartbeat and intestinal peristalsis, revealed by neuromuscular paralysis caused by the sting of a braconid wasp. *Journal of insect physiology* **57**, 251-259 (2011).
8. Malita, A. *et al.* A gut-derived hormone suppresses sugar appetite and regulates food choice in *Drosophila*. *Nature metabolism* (2022).
9. Yoshinari, Y. *et al.* The sugar-responsive enteroendocrine neuropeptide F regulates lipid metabolism through glucagon-like and insulin-like hormones in *Drosophila melanogaster*. *Nature communications* **12**, 4818 (2021).
10. Li, Y.G. *et al.* Gut AstA mediates sleep deprivation-induced energy wasting in *Drosophila*. *Cell Discov* **9** (2023).
11. Kubrak, O. *et al.* The gut hormone Allatostatin C/Somatostatin regulates food intake and metabolic homeostasis under nutrient stress. *Nature communications* **13**, 692 (2022).
12. Amcheslavsky, A. *et al.* Enteroendocrine cells support intestinal stem-cell-mediated

- homeostasis in *Drosophila*. *Cell reports* **9**, 32-39 (2014).
13. Song, W., Veenstra, J.A. & Perrimon, N. Control of lipid metabolism by tachykinin in *Drosophila*. *Cell reports* **9**, 40-47 (2014).
 14. Ameku, T. *et al.* Midgut-derived neuropeptide F controls germline stem cell proliferation in a mating-dependent manner. *PLoS biology* **16**, e2005004 (2018).
 15. Hadjieconomou, D. *et al.* Enteric neurons increase maternal food intake during reproduction. *Nature* **587**, 455-459 (2020).
 16. Kamareddine, L., Robins, W.P., Berkey, C.D., Mekalanos, J.J. & Watnick, P.I. The *Drosophila* Immune Deficiency Pathway Modulates Enteroendocrine Function and Host Metabolism. *Cell metabolism* **28**, 449-462 e445 (2018).
 17. Song, W. *et al.* Midgut-Derived Activin Regulates Glucagon-like Action in the Fat Body and Glycemic Control. *Cell metabolism* **25**, 386-399 (2017).
 18. Song, W., Veenstra, J.A. & Perrimon, N. Control of Lipid Metabolism by Tachykinin in *Drosophila*. *Cell reports* **30**, 2461 (2020).
 19. Lemaitre, B. & Miguel-Aliaga, I. The digestive tract of *Drosophila melanogaster*. *Annu Rev Genet* **47**, 377-404 (2013).
 20. Holsopple, J.M., Cook, K.R. & Popodi, E.M. Enteroendocrine cell expression of split-GAL4 drivers bearing regulatory sequences associated with panneuronally expressed genes in *Drosophila melanogaster*. *microPublication biology* **2022** (2022).
 21. Brown, M.R. *et al.* Identification of a *Drosophila* brain-gut peptide related to the neuropeptide Y family. *Peptides* **20**, 1035-1042 (1999).
 22. Wang, Y., Pu, Y. & Shen, P. Neuropeptide-gated perception of appetitive olfactory inputs in *Drosophila* larvae. *Cell reports* **3**, 820-830 (2013).
 23. Wen, T., Parrish, C.A., Xu, D., Wu, Q. & Shen, P. *Drosophila* neuropeptide F and its receptor, NPFR1, define a signaling pathway that acutely modulates alcohol sensitivity. *Proceedings of the National Academy of Sciences of the United States of America* **102**, 2141-2146 (2005).
 24. Wu, Q. *et al.* Developmental control of foraging and social behavior by the *Drosophila* neuropeptide Y-like system. *Neuron* **39**, 147-161 (2003).

Reviewer #3 (Remarks to the Author):

This study provides a quite complete picture of a gut-brain signaling axis in regulating food intake in *Drosophila*. The authors found that neuropeptide F (NPF) secreted from a subset of enteroendocrine cells in the middle midgut acts on a pair of NPFR-expressing enteric neurons, which then projects to the presumptive feeding center to inhibit food intake. Counter intuitively, the authors found that protein-rich diet can inhibit NPF secretion from these EECs, thereby eliciting a feed forward mechanism to promote food intake.

Pros:

The analysis in general is very thorough and with rigor, and most conclusions are quite compelling. The authors have developed several approaches for gut-specific manipulation of EECs, which has not been achieved by the majority, if not all, of the previously published studies in this field.

Cons:

The regulation of gut NPF secretion specifically by amino acid reported here is not consistent with the previously reported observations.

In the mechanistic part, the signaling route independent of AKH+ neurons is also not consistent with the previous observations.

Amino acid sensing by taste sensory organs is known to have a role in appetite regulation, other variables such as gender and mating status may also have an effect on food preference. These factors might complicate the interpretations of some results reported in this study.

The specificity of some GAL4 drivers used in the study warrants further validation.

Specific points:

1. The results from this study suggest the exact opposite functions of NPFs in the brain and the periphery system in controlling feeding behavior, as the reduced food intake were observed in NPF null mutant flies. To strengthen the conclusion about the opposite functions, the authors should perform brain-specific manipulation of NPF side-by-side with the gut-specific manipulations to see if the brain-derived NPF is indeed orexigenic. Perhaps the TCD device described in this study can be adopted to specifically manipulate NPF neurons in the brain.

We thank the reviewer for the advice. Unfortunately, our TCD device is currently unable to specifically heat the head of the fly while exposing the abdomen to the 18°C. This is because, while the head is embedded in the device, the fly will dehydrate and die within

12 hours of heating. To further determine whether NPF secreted from the EECs inhibits feeding while NPF in the brain promotes feeding, we specifically expressed NPF in the EECs in the context of the *NPF¹* homozygous mutant. Our results showed that NPF re-expressed in the EECs further reduced the feeding level of the NPF mutant (Figure 2p, q), supporting our conclusion that NPF expressed outside the EEC (NPF in the brain) promotes feeding, while NPF in the gut inhibits feeding.

Fig. 2p and q, CAFE assay (p) and dye-based food intake measurement (q) of heterozygous (*NPF^{1/+}*), homozygous *NPF* mutant (*NPF¹*) and EECs-specific NPF recovery under *NPF* mutant condition (*tap^{1.3}-B-Gal4>NPF¹/NPF¹*) flies. $n \geq 18$ (p) and $n \geq 13$ (q) in each genotype. Data are represented as mean \pm SD. Significance was determined using Student's *t* test. * $p < 0.05$, ** $p < 0.01$, **** $p < 0.0001$. n, number of groups performed for quantification of food intake (5 flies in each group) (p), or number of groups (20 flies in each group) performed for quantification of food consumption (q).

2. The authors show co-localization of NPF or tap 1.3B-gal4 with NPFR. However, the previous single cell RNA-seq data showed mutually exclusive expression of NPF and NPFR in EE cells (Guo et al., 2019). Malita, et al, 2022 also demonstrated the expression of NPFR in AstC+ EECs, but not TK+ EEs that express NPF. Therefore, it is worrisome whether the described pattern of NPF/NPFR is correct, and whether tap 1.3B-gal4>NPFR-RNAi can effectively deplete NPFR in EECs. The expression pattern of NPF/NPFR need to be further validated such as by co-immunostaining or by intersectional strategies (such as NPFR-LexA \cap Tap1.3B-Gal4 or NPF-Gal4).

We fully understand the concerns of the reviewer. Indeed, we noticed the single-cell sequencing paper published by the Xi lab, which shows that NPF-expressing EECs and NPFR-expressing EECs are two different cells in a pair of EECs¹, and we were also aware of that the NPFR-Gal4>GFP⁺ cells in the Rewitz lab paper are AstC+ EECs², whereas NPF and TK are supposed to be expressed in AstC⁻ EECs¹. Therefore, we originally thought that NPFR antibody and NPF-GFP reporter should be stained in different cells of a pair of EECs. However, antibody staining showed that NPFR is expressed in NPF-GFP⁺ EEC (Extended data Fig. 9b-d), and the gut NPFR expression was completely abolished in

tap^{1.3}-B-Gal4>NPFR RNAi flies (Fig. 6b). We therefore concluded that NPF and NPFR are expressed in the same EEC.

Extended data Fig. 9 b, Schematic representation of the construction of *NPF-0.7-GFP* line. **c, d**, *NPF-0.7-GFP*⁺ cells were co-stained with NPF (**c, c', c'', c'''**) and NPFR (**d, d', d'', d'''**).

To further validate our results and to follow the reviewer's suggestion, co-immunostaining of NPF and NPFR confirmed that NPF and NPFR are indeed expressed in the same EEC (Extended data Fig. 9a). Furthermore, with an intersectional strategy (*NPFR^{RA/C}-LexA*∧*tap^{1.3}-B-Gal4*)³, we determined that *NPFR^{RA/C}* is expressed in the *tap^{1.3}-B* expressing cells, and those intersectional cells stain positive for NPF (Response Fig. 5). Taken together, our current evidence shows that NPFR and NPF are expressed in the same EEC.

Extended data Fig. 9a NPF and NPFR staining is in the same EEC.

Response Fig. 5 | Intersectional strategy shows that *NPFR^{RA/C}* is expressed in the *tap^{1.3-B}* expressing cells.

a, Diagram of intersectional strategy. **B**, *NPFR^{RA/C}-lexA ∩ tap^{1.3-B}-Gal4* cells are NPF staining positive cells.

3. NPFR is expressed in many tissues and organs, and the authors only examined its role in EECs and in the AKH negative neurons in HCG. Their observation about a pair of AKH negative neurons that mediate NPF signaling from the gut appears to contradict several previous observations (such as Yoshinari et al., 2021 and Malita et al., 2022). The authors may need to examine whether AKH+ neurons in HCG are NPFR+, and whether they also mediate NPF signaling from the gut.

We would like to clarify that *NPFR-Gal4>GFP* positive cells are not all AKH positive. While a proportion of *NPFR>GFP⁺* cells had AKH staining (consistent with Yoshinari et al, 2021 and Malita et al, 2022), there was still a proportion of *NPFR>GFP⁺* cells without AKH staining (now shown in our Extended Data Fig. 10h).

It is worth noting that at certain shooting angles, a confusing conclusion may be reached that AKH staining 100% overlaps with that of *NPFR>GFP* after 3D projection (Response Fig. 6), as the image shown in Fig. 6a of Malita et al. 2022. However, a proportion of *NPFR>GFP⁺* cells without AKH staining can still be identified in both Malita's and our images (yellow arrowheads).

NPFR^{RAVC}-Gal4>mCD8:GFP

Extended Data Fig. 10h, AKH and *NPFR^{RAVC}-Gal4>mCD8:GFP* co-staining shows that some NPFR+ neurons are AKH-expressing cells (white arrowheads), while others are not (yellow arrows).

NPFR^{RAVC}-Gal4>mCD8:GFP

Melita et al. 2022 Fig 6a

a

Response Fig. 6 | At certain shooting angles, AKH staining with *NPFR>GFP* appeared to overlap after 3D projection, which is consistent with the image shown in Fig. 6a of Malita et al. 2022. However, a proportion of *NPFR>GFP*⁺ cells without AKH staining can still be identified in both Malita's and our images (yellow arrowheads).

4. The expression pattern of mGluRs should be determined. Are they expressed specifically in NPF+ EECs?

This is a very good question and we are also curious to know if mGluR expression is specific in the NPF⁺ EEC. Since the mGluR gene is located on chromosome 4, we had not previously tried to knock a tag/T2A-Gal4 into mGluR using Crispr-Cas9. We then discovered that Hugo Bellen's lab and the Fourth Chromosome Resource Project had inserted a Trojan GAL4 or a double-headed (DH) trap cassette into a coding intron of mGluR (Response Fig. 7a)⁴. The DH cassette comprises two independent trapping modules oriented in opposite directions: 1) a protein-trap (PT) module for tagging endogenous proteins, composed of a splice acceptor site, a EGFP-FIAsH-StrepII-TEV-3xFlag ("GFSTF") multi-tag cassette and a splice donor site, and 2) a gene-trap (GT) module designed to truncate the endogenous gene into which it inserts, as it contains T2A-GAL4 followed by a polyadenylation signal. Translation should skip at the T2A sequence, truncating the endogenous protein and producing a separate GAL4 protein. Depending on the orientation in which the DH cassette is inserted relative to the direction of transcription of an endogenous gene, the inserted element can either be in the 'protein-trap' or 'gene-trap' orientation for that gene.

We ordered and studied a Trojan Gal4 insertion and all the 4 DH insertions at the mGluR intron site, as follows.

1: Mi{Trojan-Gal4.un}mGluR[MI02169-TG4.un]

2: Mi{DH.1}mGluR[MI02169-DH.GT-TG4.1]

3: Mi{DH.2}mGluR[MI02169-DH.GT-TG4.2]

4: Mi{DH.1}mGluR[MI02169-DH.PT-GFSTF.1]

5: Mi{DH.2}mGluR[MI02169-DH.PT-GFSTF.2]

Unfortunately, while three Gal4 lines (MI02169-TG4, MI02169-DH.GT-TG4.1, and MI02169-DH.GT-TG4.2) had GFP expression in the brain when driving a strong UAS-GFP reporter (20xUAS-6xGFP, 120xGFP), all the five lines failed to drive GFP expression in the EEC (Response Fig. 7 b-f). We also examined whether there was GFP or Flag expression in the four DH lines (GFSTF), but detected no GFP or Flag expression in either the brain or the EEC.

Not only our genetic results suggest that mGluR should have expression in the EEC, but in the Nature paper published by the Jasper lab in 2015, their data also suggested that mGluR should be expressed in the ISC⁵. However, none of the 5 insertion lines showed detectable expression in the midgut. Since all 5 insertion lines were inserted at the same intron site (Response Fig. 7a), we speculate that either T2A-Gal4 inserted at this site does not reflect mGluR expression in EEC/ISC, or MiMIC inserted at this site is too weakly

expressed in EEC/ISC to reach the detection threshold. Therefore, we cannot answer this question at the moment, and will need to insert other site-specific Marker/T2A-Gal4 KIs in the future to verify the exact cellular expression pattern of mGluR.

Response Fig. 7 | mGluR-Gal4 has no expression in EECs

a, The genomic map (Flybase: <https://flybase.org/reports/FBti0196309>) shows that these five mGluR-Gal4 are located at the same site. **b-f**, The expression pattern of five different *mGluR-Gal4*>*120xGFP* in brains (**a-e**) and EECs (**a'-e'**). **a''-e''** show the GFP channel of **a'-e'**. Scale bars, 20 μ m unless otherwise specified.

5. The authors showed that the frequency of the Ca⁺⁺ oscillation, rather than absolute Ca⁺⁺ concentration, is associated with the secretory capacity of EECs. In order to reliably evaluate the intensity of the calcium oscillation on the secretory capacity, a reference signal such as UAS-RFP should be introduced to the calcium imaging assay, and the GCaMP fluorescence intensity should be measured as normalized to the RFP fluorescence intensity.

Following the reviewer's suggestion, we have used *tap^{1.3}-B-Gal4* to drive the expression of both *UAS-tdTomato* and *UAS-GcaMP*, and further normalized calcium signal intensity to the tdTomato. Due to time and workload constraints, we only compared the difference in calcium oscillations upon cornmeal feeding and upon addition of 1% L-Glu (Extended Data Fig. 7b-d), as well as the changes in calcium oscillations upon knockdown of *atp empty* (control), *stim*, *SERCA*, *PMCA* and *IP3R* in *tap^{1.3}-B-Gal4* expressing cells (Extended Data Fig. 7e-j). Our results show that normalizing calcium signal intensity to the tdTomato signal as a reference does not alter the pattern of calcium oscillations in EECs, further validating the reliability of our previous results.

Extended Data Fig. 7 | Dietary L-Glu inhibits calcium oscillations of EECs.

b-d, Representative heatmap records of GcaMP/tdTomato ratio of 10 individual EECs (**b,c**) and quantification of calcium peaks in EECs (**d**) within 10 minutes (300 frames) under cornmeal food (**b**) and 1% L-Glu (**c**) feeding conditions. $n \geq 21$ in each group (**d**). **e-j**, Representative heatmap records of GcaMP/tdTomato ratio of 10 individual EECs (**e-i**) and quantification of calcium peaks in EECs (**j**) of flies with the indicated genotypes within 10 minutes. $n \geq 19$ (**j**) in each group.

6. Please show separate channels for Figures 2D-F and Figures 6b-c.

Sure. We made the following changes:

Fig. 2d, *tap^{1.3}-B-Gal4>GFP* cells were co-stained with NPF.

Extended Data Fig. 3e, *tap^{1.3}-B-Gal4>GFP* cells were co-stained with Tk.

Fig. 2e, Emerging EECs were co-stained NPF in *esg^P>scute^{RNAi}* midguts after 10 days recovery.

Extended Data Fig. 3f, Emerging EECs were co-stained Tk in *esg^P>scute^{RNAi}* midguts after 10 days recovery.

Fig. 6b, NPFR staining in EECs of *control* and *tap^{1.3}-B-Gal4>NPFR^{RNAi}* flies.

7. Extended Data Fig.1 vs. Extended Data Fig.3, knocking down NPF by *tap1.3B-gal4* suppresses overeating and promotes energy wasting, but depleting the entire EECs only causes overeating. Could authors speculate on what causes the difference?

This is a similar question as part of point #6 raised by Reviewer 2. Our current

understanding is that EECs, as a population of cells, have very complex and diverse functions in food digestion, energy sensing and metabolic regulation. And it is worth noting that EECs can regulate opposite metabolic processes by secreting different neuropeptides depending on the physiological environment. For example, in the data from ourselves (Extended Data Fig. 4 k-p), Rewitz lab⁶ and Niwa lab⁷, EECs promote lipid anabolism in adipose tissue by secreting NPF. However, in addition to NPF, EECs also secrete other neuropeptides such as AstA and AstC, both of which have been reported to promote the release of the adipokinetic hormone (Akh), thereby mobilizing systemic energy reserves^{8, 9}. We therefore speculate that although EEC-less flies lost the ability to respond to changes in feeding conditions, they may still exhibit a similar level of metabolic phenotypes to those of the control flies under laboratory SCD rearing conditions.

In addition, it has been reported that EEC-less flies have a substantial reduction in intestinal digestive enzyme activities, including trypsin, chymotrypsin, aminopeptidase and acetate esterase¹⁰. Reduced digestive capacity is consistent with the fact that EEC-less flies eat more but do not show significant changes in metabolism.

Finally, we thank all the reviewer for their positive comments and constructive suggestions on our work.

1. Guo, X. *et al.* The Cellular Diversity and Transcription Factor Code of Drosophila Enteroendocrine Cells. *Cell reports* **29**, 4172-4185.e4175 (2019).
2. Malita, A. *et al.* A gut-derived hormone suppresses sugar appetite and regulates food choice in Drosophila. *Nature metabolism* (2022).
3. Chen, D. *et al.* Genetic and neuronal mechanisms governing the sex-specific interaction between sleep and sexual behaviors in Drosophila. *Nature communications* **8**, 154 (2017).
4. Lee, P.T. *et al.* A gene-specific library for. *eLife* **7** (2018).
5. Deng, H., Gerencser, A.A. & Jasper, H. Signal integration by Ca(2+) regulates intestinal stem-cell activity. *Nature* **528**, 212-217 (2015).
6. Malita, A. *et al.* A gut-derived hormone suppresses sugar appetite and regulates food choice in Drosophila. *Nature metabolism* (2022).
7. Yoshinari, Y. *et al.* The sugar-responsive enteroendocrine neuropeptide F regulates lipid metabolism through glucagon-like and insulin-like hormones in Drosophila melanogaster. *Nature communications* **12**, 4818 (2021).
8. Li, Y.G. *et al.* Gut AstA mediates sleep deprivation-induced energy wasting in Drosophila. *Cell Discov* **9** (2023).
9. Kubrak, O. *et al.* The gut hormone Allatostatin C/Somatostatin regulates food intake and metabolic homeostasis under nutrient stress. *Nature communications* **13**, 692 (2022).
10. Amcheslavsky, A. *et al.* Enteroendocrine cells support intestinal stem-cell-mediated homeostasis in Drosophila. *Cell reports* **9**, 32-39 (2014).

REVIEWER COMMENTS

Reviewer #1 (Remarks to the Author):

I would like to thank the authors for addressing all my concerns.

Reviewer #2 (Remarks to the Author):

I appreciate the authors' considerable effort in addressing my concerns. While I am generally satisfied with the revision, two points remain incomplete:

1. To establish a clear distinction between the present work and Kim Rewitz's research, it is essential that they demonstrate, at the very least, that NPFR+ AKH+ cells are not implicated in food intake under their experimental conditions.

2. In extended data figure 10f, the connection between SEZ and the projection of NPFR-ENS+ neurons is not clearly discernible. I highly recommend employing the MultiColor-FlpOut approach, which is more suitable than using GFP alone for labeling fine neuronal processes and accurately tracking the exact projections of neurons.

Reviewer #3 (Remarks to the Author):

The authors have addressed all of my previously comments with satisfaction, and I believe the paper is now ready for publication.

We would like to thank Reviewer #2 for their constructive suggestions for improvement. Below we provide a point-by-point response (in blue) to the Reviewers' feedback.

Reviewer #1 (Remarks to the Author):

I would like to thank the authors for addressing all my concerns.

Reviewer #2 (Remarks to the Author):

I appreciate the authors' considerable effort in addressing my concerns. While I am generally satisfied with the revision, two points remain incomplete:

1. To establish a clear distinction between the present work and Kim Rewitz's research, it is essential that they demonstrate, at the very least, that NPFR+ AKH+ cells are not implicated in food intake under their experimental conditions.

Response: Thank you for your suggestions. Consistent with the results published by Malita, A. et al., knockdown of NPFR in corpora cardiaca using AKH-Gal4 with 5% sucrose as a food caused an increase in *Drosophila* feeding levels (Response Fig.1). Our results validate the conclusion of Kim Rewitz's group that NPFR+ AKH+ cells may regulate feeding through glucagon-like signaling¹. However, knockdown of NPFR in AKH+ cells did not result in a significant increase in food intake when 1% L-Glu was added to 5% sucrose food compared to the control group (Response Fig.1). This suggests that knockdown of NPFR in AKH+ cells does not cause a further increase in feeding when L-Glu inhibits NPF release in EECs.

We do not consider that we are distinguished from the Kim Rewitz's work by whether or not NPFR+AKH+ cells are involved in feeding regulation. Given that NPFR is expressed both in AKH+ cells and in the enteric neurons, EEC-secreted NPF is fully capable of regulating feeding through both metabolic and enteric neural pathways.

Response Fig. 1 | Knocking down of *NPFR* in *AKH*⁺ cells did not increase food intake under high L-Glu feeding conditions.

Food intake of control (+>NPFR^{RNAi}) and AKH-Gal4>NPFR^{RNAi} flies under 5% sucrose and high L-Glu (5% sucrose+ 1% L-Glu) feeding conditions. n ≥ 19 in each group. Data are represented as mean ± SD. Significance was determined using Student's t test. n.s., non-significant; *p < 0.05, **p < 0.01, ****p < 0.0001. n, number of groups (5 flies in each group) performed for quantification of food intake.

2. In extended data figure 10f, the connection between SEZ and the projection of NPFR-ENS⁺ neurons is not clearly discernible. I highly recommend employing the MultiColor-FlpOut approach, which is more suitable than using GFP alone for labeling fine neuronal processes and accurately tracking the exact projections of neurons.

Response: Thank you for your suggestions. We have stochastically labelled two NPFR⁺ enteric neurons using the MultiColor-FlpOut technique². Here we show an example where one neuron was labelled with Flag and the other with HA. These two neurons have similar but diverse projections to the SEZ (Response Fig. 2). We have added this part of the result as Extended Data Fig. 10g.

g

NPFR^{ENS}-Gal4, nsyb-Flp, UAS>stop>FLAG, UAS>stop>HA

Response Fig. 2 | Stochastic labeling by MultiColor-FlpOut technique (*NPFR^{ENS}-Gal4, nsyb-Flp, UAS>stop>FLAG, UAS>stop>HA*) shows that these two *NPFR^{ENS}* neurons have similar but diverse projections to the SEZ. Yellow arrowheads point to the location of neuron bodies.

Reviewer #3 (Remarks to the Author):

The authors have addressed all of my previously comments with satisfaction, and I believe the paper is now ready for publication.

Finally, we thank all the reviewer for their positive comments and constructive suggestions on our work.

1. Malita, A. *et al.* A gut-derived hormone suppresses sugar appetite and regulates food choice in *Drosophila*. *Nature metabolism* (2022).
2. Nern, A., Pfeiffer, B.D. & Rubin, G.M. Optimized tools for multicolor stochastic labeling reveal diverse stereotyped cell arrangements in the fly visual system. *Proceedings of the National Academy of Sciences of the United States of America* **112**, E2967-2976 (2015).

REVIEWERS' COMMENTS

Reviewer #2 (Remarks to the Author):

I appreciate the authors' efforts to address my remaining concerns, and I believe that the manuscript is now ready for publication.